# Spatial probabilistic mapping of metabolite ensembles in mass spectrometry imaging

Denis Abu Sammour [1,2], James L. Cairns [1,3], Tobias Boskamp [4,5], Christian Marsching[1,4], Tobias Kessler [6,7], Carina Ramallo Guevara [1], Verena Panitz[7,8], Ahmed Sadik [7,9], Jonas Cordes [10], Stefan Schmidt [1], Shad A. Mohammed [1,2], Miriam F. Rittel [1,2], Mirco Friedrich[2,11,12], Michael Platten [2,11,12], Ivo Wolf [10], Andreas von Deimling [13,14], Christiane A. Opitz[7,8], Wolfgang Wick [6,7] & Carsten Hopf [1,2,3] ✉

Mass spectrometry imaging vows to enable simultaneous spatially resolved investigation of hundreds of metabolites in tissues, but it primarily relies on traditional ion images for non-data-driven metabolite visualization and analysis. The rendering and interpretation of ion images neither considers non-linearities in the resolving power of mass spectrometers nor does it yet evaluate the statistical significance of differential spatial metabolite abundance. Here, we outline the computational framework molecularR (https://github.com/CeMOS-Mannheim/molecularR) that is expected to improve signal reliability by data-dependent Gaussian-weighting of ion intensities and that introduces probabilistic molecular mapping of statistically significant nonrandom patterns of relative spatial abundance of metabolites-of-interest in tissue. molecularR also enables cross-tissue statistical comparisons and collective molecular projections of entire biomolecular ensembles followed by their spatial statistical significance evaluation on a single tissue plane. It thereby fosters the spatially resolved investigation of ion milieus, lipid remodeling pathways, or complex scores like the adenylate energy charge within the same image.

Mass spectrometry imaging (MSI) has evolved into a label-free core technology for visualization and spatially resolved analysis of digested proteins, drugs, glycans, and metabolites, including lipids, in basic, clinical, and pharmaceutical research[1,2]. Despite enormous advances in speed, sensitivity, and spatial resolution of MSI instruments and despite a growing number of successful MSI applications[3,4], the fundamental concept in MSI data processing, the conversion of raw data into ion images for visualization, spatial interpretation, and molecular analysis, has not changed since the inception of the technology[5]. Ion images, i.e., false color renderings of *m/z* intervals that often contain

[1]Center for Mass Spectrometry and Optical Spectroscopy (CeMOS), Mannheim University of Applied Sciences, Mannheim, Germany. [2]Mannheim Center for Translational Neuroscience (MCTN), Medical Faculty Mannheim, Heidelberg University, Mannheim, Germany. [3]Medical Faculty Heidelberg, Heidelberg University, Heidelberg, Germany. [4]Bruker Daltonics GmbH & Co. KG, Bremen, Germany. [5]Center for Industrial Mathematics, University of Bremen, Bremen, Germany. [6]Clinical Cooperation Unit Neurooncology, German Cancer Consortium, German Cancer Research Center, Heidelberg, Germany. [7]DKTK Metabolic Crosstalk in Cancer, German Consortium of Translational Cancer Research (DKTK), German Cancer Research Center (DKFZ), Heidelberg, Germany. [8]Department of Neurology and National Center for Tumor Diseases, Heidelberg University Hospital, Heidelberg, Germany. [9]Faculty of Bioscience, Heidelberg University, Heidelberg, Germany. [10]Faculty of Computer Science, Mannheim University of Applied Sciences, Mannheim, Germany. [11]Department of Neurology, Medical Faculty Mannheim, Heidelberg University, Mannheim, Germany. [12]DKTK Clinical Cooperation Unit Neuroimmunology and Brain Tumor Immunology, German Cancer Research Center (DKFZ), Heidelberg, Germany. [13]Department of Neuropathology, University Hospital Heidelberg, Heidelberg, Germany. [14]DKTK Clinical Cooperation Unit Neuropathology, German Cancer Research Center (DKFZ), Heidelberg, Germany. ✉e-mail: c.hopf@hs-mannheim.de

multiple unassigned peaks, like other types of images, can be prone to technical artifacts[6] and user perception bias[7]. Filtering, cross-normalization, and collective judgment approaches, among others, have been suggested as remedies[8–11]. Nevertheless, procedures for the processing of raw data into ion image renderings typically do not include built-in methods that account for mass accuracy and the resolving power- and instrument-dependent peak width at peak-of-interest (POI) *m/z*. They normally use the sum of ion intensities of all peaks present in an end user-defined mass range centered on the POI *m/z* instead in a data-independent uniform weighting (i.e., all peaks treated equally) approach (Supplementary Fig. 1a). Recently, some probabilistic concepts in MS-based omics, especially false-discovery rates (FDR) typically assessed by target-decoy searching[12], have been introduced in MS imaging[13]. However, a coherent probabilistic approach for evaluation of the spatial aspect in MSI and for identification of areas with significantly higher relative spatial abundance ("hotspots"; i.e., nonrandom patterns of spatial accumulation) or deficiency ("coldspots"; i.e., nonrandom patterns of spatial depletion) of a single metabolite-of-interest (MOI) in tissue is lacking.

Likewise, methods for probabilistic mapping of metabolite ensembles comprising tens or hundreds of named MOIs such as lipid classes[14], amino acids[15], or nucleotides[16] that could be useful as quantitative MSI scores (e.g., overall energy charge score) are required. Moreover, techniques for spatial probing of global tissue characteristics such as ion milieu, the degree of lipid unsaturation, or even the distribution of entire lipid classes as a function of tissue morphology are much needed. Biomedical or pharmaceutical scientists could select such molecular ensembles based on their research topics, e.g., phosphatidylcholines containing saturated fatty acids. Molecular ensembles would complement peak lists (POI) interpreted by MS experts, and they could be collectively interrogated to probe, for example, entire molecular pathways in MSI, as is required for translational applications.

Here, we report the computational framework *moleculaR* that suggests peak width-dependent Gaussian-weighting for improved reliability of metabolite/lipid signals in MSI and that introduces molecular probabilistic maps (MPMs) based on Kernel density estimation against a complete spatial randomness model of the same dataset. This procedure enables the plotting of probabilistic "hotspot" and "coldspot" contours for any given MOI, independent of how an end user may perceive its spatial relative abundance or deficiency. The framework *moleculaR* also allows for comparisons of different tissues (cross-tissue MPMs) and for collective projections of metabolite ensembles onto a single tissue plane, followed by computation of collective-projection probabilistic maps (CPPMs). These are MPM "hotspot" and "coldspot" contours for complex metabolite examples, e.g., alkali metal adducts of lipids as ion milieu indicators, adenylate energy charge as a possible correlate of metabolic cancer hotspots or entire lipid classes as the basis for analysis of lipid remodeling pathways. The software may therefore contribute to the expanded application of MSI by non-mass spectrometrists and to more applications of the technology in translational science.

## Results

### Data-dependent Gaussian-weighting of ion intensities

Ion images currently used in matrix-assisted laser desorption/ionization (MALDI) MSI do not strictly represent the ion intensity of a single observed peak-of-interest (POI) *m/z*. Instead, to compute ion intensities, all peaks in a user-defined mass range centering on a POI are weighted equally and then summed up ("uniform weighting approach"; Supplementary Fig. 1a). This central POI can be picked using suitable algorithms (POI-centric approach). It is subsequently annotated in a false-discovery-rate (FDR)-controlled fashion[13] by computational platforms like METASPACE (https://metaspace2020.eu) that estimate if the POI may correspond to a certain biologically

relevant metabolite-of-interest (MOI), i.e., a database entry with known *m/z* such as the potassium adduct of heme [Heme+K]$^+$ (Supplementary Fig. 1). POI *m/z* and MOI *m/z* typically differ, and the molecular identity of POIs is a statistical consideration.

Many biomedical scientists, however, do not have this POI-centric analytical perspective, but rather welcome support in their quest to visualize and analyze single MOI or ensembles of MOI, such as the oncometabolite R-2-hydroxyglutarate or tryptophan and its catabolites, some of which promote immunosuppression in glioblastoma[17,18] (MOI-centric perspective). Therefore, we aimed to introduce a spatially aware perspective that systematically analyzes and visualizes if biomedically relevant MOI has a statistically validated relative spatial abundance in defined areas of a heterogeneous tissue slice. To this end, we propose molecular probabilistic maps (MPMs) that may complement ion images for data-driven computational spatial analysis of MOI by MOI-centric spatial statistical testing (Supplementary Fig. 2). MPMs are based on the assumption that for any observed (MOI-matched) POI, an increase in its spatial autocorrelation[19], i.e., systematic spatial variations of its intensities, could be an indicator of a biological process or a spatially confined tissue morphology linked directly or indirectly to this MOI. Instead of estimating this correlation intensity for the entire image, the MPM approach localizes areas of heightened relative "activity" in terms of points' spatial densities and signal intensities in a spatial point pattern (SPP) representation of the MSI raw data for any given MOI *m/z*.

SPP representations are widely applied in statistical data interpretation in other fields of biomedical imaging, such as digital histology[20]. This transformation of MSI raw data into SPPs features data—rather than user-defined mass-windows: First, full-width-at-half-maximum (FWHM) is plotted against *m/z* for at least one randomly chosen single-pixel full (profile) spectrum. The nonlinear mass resolving power across a mass range is calculated from that spectrum (or several spectra) for any given experiment (Fig. 1, Supplementary Fig. 1b) and any given mass spectrometer such as the three platforms used here, magnetic resonance MS (MRMS) based on Fourier transform ion cyclotron resonance (FTICR), linear time-of-flight (TOF) MS or orthogonal trapped ion mobility spectrometry (tims) TOF MS (Supplementary Figs. 3 and 4). Experimental peak width is subject to multiple influences in a sample that do cause variance. Nevertheless, FWHM curves that were fit via locally estimated scatterplot smoothing (LOESS) based on a single or 100 randomly chosen full spectra for positive and negative ion modes were very similar for two replicate tissue datasets (Supplementary Fig. 3). For any metabolite that an MSI user may be interested in, a Gaussian centered on that MOI whose standard deviation $\sigma_G$ is computed based on estimates of FWHM at the MOI's *m/z*. All observed peaks within this data-dependent Gaussian mass-window are then Gaussian-weighted, summed up, and transformed into an SPP (Fig. 1). While other peak shapes might be considered, Gaussians have frequently been used in MSI[21–23].

To explore the effects of Gaussian mass-window weighting, we "contaminated" data for an example MOI, phosphatidylethanolamine PE(20:1)[M+Na]$^+$ (monoisotopic *m/z* 544.3009), by computationally spiking in an interferent *m/z* at successive mass intervals (multiples of $\sigma_G$) away from MOI *m/z* (Supplementary Fig. 5). Compared to traditional uniform mass-window weighting, Gaussian-weighting removed interference noise (modeling proximal background signals) more effectively, as indicated by the computed mean squared error (MSE) (Supplementary Fig. 5b) and by visual inspection (Supplementary Fig. 5c). The efficiency of handling such spiked interferences was clearly dependent on $\sigma_G$, suggesting that it would be most effective in high-resolution MSI. To test this further, we generated lipid MSI datasets of adjacent sagittal mouse brain sections for a cross-platform comparison using FTICR-, timsTOFflex- and MALDI-TOF mass spectrometers (Supplementary Fig. 6–8). Two almost isobaric MOIs, phosphatidylserine PS(40:6)[M−H]$^-$ (*m/z* 834.5290) and

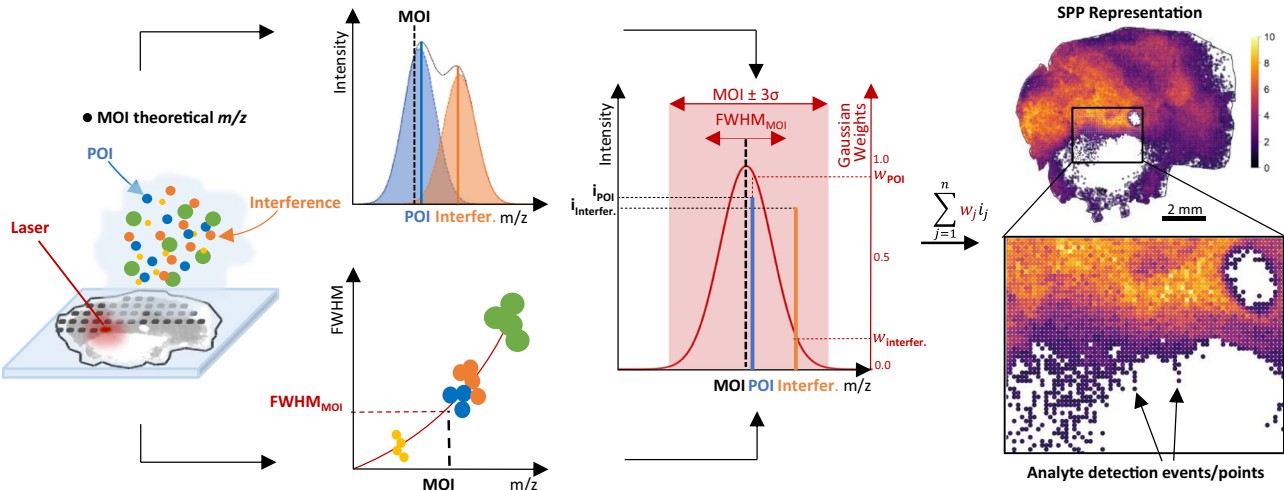

**Fig. 1 | Data-dependent metabolite-of-interest (MOI)-centric Gaussian-weighting of ion intensities and transformation into a spatial point pattern (SPP) representation of the MOI.** Full-width-at-half-maximum (FWHM) values are computed for all peaks of a randomly chosen full-profile mass spectrum (Supplementary Fig. 2a, b) and curve-fitted to describe FWHM as a function of $m/z$. For any MOI $m/z$ (dashed black line), a Gaussian envelope is computed whose $\sigma_G$ is inferred from the estimated FWHM at MOI $m/z$. All observed peaks (POI: solid blue line; interference: solid orange line), which fall within the span of the calculated Gaussian envelop centered on MOI, are Gaussian-weighted (projections onto Gaussian envelope), thereby down-weighting proximal interfering signals: The further the measured $m/z$ (= POI or interference) from the theoretical $m/z$ (= MOI), the lower the weight it receives in the final SPP representation.

C20:0 sulfatide (3′-sulfo)GalCer(38:1)[M−H]⁻ ($m/z$ 834.5770), were well separated by FTICR-MSI, but the separation was improved by Gaussian mass-window versus uniform "sum" weighting. Interestingly, for PS(40:6)[M−H]⁻ no difference between Gaussian and uniform weighting was observed, as the apparently "interfering" peaks represent the same metabolite and can be attributed to "side lobe" signals that can accompany FTICR runs with longer transients (Supplementary Fig. 6[24]). They are expected to yield identical images. The impact of Gaussian-weighting was most evident in high-resolution orthogonal timsTOF MS data (Supplementary Fig. 7), whereas it did not improve a substandard MALDI-MSI-TOF measurement of the same tissue (Supplementary Fig. 8). Uniform weighting was ineffective in both cases. Taken together, we concluded that peak FWHM-dependent Gaussian-weighting, which does not affect all sources of noise or batch effects in MSI[6], did improve signal reliability and introduced an MOI-centric down-weighting of neighboring signals.

## Molecular probabilistic maps for metabolite hotspot/coldspot contours

To evaluate whether point intensities in SPPs and spatial relative accumulation or depletion of a given MOI in tissue sections were statistically significant, a complete spatial randomness (CSR) model of that MOI was created by random spatial permutations of MOI points (see "Methods"), which was then used as the spatial null hypothesis for significance testing (Fig. 2a; Supplementary Fig. 9).

For kernel density estimations (KDE) via an isotropic Gaussian with an MOI-specific bandwidth estimation (Supplementary Fig. 9–11 and "Methods"), the intensity distribution of the CSR density image was expected and observed to converge toward a normal distribution with increasing kernel size (Supplementary Fig. 12). This then forms the basis for inferring intensity cutoffs, beyond which the intensities of MOI's density image are unlikely to occur if generated by a random spatial process (Fig. 2a, Supplementary Fig. 9). More precisely, for each pixel intensity value in the MOI's spatial density function, $\rho_{MOI}(x,y)$, a lower- and an upper-tail $P$-value is computed based on the null distribution $f_{CSR}(k)$ resulting in two spatial maps of lower and upper-tail $P$-values $P_{lwr}(x,y)$ and $P_{upr}(x,y)$, respectively (Supplementary Fig. 9b). These $P$-values are then Benjamini–Hochberg-corrected[25]. Spatial null-hypothesis significance testing is carried out against a significance level $\alpha$ of 0.05. Consequently, an MOI's MPM hotspot and coldspot contours are accordingly defined as locations where the null hypothesis is rejected for the upper- or lower-tail corrected $P$-values, thus signifying areas of significant MOI relative spatial abundance and deficiency, respectively (Fig. 2a, Supplementary Fig. 9). MPMs are therefore composite representations of an MOI's spatial distribution on a raster grid with data-dependent Gaussian-weighted intensities (Fig. 1) and superimposed hotspot and/or coldspot contours indicating areas of statistically significant nonrandom spatial patterns of MOI intensities.

The bandwidth estimation for KDE is a critical part of the MPM computational workflow since too low or too high a bandwidth would lead to under- or overestimation, respectively, of areas of significantly different MOI relative spatial abundance/deficiency (Supplementary Fig. 10). We employed the Moran's I statistic as a measure of autocorrelation and determined the optimal bandwidth for KDE as the "knee" point in the Moran's I vs bandwidth plot[26]. For bandwidths larger than this parameter, increased bandwidth does not result in considerable increases in spatial autocorrelation of the smoothed density image (Supplementary Fig. 10; see "Methods").

In order to test the validity of the proposed MPMs against ground truth, SPP data was simulated based on four different spatial patterns of simulated metabolite ground truths (Fig. 2b, c, Supplementary Fig. 13). For this purpose, points' intensities were sampled from above and below the upper quartile of the empirical intensities of a MALDI-FTICR-MSI measurement of a human glioblastoma (GB) tissue sample (Fig. 2b, c, Supplementary Fig. 13) and a MALDI-TOF-MSI measurement of a human gastrointestinal stromal tumor tissue sample (Supplementary Fig. 14). *moleculaR* was able to reliably localize ground-truth high-abundance areas, i.e., to identify points exhibiting significantly different relative spatial abundance (Fig. 2b, c, Supplementary Figs. 11, 13, 14). High overlaps, as judged by high Dice similarity coefficient (DSC) values, between the estimated hotspot contours and ground-truth shapes, were observed (Supplementary Fig. 11). Interestingly, DSC values were the highest when a bandwidth corresponding to the "knee" point in Moran's I was chosen for KDE, thus further supporting this method for bandwidth estimation (Supplementary Figs. 10 and 11).

Applying the MPM workflow to real MSI data, we first investigated a neurooncology example (FTICR-MSI data). MPMs of two example

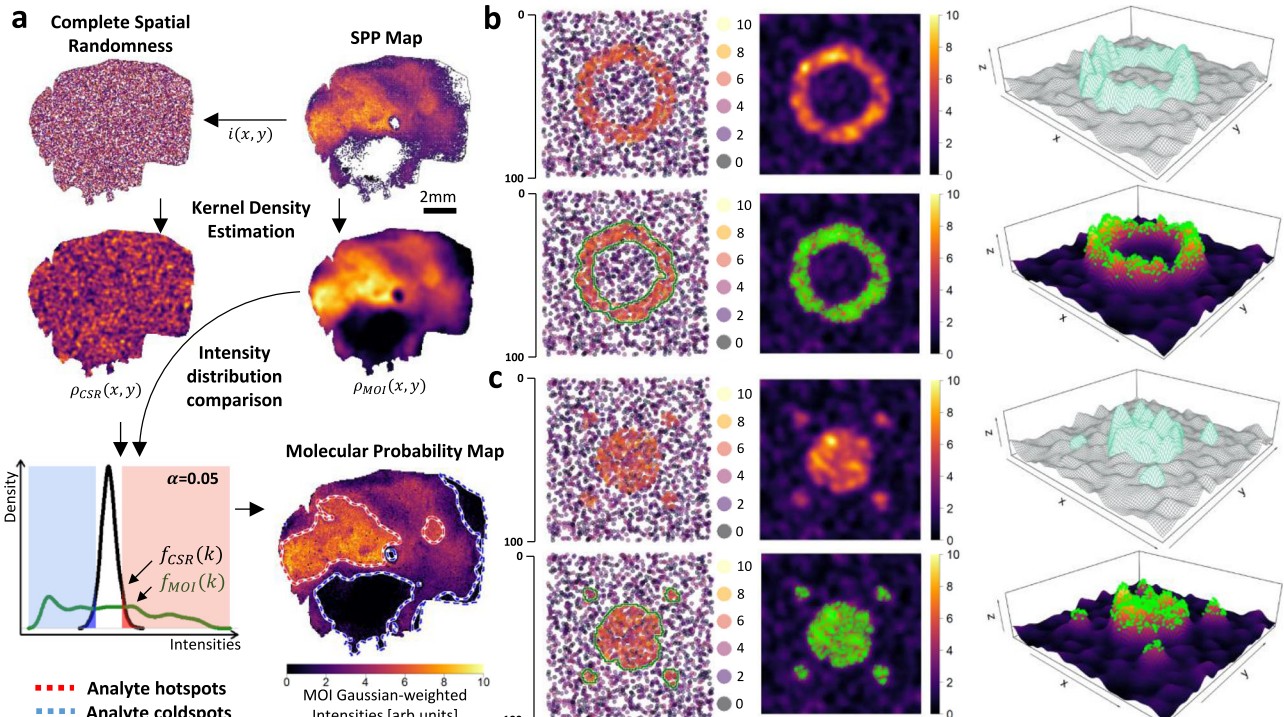

**Fig. 2 | Computational molecular probabilistic map (MPM) workflow and identification of ground-truth in simulated datasets. a** MPM computational workflow. A corresponding complete spatial randomness (CSR) model is created for each metabolite-of-interest's (MOI) spatial point pattern (SPP) with equal spatial point density. Kernel density is estimated for both, thus resulting in spatial density functions, $\rho_{MOI}(x,y)$ and $\rho_{CSR}(x,y)$. The intensity distribution function $f_{CSR}(k)$, which converges to a normal distribution (Supplementary Fig. 12), then serves as the null distribution based on which null-hypothesis testing is carried out for each pixel intensity in $\rho_{MOI}(x,y)$ with a significance level $\alpha$ of 0.05. MPM hotspots (red/white contours) and coldspots (blue/white contours) are accordingly defined as locations where the null hypothesis is rejected for the upper or lower-tail Benjamini–Hochberg corrected P-values (Supplementary Fig. 9), thus signifying areas of significant MOI relative spatial abundance and deficiency, respectively. **b** Simulated uniform Poisson SPP showing a ring-like area of high MOI abundance) with 30 and 20 length units for outer and inner radii, respectively (also see Supplementary Fig. 11d). Intensity values were sampled from above and below the

upper quartile of the empirical intensities of a MALDI-FTICR-MSI measurement of a human glioblastoma tissue sample (Fig. 3a) at *m/z* 544.3009 (PE(20:1)[M+Na]⁺; FDR ≤ 0.2) for the simulated high-MOI and background areas, respectively (simulated high-MOI area: n = -520 with a spatial point density of -0.4 points per unit area; background n = -1950 with a spatial point density of -0.3 points per unit area; mean signal intensity of simulated high-MOI area/mean signal intensity background = -2.3). First row: corresponding spatial density and 3D surface plots. MPM hotspot contours were able to localize the simulated high-MOI area (green contours in the SPP plot) and identify points exhibiting significant relative spatial abundance (green points on density and surface plots; bottom row). **c** Simulated SPP with a central circle of 20 length units radius and four adjacent smaller circles of 5 length units radius as simulated high-MOI areas (central circle area/peripheral circle area = 16; high-MOI area n = -430; background n = -1950; same spatial point density as in (**b**); mean signal intensity of simulated high-MOI area /mean signal intensity background = -2.3).

MOIs, the sphingomyelin SM(d36:4)[M+H]⁺ (*m/z* 725.5592; FDR ≤ 0.10) and the phosphatidylserine PS(36:1)[M−H]⁻ (*m/z* 788.5447; FDR ≤ 0.05), illustrate how spatial probabilistic mapping aids in outlining MOIs' significant relative spatial abundance or deficiency relative to vital tumor regions, as inferred from a neuropathologist's annotation of a fresh-frozen tissue section of GB (Fig. 3a, b). PS(36:1)[M−H]⁻ but not SM(d36:4)[M+H]⁺, possibly arachidonic acid-containing sphingomyelin, had higher relative spatial abundance in the viable tumor (and surrounding) area, as indicated by the respective MPM hotspot contours. Viable tumor rather overlapped to some extent with an MPM coldspot contour of SM(d36:4)[M+H]⁺ that indicated relative spatial deficiency (Fig. 3b). Since true locations of these metabolites in the GB example were unknown, we evaluated two cases, for which MOI distributions had previously been published. Interestingly, MPM mapping of C24:1 sulfatide ((3′-sulfo)GalCer(d42:2)[M−H]⁻; *m/z* 888.6240), previously reported to be present in midbrain and white matter[27] in a sagittal mouse brain section (FTICR-MSI data), revealed MPM hotspot contours that coincided well with these regions as referenced by the Allen mouse brain atlas[28] (Allen Reference Atlas−Mouse Brain [brain atlas]. Available from atlas.brain-map.org; Fig. 3c). Next, we compared *moleculaR* with our reported *PlaquePicker* method[29] for their ability to detect amyloid peptide Aβ₁₋₃₈ (*m/z* 4060.5)−containing plaques in a

brain section of an Alzheimer's disease mouse model (TOF-MSI data). As with *PlaquePicker*, *moleculaR* was able to localize pockets of Aβ₁₋₃₈ well (Fig. 3d). One notable distinction was that *moleculaR* disregarded subsets of single-pixel low-intensity signals, perhaps plaques, which were counted as such by *PlaquePicker*. This could be explained by the fact that *moleculaR* also imposes a spatial co-dependency criterion for intensities, which could effectively filter out spurious outlier single-pixel signals. In cases like this, MOI hotspots visualized in MPMs can be compared with an orthogonal (e.g., optical) method or with what is theoretically expected for the imaged object (e.g., minimum theoretical amyloid plaque diameter; Supplementary Fig. 15). A ground truth for evaluation of new computational approaches in MSI is hard to find, but even in more mature omics sciences like proteomics defining experimental ground truths is still cutting-edge science with several emerging concepts[30]. After validating the MPM workflow (1) using simulated data, (2) using a known higher presence of a defined sulfatide in defined brain areas, and (3) using intense but spatially sparse peptide signals previously verified by LC-MS and by comparison with a wild-type mouse, we considered a fourth example. To lend even more credibility to MPM hotspot and coldspot contours for previously uncharacterized MOIs in the GB example, we reanalyzed calibration curves for drug dilution series (imatinib) spotted onto porcine liver

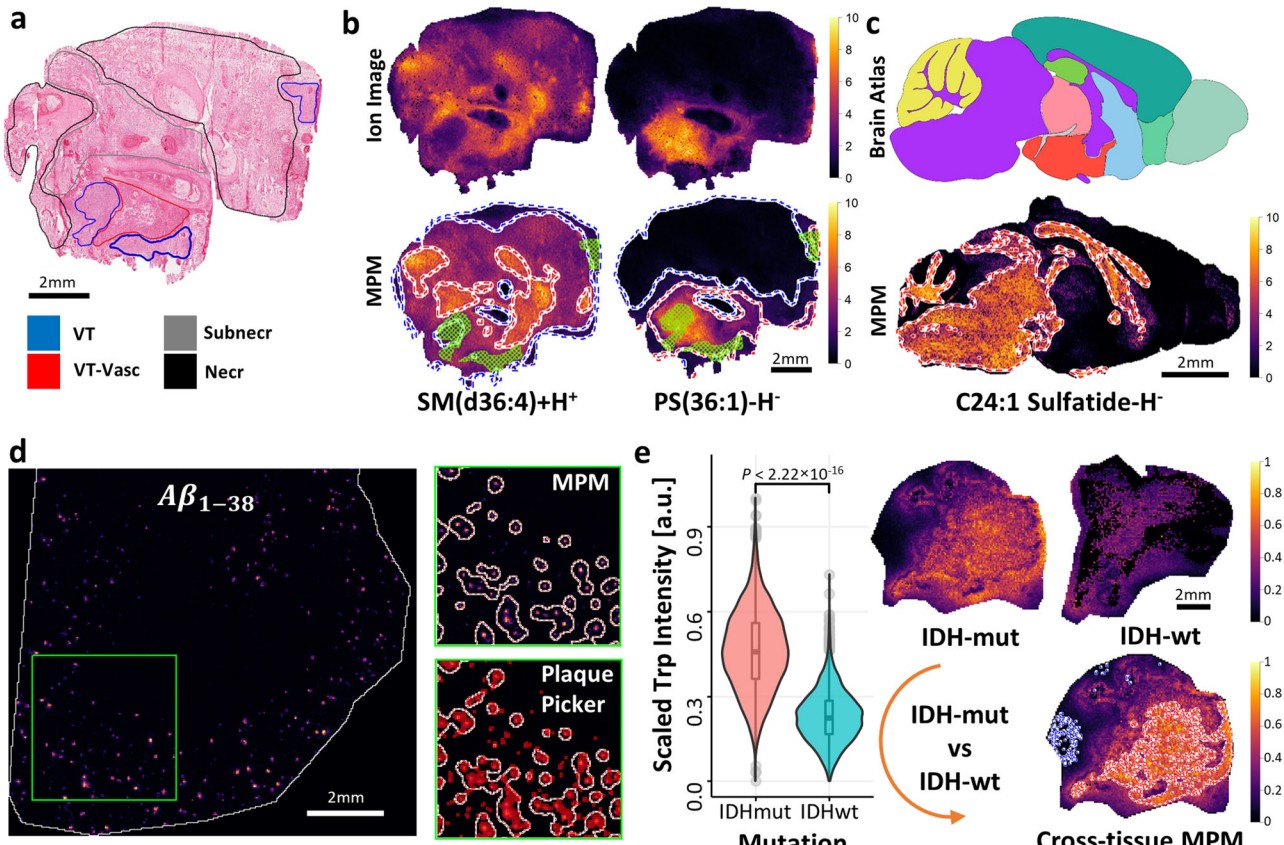

**Fig. 3 | Molecular probabilistic maps (MPM) for spatial probabilistic mapping of MPM hotspot and coldspot contours indicating areas of MOI's increased relative spatial abundance or deficiency in tissue, respectively. a** Hematoxylin and eosin (H&E)-stained human glioblastoma (GB) tissue section (VT: vital tumor; VT-Vasc: vascularized vital tumor; Subnecr: pre-necrotic; Necr: necrotic). **b** Comparison of MALDI-MSI ion images and corresponding MPM hotspot (red/white) and coldspot (blue/white) contours of SM(d36:4)[M+H$^+$] and PS(36:1)[M−H]$^-$ (FDR ≤ 0.10; FTICR-MSI) relative to VT regions (green mesh). **c** MPM of $m/z$ 888.6240 (C24:1 Sulfatide[M−H]$^-$; FTICR-MSI), previously reported to be present in the mid-brain and white matter (fiber tracts) regions, in a sagittal mouse brain section. Hotspot contours correctly outline these regions, as referenced by purple areas in a brain atlas example (adapted from the Allen Reference Atlas−Mouse Brain; atlas.brain-map.org). **d** MPM of amyloid peptide Aβ$_{1-38}$ ($m/z$ 4060.5; TOF-MSI) in plaques in an Alzheimer's disease mouse model (adapted from ref. [29]). MPM hotspot contours correctly localized pockets of Aβ$_{1-38}$-containing amyloid plaques, as

referenced by the previously reported *PlaquePicker* method[29] (pixels highlighted in red). **e** Cross-tissue MPMs (CT-MPMs) enable spatially aware comparison of tryptophan [Trp−H]$^-$ in isocitrate dehydrogenase-mutant (IDH-mut) glioma (test tissue) with IDH-wild-type (IDH-wt) glioma (reference tissue). Test tissue intensities, which display a nonrandom spatial distribution and that significantly differ from those of the reference tissue, are designated as having significant cross-tissue relative spatial abundance/deficiency. This MPM workflow variant enables spatial statistical testing where normally only pooled Trp ion intensities are used and localization of MOI is disregarded for statistical comparisons (exemplified here by box/violin plots; $P < 2.22 \times 10^{-16}$; two-sided Wilcoxon rank-sum test; $n = 3195$ and $n = 3480$ detected signals for IDH-mut and IDH-wt glioma samples, respectively; adapted from ref. [17]). Boxplots indicate median (middle line), 25th, 75th percentile (box) and whiskers which extend to the most extreme data point which is no more than 1.5 times the length of the box away from the box. Source data are provided as a Source Data file.

using the MPM workflow (FTICR-MSI)[31]. Interestingly, following MPM processing, the dilution series plots showed higher linearity, as suggested by higher $R^2$ and $x$-exponent of the linear and nonlinear curve fits, respectively (Supplementary Fig. 16). This initial data suggests that the use of the MPM workflow in quantitative MSI may be worth exploring more systematically. It should be noted that MPM contours could, in principle, be calculated for data from various MSI platforms (Supplementary Figs. 6−8, 17). Our initial experiments suggest that it might be most meaningful for high-mass-resolution MSI, especially for MOIs that are in close proximity or have other proximal background signals.

One important application of MPMs is the spatially aware comparison of drug or metabolite distribution in test versus reference tissues, e.g., those dosed with drugs or carrying mutations versus controls. As an example, we reanalyzed data that we had published earlier. For instance, such cross-tissue MPMs (CT-MPMs; Supplementary Figs. 2, 18−20) can map statistically validated significant cross-tissue relative spatial abundance of immunosuppression-associated tryptophan (Trp) in isocitrate dehydrogenase-mutant (IDH-mut)-

compared to IDH-wild-type (IDH-wt) glioma[17] as test and reference tissues, respectively. Cross-tissue MPMs thereby enable spatial probabilistic comparisons of two tissues where currently often only comparative boxplots of pooled MOI signals are used that disregard any, sometimes important, information on the spatial distribution of the MOI (Fig. 3e, Supplementary Fig. 18). This is achieved by first finding areas of significant relative spatial abundance/deficiency (i.e., MOI hotspots/coldspots; here MOI: [Trp−H]$^-$) in the test tissue. Then all pixel intensities of the test tissue (IDH-mut glioma) are tested against the empirical cumulative distribution function inferred from the pixel intensities of the reference tissue (IDH-wt glioma). Test tissue intensities, which reject both the spatial null hypothesis ("spatial distribution of MOI is random") and the test-vs-reference intensity distributions null hypothesis ("no significant difference between intensity distributions in test and reference tissues"), are designated as having significant cross-tissue relative spatial abundance/deficiency. In other words, CT-MPMs identify areas of the test tissue, which exhibit a statistically significant nonrandom spatial MOI abundance/deficiency pattern and contain intensities that are unlikely to belong to the

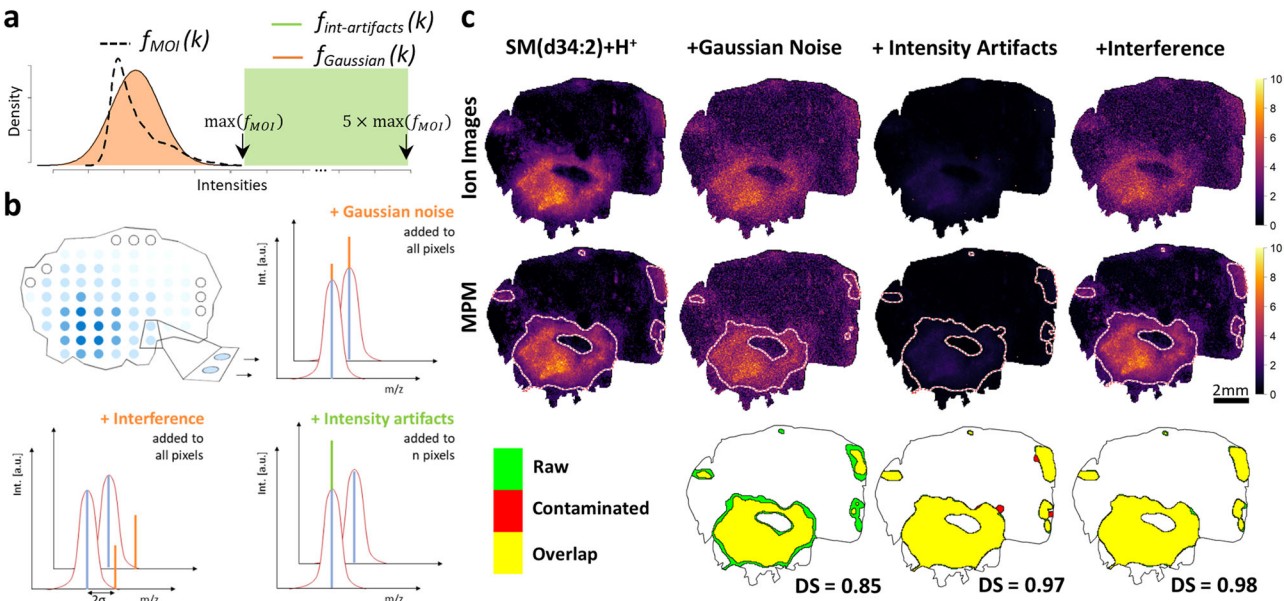

**Fig. 4 | Molecular probabilistic maps (MPMs) are robust against various forms of artificially added noise. a** Schematic representation of ion intensity distributions of a typical metabolite-of-interest (MOI) ($f_{MOI}(k)$; dashed black curve) and of the corresponding Gaussian distribution ($f_{Gaussian}(k)$; orange) from which artificial Gaussian noise and interference noise (**b**) were sampled. Mean and standard deviation of $f_{Gaussian}(k)$ are equal to those of $f_{MOI}(k)$. $f_{int\text{-}artifacts}(k)$ is a uniform rectangular distribution whose range exceeds the range of $f_{MOI}(k)$ (see dotted *x*-axis). **b** Gaussian noise is sampled from $f_{Gaussian}(k)$ and is added for all pixels to the raw signal of MOI *m/z* present in each pixel. Interference noise is also sampled from $f_{Gaussian}(k)$ but is added for all pixels arbitrarily at MOI $m/z + 2\sigma_G$ where $\sigma_G$ is the standard deviation of the Gaussian-weighting envelop. The latter is a function of the mass resolving power at MOI *m/z*. Intensity artifact noise is sampled from $f_{int\text{-}}$ $_{artifacts}(k)$ and is added to $n = 10$ randomly selected pixels at *m/z* MOI. **c** MPMs (middle row) but not ion images (upper row) of a sphingomyelin SM(d34:2)[M+H]$^+$ (*m/z* 701.5592; FDR ≤ 0.1) are robust against various forms of artificially added noise and signal artifacts: random Gaussian noise (second column), presence of abnormally high-intensity peak artifacts (third column), and added overlapping peaks $2\sigma_G$ away from MOI *m/z* (fourth column). Despite the degraded visual quality of artificially "contaminated" data, MPMs are able to identify areas of significant metabolite spatial relative abundance. This is demonstrated by the high degree of overlap (yellow) for all noise types between estimated MPM hotspot contours of raw (green) and artificially "contaminated" data (red), as judged by their Dice similarity coefficient (DSC).

distribution of MOI intensities in the reference tissue. In such scenarios, however, it is important to ensure that the signal intensities of both test and reference tissues are comparable by observing appropriate experimental design, which deliberately minimizes technical variation (e.g., placing them on the same slide to be measured in a single measurement)[6] and/or by relying on robust intensity normalization methods[10,32]. To further validate this cross-tissue variant of the MPM method, we simulated three cases of test and reference SPPs (Supplementary Fig. 19). In case (1) it is expected that both the spatial and test-vs-reference intensity distributions null hypotheses are rejected, since a simulated spatial structure of high MOI abundance is present and intensity values are sampled from different normal distributions (Supplementary Fig. 19a). In case (2) the spatial null hypothesis is accepted, but the the intensity distribution null hypothesis is rejected, because no simulated high-MOI area is present but intensity values are again sampled from different normal distributions (Supplementary Fig. 19b). In case (3), the spatial null hypothesis is rejected while the intensity distributions null hypothesis is accepted, since a simulated high-MOI area is present, but here intensity values are sampled from the same normal distribution (Supplementary Fig. 19c). The generated CT-MPMs on simulated data correctly identified only the first case to include simulated significant cross-tissue relative spatial abundance (Supplementary Fig. 19a). We also used CT-MPMs to spatially localize areas of significant cross-tissue relative spatial abundance of the kinase inhibitor imatinib in a gastrointestinal stromal tumor (GIST) tissue sample when compared against a series of imatinib dilution spots in MALDI-TOF-MSI data from a previous study[31] (Supplementary Fig. 20). There, the reported mean imatinib content in that sample was 7.78 pmol (95% CI 7.28, 8.46 pmol) and 7.81 pmol (95% CI 7.63, 7.99 pmol) based on MALDI-TOF-MSI and UPLC-ESI-QTOF-MS

quantification, respectively. Consecutive comparison of the imatinib-tissue content against four imatinib dilution spots (3.13, 6.25, 12.5, and 25 pmol) showed a gradual decrease in the number of pixel intensities (zero at 25 pmol) that were detected as significant cross-tissue relative spatial abundance of imatinib (MPM hotspot contours in Supplementary Fig. 20e–h). The cross-tissue test carried out against the imatinib dilution spot of 6.25 pmol, i.e., the closest to the reported mean imatinib-tissue content, revealed that the tissue areas with significant cross-tissue relative spatial abundance of imatinib (MPM hotspot contours in Supplementary Fig. 20f) were spatially restricted and coincided with the high-intensity pixels in the imatinib intensity image of Supplementary Fig. 20d.

**Metabolite probabilistic maps are robust against spiked-in noise**

We had already considered noise at the level of data-dependent Gaussian-weighting and SPP transformation. To further validate the concept of molecular probabilistic mapping, we sought to systematically investigate the robustness of MPMs against various types of artificially added noise.

As exemplified for the sphingomyelin SM(d34:2)[M+H]$^+$ (*m/z* 701.5592; FDR ≤ 0.1), MPMs were rather robust against different types of computationally added noise. This was evidenced by DSC values of 0.85, 0.97, and 0.98 for comparisons of MPMs based on raw data versus data with artificially added Gaussian noise, intensity artifacts (i.e., isolated very high-intensity pixels) or interference peaks (i.e., peaks placed in the m/z proximity of the MOI), respectively (Fig. 4). Applying the same testing procedure to 142 MPMs of MOIs (positive ion mode; all METASPACE-verified at FDR ≤ 0.2) revealed median DSC values of 0.91, 0.98, and 0.98 for these three types of added noise, respectively (Supplementary Fig. 21), suggesting substantial

robustness for a wide range of MOIs. We also performed rigorous noise testing by varying the standard deviation of the sampled noise between a rather low noise dispersion (resembling Poisson noise) and up to 10 times the standard deviation of the raw MOI signal. We found that MPMs were robust against this type of noise up to four times the standard deviation of the raw MOI signal: MOI hotspot areas retained overlaps above 0.75 DSC, even though visual image degradation was observed (Supplementary Fig. 22). A similar computational experiment with sampled noise spiked in the vicinity of MOI (Supplementary Fig. 23) confirmed that Gaussian mass-window weighting contributes to MPMs robustness against interfering proximal signals. MOI hotspot identification was also found to be resilient to substantial numbers of spiked single-pixel high-intensity artifacts randomly placed within the tissue image (Supplementary Fig. 24). Single-pixel high-intensity artifacts may result from tissue tears, inhomogeneous matrix crystal distribution, ion source contamination, or other abrupt chemical inhomogeneities. They are typically rare, but for up to 450 such spiked signals (~2% of all pixels), DSC remained above 0.75 (Supplementary Fig. 24).

Since each MOI-specific CSR model is unique for every MPM evaluation, we also tested MPM stability across many runs by repeating the same evaluation 100 times, each time with a different CSR permutation. We found that estimated MPM hotspot and coldspot areas relative to the total tissue area were stable across all iterations, with mean overlap DSC values of 0.988 and 0.991 between the hotspot and coldspot areas, respectively, for each of the 100 iterations relative to that of the first iteration (Supplementary Fig. 25). We also considered the impact on MPM hotspot contours of additional areas of high-intensity points that were cumulatively or iteratively spiked into a simulated SPPs (Supplementary Fig. 26). However, we observed that the estimated MPM hotspot contours were largely unaffected (Supplementary Fig. 26). To test how MPMs compare in intersample and intermeasurement scenarios, we evaluated MPMs of four different lipid MOIs (FDR ≤ 0.1) in two serial sections of a human GB tissue (Supplementary Fig. 27) and in six serial mouse brain sections measured separately with MALDI-FTICR-MSI (Supplementary Fig. 28). We also tested the effects of common intensity normalization techniques, i.e., total-ion-count (TIC) and root-mean-squared (RMS) normalization, on the observed outcomes (Supplementary Figs. 29 and 30). In all cases, we observed good agreement of MPM hotspot/coldspot contours across serial sections and intensity normalization methods for several example lipid MOIs.

## Collective projections of molecular ensembles onto a single tissue plane

Perhaps even more far-reaching than single-molecule MPMs, data-integrating probabilistic maps of larger metabolite (or other biomolecules) sets or ensembles, typically assembled based on MSI scientists' research interests, may pave the way for visualization, exploration, and advanced analysis of integrated MSI data. We refer to them as collective-projection probabilistic maps (CPPMs; Supplementary Fig. 2). Biomedically relevant examples of metabolites include entire lipid classes in SwissLipids (https://www.swisslipids.org)[33], amyloid peptides[29], nucleotides[34] and other low-mass hydrophilic metabolites, potassium or sodium adducts of lipids across lipid classes[35] or any other scientist-defined set of metabolites.

To generate CPPMs, MSI data for every metabolite in a molecular ensemble is transformed to its respective SPP representation, and then all of these SPPs are collectively projected into a single SPP in a single image space (Fig. 5a). Finally, this collective SPP is subjected to spatial probabilistic mapping into CPPMs using the MPM workflow (Fig. 5b–d, Supplementary Fig. 34). Importantly, this computational framework permits spatial evaluation of composite numeric scores obtained by applying basic arithmetic operations on spatial point patterns of multiple MOIs in different ways than before. For example, the relative

spatial abundance or deficiency of the adenine nucleotides $[ATP-H]^-$, $[ADP-H]^-$ and $[AMP-H]^-$, individually relative to their collective sum, or more complex scores such as the adenylate energy charge[34,36] and the adenylate kinase mass action ratio[37] can be probabilistically mapped within the limits of error propagation (Fig. 5b, Supplementary Fig. 34b). Our data suggest that the latter two scores, indicative of areas of high energy metabolism, overlap with tissue regions annotated as viable tumor (VT), suggesting that CPPMs of molecular ensembles as innovative use of MSI data may provide insights into spatially resolved pathophysiology that would not be possible by single-molecule ion images or MPMs not involving collective projections. This and the following observations will obviously require extensive follow-on studies with larger sample cohorts before clinically valid statements can be made. It should be noted, however, that Gaussian-weighting and MPM-internal KDE smoothing are both expected to reduce the impact of signal uncertainty on the rendered intensities and hotspot/coldspot contours compared to ion images. Nevertheless, each ion intensity image will still contain an unknown amount of nonbiological technical variability[6], which could be carried on to the composite image representation. In particular, the division of variables may be prone to uncertainty amplification.

Another set of examples that illustrates the type of analyses that CPPMs enable, examined all glycerophospholipids (GPLs) and lyso-glycerophospholipids (lyso-GPLs), referred to here as (lyso) GPLs, together in the two GB tissue sections (technical replicates) in detail. A recent report suggested that monounsaturated GPLs were enriched and polyunsaturated GPLs were depleted in tumor microenvironments of various types of cancer[38]. In the samples analyzed here, ion images seemed to support this notion. However, CPPMs suggested that higher relative spatial abundance/deficiency of both classes of lipids was not significant, whereas di-unsaturated GPLs displayed higher relative spatial abundance—but in areas just outside of those annotated as viable tumors (Supplementary Fig. 31).

Collective projections of metabolite ensembles support initial surveys of entire molecular pathways. For instance, >150 lipids involved in GPL biosynthesis and remodeling can be interpreted in a single pathway overview (Supplementary Figs. 32 and 33). Interestingly, CPPMs of PC, PA, and PS but less so of PI and their corresponding lyso-GPL cleavage products suggested alterations in the Lands' cycle of phospholipid remodeling in GB, i.e., enrichment of GPLs and concomitant depletion of lyso-GPLs in viable tumor (Fig. 5c). Retrospective transcript expression profiling of Lands cycle enzymes revealed overexpression of various acyltransferase genes (*LPCAT1*, *AGPAT1*, *LPCAT3*, *MBOAT7*) in GB compared to normal brain tissue but less changes in phospholipase A2 (*PLA2G6*) expression that underline the CPPM-based assessment. Taken together, spatial probabilistic mapping of molecular ensembles supports the global interrogation of metabolic pathways, hence opening up new avenues for the comprehensive analysis of metabolite classes.

As a final example, and analytical considerations such as region-specific ion suppression notwithstanding, we reasoned that CPPMs of all potassium or sodium adducts of lipids in the SwissLipids database could serve as indicators of the ion milieu in a cancer tissue sample. Analogously, it has been noted that sodium MRI can serve as an indicator of vital tumor in vivo.[39] $Na^+/K^+$-ATPase maintains high overall potassium and low tissue sodium concentrations in viable cells, and higher cellularity corresponds to a lower tissue sodium concentration[39]. In contrast, highly abundant $Na^+$-adducts colocalized with necrotic tissue in xenografts of five different tumor cell lines[35]. Similarly, the projected molecular ensemble (=CPPM) of potassium adducts of all (lyso-)GPLs was more abundant in vital tumor and surrounding areas, whereas the CPPMs of projected sodium adducts were more pronounced in necrotic tissue showing significant relative spatial deficiency (coldspot) in vital tumor (Fig. 5d, Supplementary Fig. 34).

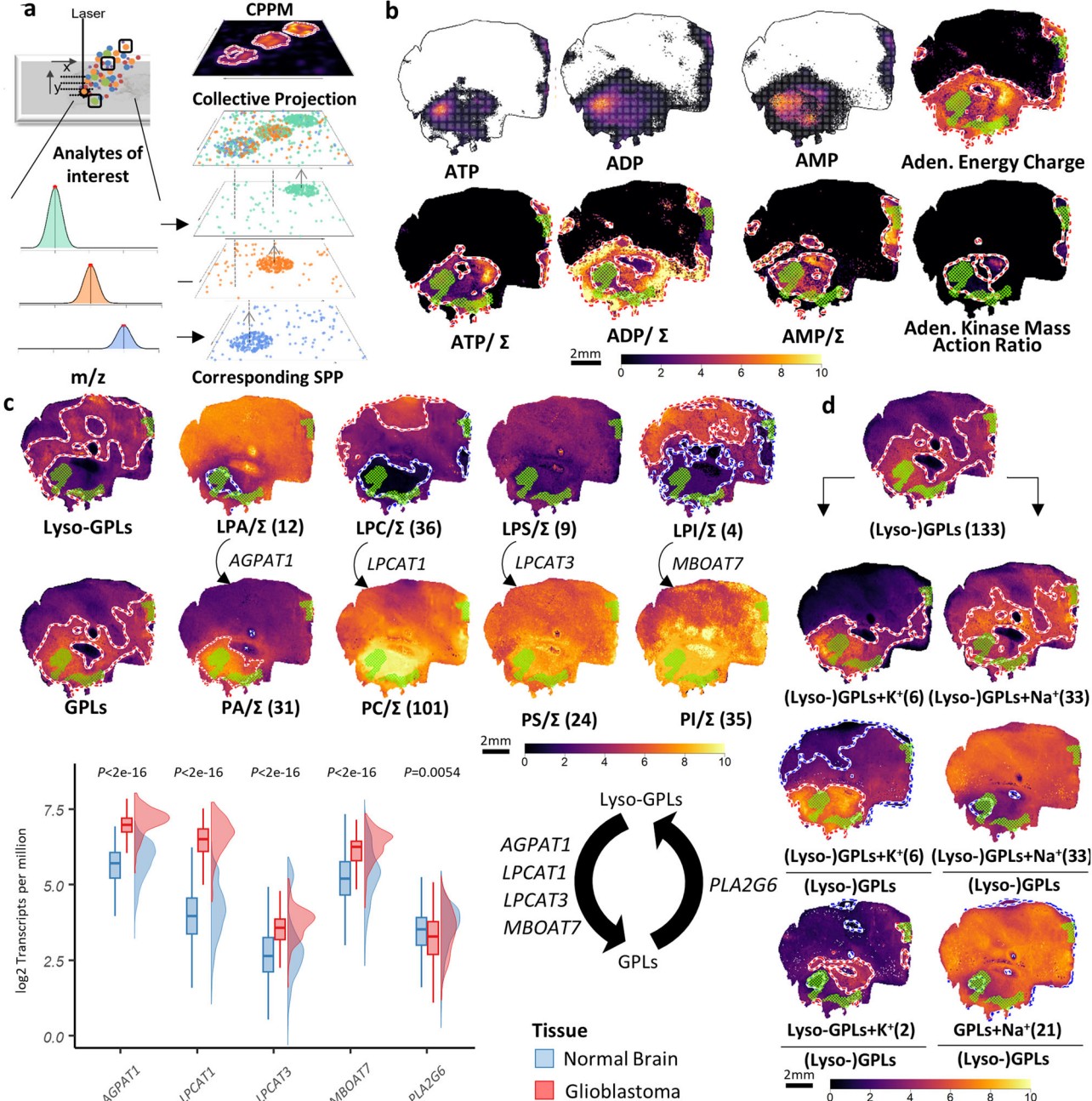

**Fig. 5 | Collective-projection probabilistic maps (CPPMs) of metabolite ensembles for visualization and interpretation of scores for energy metabolism, of (lyso-) glycerophospholipid (GPL) remodeling pathways, or of ion milieu. a** Computation of CPPMs: Spatial point patterns (SPPs) of user-curated ensemble of metabolites-of-interest (MOIs) is collectively projected onto the same tissue plane and a molecular probabilistic map (MPM) is computed. **b** CPPMs enable basic arithmetic manipulations on SPPs of multiple MOIs such as the nucleotides [ATP-H]⁻, [ADP-H]⁻ and [AMP-H]⁻ (FDR ≤ 0.2; upper row), e.g., normalization against their sum (bottom row; Σ = [ATP-H]⁻+[ADP-H]⁻+[AMP-H]⁻). CPPMs also enable complex spatial quantitative scores such as adenylate energy charge (([ATP-H]⁻+0.5[ADP-H]⁻) / ([ATP-H]⁻+[ADP-H]⁻+[AMP-H]⁻); top right) and adenylate kinase mass action ratio ([ATP-H]⁻ [AMP-H]⁻ / [ADP-H]²⁻; bottom right). Green mesh indicates co-registered vital tumor regions. **c** CPPMs enable spatial investigation of GPL remodeling (Lands' cycle) in glioblastoma by collectively visualizing lipid classes. Upper panel: CPPMs of all lyso-GPLs and single classes (LPC, LPE, LPS, LPI; top row) compared to all GPLs and GPL classes (PC, PE, PS, PI;

bottom row). Lyso- and non-lyso-GPL pairs are normalized to their sum (e.g., for LPC and PC, Σ = LPCs plus PCs). Lower panel: Rainfall plot of expression levels of select Lands' cycle enzymes in normal brain (blue; GTEx data) and glioblastoma (red; TCGA data) represented as log2 transcripts per million (two-sided Wilcoxon rank-sum test; $P = 0.0054$ for *PLA2G6* and $<2.22 \times 10^{-16}$ for the rest; $n = 1671$ and $n = 156$ for every normal brain and Glioblastoma boxplot, respectively). Boxplots indicate median (middle line), 25th, 75th percentile (box) and whiskers which extend to the most extreme data point, which is no more than 1.5 times the length of the box away from the box. Numbers in parenthesis = METASPACE-verified lipids at FDR ≤ 0.2. **d** Analysis of the tissue's alkali ion milieu. Top: CPPMs for all (lyso-)GPLs, i.e., lyso-GPLs plus GPLs (FDR ≤ 0.5). Left column from top: (1) CPPMs of (Lyso-)GPLs potassium adducts, (2) of (Lyso-) GPLs potassium adducts relative to the sum of all (lyso-)GPL adducts, and (3) CPPMs of only lyso-GPLs potassium adducts relative to the sum of all (Lyso-) GPL adducts. Right column: As left column but showing sodium adducts. Source data are provided as a Source Data file.

In conclusion, with this study, we make the *moleculaR* framework available for the scientific community as an R package complementing leading MSI-bioinformatics packages[40–42]. *moleculaR* is capable of importing metabolite annotation results from the METASPACE platform to compute FDR-verified MPMs and CPPMs. *moleculaR* is equally applicable for ultrahigh-resolution MSI like MRMS, or for high-resolution instruments like MALDI-timsTOF. It could also be deployed and hosted on a centralized server and is equipped with a web-based graphical user interface (GUI).

# Methods

## Ethics statement

The research, which this study is part of, was conducted in concordance with the declaration of Helsinki and was approved by the Ethics Committee at Heidelberg University, Germany (applications S-130/2022 and AFmu-207/2017). Participants were recruited through the Heidelberg University Hospital and gave informed consent prior to study inclusion.

## Materials

All reagents were of HPLC grade. Milli-Q water (ddH$_2$O; Millipore) was prepared in-house. Conductive indium tin oxide (ITO)-coated glass slides were purchased from Bruker Daltonics (Bremen, Germany). Adhesive slides SuperFrost Plus™ were obtained from Thermo Fisher Scientific (Waltham, Massachusetts, USA) for histological analysis. Trifluoroacetic acid (TFA) and 1,5-diaminonaphtalene (1,5-DAN, ≥97.0%) MALDI matrix were purchased from Sigma-Aldrich (St. Louis, MO). 4-phenyl-α-cyanocinnamic acid amide (PhCCAA) matrix was obtained from SiChem (Bremen, Germany). Acetonitrile (ACN) was obtained from VWR (Darmstadt, Germany). For external calibration of the Bruker solarix magnetic resonance mass spectrometer (MRMS), a mixture of poly-DL-alanine (10 mg/mL), L- alanine ≥ 99.5% (5 mg/mL), and taurine ≥ 99% (5 mg/mL; all from Sigma-Aldrich) was used in water.

## MALDI MSI of human glioblastoma specimens

All patients have been treated at the Heidelberg University Hospital. Patients gave informed consent prior to inclusion in exploratory molecular analysis, including, but not limited to, MALDI MSI. The research is conducted in concordance with the declaration of Helsinki and was approved by the Ethics Committee at Heidelberg University, Germany (applications S-130/2022 and AFmu-207/2017). Tissue samples were taken through the primary operation of the brain tumor. Frozen resected tumor material was retrieved from the Department of Neuropathology in Heidelberg and reviewed by a board-certified neuropathologist. Diagnoses were molecularly confirmed according to the recent WHO classification and methylation profiles were confirmed with methylation EPIC array (#WG-317-1003, Illumina, San Diego, California, USA). Hematoxylin and eosin (H&E)-stained tissues were scanned using an Aperio CS2 scanner (Leica Biosystems, Nussloch, Germany) and annotated by an expert neuropathologist. Frozen GB tissue was cryosectioned (10 µm; Leica CM1950; Leica Biosystems). Sections were mounted onto ITO slides (Bruker Daltonics), and adjacent sections were placed on SuperFrost slides (Thermo Fisher Scientific) for H&E staining. Cryosections were dried for 15 min in a desiccator and stored at −80 °C. Tissue sections on ITO slides were coated with 10 mg/mL 1,5-DAN matrix in 50% ACN/water using an M3 TM-Sprayer (HTX Technologies, LLC, North Carolina, USA): Temperature: 75 °C; No. of Passes: 17; Flow Rate: 0.1 mL/min; Velocity: 1200 mm/min; Track spacing: 3 mm; Pattern: CC; Pressure: 10 psi; Gas Flow Rate: 2 L/min; Nozzle Hight: 40 mm; Drying Time: 0 s. MALDI-High-mass-resolution-imaging was performed on a solariX 7T XR (Bruker Daltonics) FTICR MRMS using *Compass ftmsControl* (Version 2.2) and *flexImaging* (Version 5.0) software (both Bruker Daltonics). All measurements were performed at 50-µm lateral resolution in the mass range between *m/z* 100 and 1200 in negative mode followed by measurement in positive mode, on the same spots. Spectra were recorded with a 1M data point transient, a mass resolving power of 85k at *m/z* 400, 98k at *m/z* 314, 123k at *m/z* 249, and a FID of 0.4893 s. One scan from 100 laser shots with a frequency of 1000 Hz was used per pixel. Q1 mass was set to *m/z* 120, while Time-of-Flight was adjusted to 0.9 ms. In both modes, a Plate Offset of 100 V was used in combination with a deflector plate voltage of 200 V. External mass calibration was performed using poly-alanine with the addition of taurine (*m/z* 125.014664) to cover the whole mass range[43]. For internal lock mass calibration in negative and positive modes, the [M−H]⁻ signal of phosphatidylinositol (38:4) (*m/z* 885.54875) and the [M+H]⁺ signal of phosphatidylcholine (34:1) (*m/z* 760.58508) were used, respectively. To minimize data load, data was saved as Profile Spectrum with a Data Reduction Factor of 97% in addition to the centroided mass spectra (SQLite peaks list), which are generated during data acquisition (method: Apex; SNR ≥ 3). The reduced profile data was used to sample a single spectrum for the estimation of FWHM as a function of *m/z*, while centroided data was preprocessed and ultimately served as an input for *moleculaR* as described below.

## MALDI MSI of mouse brain tissue

Deep-frozen mouse brains from 12-week-old female wild-type C57BL/6N mice were obtained from the German Cancer Research Center (DKFZ). Organs were from excess mice, which did not participate in studies and had to be euthanized. Sex-based analysis has not been performed since the study did not contain research/comparison of cohorts of animals. Tissues were sliced and mounted as described above. After drying, sections were spray-coated with 2.5 mg/mL PhCCAA matrix[44] in 70% ACN/H$_2$O using a TM-sprayer (HTX Technologies). MALDI-MSI was performed as described above. Measurements were performed at 20-µm lateral resolution in the mass range between *m/z* 100 and 3000 in negative ion mode. Q1 mass was set to *m/z* 600, while Time-of-Flight was adjusted to 1 ms. For internal lock mass calibration, the [M−H]⁻ signal of phosphatidylinositol (38:4) (*m/z* 885.5498553) was used. All other parameters were similar to those described in the preceding section. Orthogonal MALDI timsTOF MSI data acquisition was performed on a timsTOF fleX (Bruker Daltonics) in negative ion mode in the range of *m/z* 600–1800. Spectra were recorded using 500 shots per pixel with a laser repetition rate of 10 kHz using a 20-µm step size. Transfer parameters were as follows: Funnel 1 RF 150 Vpp; Funnel 2 RF 200 Vpp; Multipole RF 200 Vpp. Quadrupole parameters: Ion energy 5.0 eV; Low Mass *m/z* 900. Focus PreToF Parameters: Transfer Time 100 µs; PrePulse Storage 15 µs. External calibration was performed in the electrospray mode using ESI-Low Concentration Tuning Mix (Agilent Technologies, Santa Clara, USA). MALDI-TOF-MSI measurements were performed on a Rapiflex MALDI-TOF MS (Bruker Daltonics) in negative reflector mode with *m/z* 600–1800 using *FlexImaging* 5.0 software (Bruker Daltonics). The acquisition method was calibrated using quadratic calibration; 400 laser shots at 10-kHz repetition rate were accumulated for each raster spot with a lateral resolution of 20 µm. Ion Source 1 was set to 20 kV, PIE to 2.47 kV, and the ion lens to 1.75 kV, and the delayed extraction time was set to 90 ns. The digitizer was set to 0.8 GS/s, and the deflector cutoff mass for matrix suppression was set to *m/z* 590. Some MALDI-MSI data used in previously published studies were reused: (1) TOF-MSI of the APP NL-G-F Alzheimer's disease mouse model[29] (Fig. 3d and Supplementary Fig. 12). (2) FTICR-MSI of IDH-mut and IDH-wt glioma human tissue[17] (Fig. 3e and Supplementary Fig. 18). (3) FTICR-MSI of imatinib drug dilution series on porcine liver[31] (Supplementary Fig. 16). (4) TOF-MSI of gastrointestinal stromal tumor (GIST) tissue[31] (Supplementary Fig. 20).

## Semi-automatic multimodal image registration

To transform H&E annotations to the MSI image domain, the optical images (5-µm lateral resolution) acquired prior to MALDI-MSI

acquisitions (and intrinsically registered with the MSI image information) and the H&E images (0.5-μm lateral resolution) were used. Briefly, H&E and optical images were transformed into grayscale images using the luminosity method (weighted average of the red, green, and blue channels). Then, acquisition regions within the MSI data and annotations of the H&E files were used to define a minimal bounding box for each sample region. Subsequently, all sample regions were cropped out of the grayscale images. Cropped images of both modalities were resampled to a lateral resolution of 7.5 μm per pixel. Afterward, images were exported in the *nrrd* image file format. Image-based registration is used to transform the cropped images from the MSI to the H&E image domain by using *elastix*[45]. The full registration is composed of a rigid step, followed by a deformable step. Each registration results in a set of parameters describing the transformation from the H&E to the MSI image domain. Those parameters are used to transform point information accordingly. Transformed polygons and corresponding annotation labels were written to mis-files.

Rigid registration is based on a multiresolution registration strategy (Gaussian pyramid[46] with three levels and down-sampling factors of 4,2,1, each of which represents pattern information at a different scale allowing for a course-to-fine image registration paradigm). The Advanced Mattes Mutual Information in *elastix* was used as multimodal metric for the optimization of a rigid transformation using linear interpolation and 250 iterations. For the subsequent deformable registration steps, the same multiresolution scheme and metric were applied. For the deformable transformation, a recursive B-Spline transformation was used with interpolation using third-order B-Splines. The optimization was run for 750 iterations. For cases of failed image registration, a multimetric registration approach was used, and manually defined control points were added at corresponding locations in both modalities to support the registration process. In this case, the multimetric output was a composition of the Mattes Mutual Information metric and the Corresponding Points Euclidean Distance metric (equally weighted). The *M²aia* (RRID:SCR_019324; https://www.github.com/jtfcordes/m2aia)[47] desktop application was used to view registration results, to control registration parameters, and to interactively define pairwise corresponding control points within both image modalities.

## Data preprocessing

The centroided FTICR-MSI data (SQLite peaks list) was first imported into *SCiLS Lab* Software version 2016a (Bruker Daltonics) and then exported into imzML format. Further analysis proceeded in R, using the *MALDIquantForeign* R package for data import[42]. Positive and negative mode spectra were stored internally in sparse-matrix representation (*Matrix* package) for computation efficiency. Bulk data analysis and preprocessing were carried out via *MALDIquant*[42]. One pixel representing a continuous (profile) mass spectrum was randomly chosen from the corresponding profile data and exported as a CSV file via *flexImaging* software version 5.0 (Bruker Daltonics). FWHM values were computed per peak and plotted against the *m/z* axis (see below). A locally estimated scatterplot smoothing (LOESS) was used for fitting a smoothing curve to describe FWHM as a continuous function of the *m/z* axis (see below), which was then used to estimate FWHM at any given *m/z*. For the centroided MSI data, peaks that occurred in less than 1% of the pixels were filtered out to limit the presence of spurious random peaks. Peak binning was performed via the peak binning routine of *MALDIquant*; the observed peak masses of the entire MSI data (all peaks of all pixels) were grouped and sorted into a single vector and the difference between each neighboring pair was computed. Then a series of iterative bisecting was applied on the mass vector, each time at the largest difference until all peaks within each bin fulfilled the criterion $|peak_{ij} - \mu_j|/\mu_j$ <tolerance; where $peak_{ij}$ the mass of the *i*-th peak at the *j*-th bin, $\mu_j$ is the mean mass of all peaks present in the *j*-th bin (bin center; the new peak position) and tolerance is the maximal relative peak deviation ($\triangle m/m$; see next section) of

peak positions to be considered as identical which, for this study, was set to 12 ppm ( $= \triangle m/m; \triangle m =$ FWHM at *m/z* 400 ≈ 0.0048 Da; *m* = *m/z* 400). Since the focus of this study was on the lipidome, *m/z* 400 was chosen. If the study focus was on smaller metabolites, then *m* should be chosen accordingly. Processed centroided datasets (negative and positive ion modes) were exported into processed (centroided) imzML files via *MALDIquantForeign*. Data-adaptive pixel-wise recalibration based on endogenous biological signals was conducted using the MSI-recalibration tool[48]. The centroided imzML data was uploaded into the METASPACE annotation platform[13] (https://metaspace2020.eu) and lipid search was performed against the SwissLipid database[33]. The corresponding annotations were then downloaded as csv files and used as metabolites-of-interest (MOIs) for the molecular probabilistic map (MPM) and collective-projection map (CPPM) workflows; in other words, only METASPACE-verified MOIs were considered for subsequent analysis.

## Full-width-at-half-maximum (FWHM) model fitting

The physical basis of mass resolving power is different for each type of mass spectrometer. For an FTICR MS, the mass resolving power can be expressed as[49]

$$\frac{m}{\triangle m} = -\frac{\omega_c}{\triangle \omega_c} = -\frac{qB}{m\triangle \omega_c}, \tag{1}$$

where $\triangle m$ and $\omega_c$ define the mass resolution at FWHM and the cyclotron frequency of an ion inside the trap. $q$ and $m$ are the charge state and mass of the ion, respectively, while $B$ is the magnetic field strength at the center of the trap of the mass spectrometer. If the ICR signal is undamped within the free-induction-decay time $T_{FID}$ (acquisition time of the time-domain signal), the mass resolution is given by[49]

$$\triangle m = \frac{7.589m^2}{qB T_{FID}} \tag{2}$$

This state is referred to as the low-pressure limit since ion-neutral collisions inside the trap are neglected. However, if ion-neutral as well as ion–ion collisions cannot be neglected, the mass resolution can be described by[49]

$$\triangle m = \frac{2\sqrt{3}m^2}{qB\tau} \tag{3}$$

where, $\tau$ is the damping constant of the radial ion motion. Equations (2) and (3) show that the mass resolution scales with $m^2$ and inversely with $q$.

For this study, FWHM as a function of mass for a particular MOI is estimated by a LOESS, based on the FWHM values of single peaks (SNR ≥ 3) extracted from profile mass spectra. In addition, a direct comparison to the theoretical expectations according to Eqs. (2) and (3) is shown in Supplementary Fig. 4a for $z = 1$ and $q = z \times e$ where $e$ is the elementary charge.

The resolving power for a TOF mass spectrometer is given by[50]

$$\frac{m}{\triangle m} = \frac{t}{2\triangle t} \tag{4}$$

where $t$ denotes the time-of-flight of an ion with mass $m$, and $\triangle t$ is the corresponding peak width. Since TOF does not only depend on fundamental parameters like *m/z* but is also affected by multiple other aspects, a theoretical prediction of the FWHM as a function of mass is difficult. Nevertheless, the same empirical model fitting described above as for MALDI-FTICR data could still be used to model FWHM data for TOF and timsTOF devices.

*moleculaR* relies on centroided MSI data stored in imzML format, which typically does not provide FWHM information of the peaks list. Hence, *moleculaR* relies on an externally provided randomly chosen full-profile spectrum of the MSI data under study. It is assumed (and observed; Supplementary Fig. 3a, b) that FWHM as a function of *m/z* should be similar across the tissue.

## Gaussian mass-window weighting

The rationale behind using Gaussian mass-window weighting in place of traditional uniform mass-window weighting is to incorporate FWHM versus *m/z* relationships (especially nonlinear relationships) into the calculation of MOI intensities. This is done based on the relation between FWHM and $\sigma_G$ of the weighting Gaussian envelop (FWHM $= 2\sqrt{2\ln2}\,\sigma_G$). This enables data-driven calculation of the mass-window width taken as $m_{MOI} \pm 3\sigma_G$ (i.e., the span of the erected Gaussian window; $m_{MOI}$ is the *m/z* value at MOI) independent of the user, measurement device, and measurement parameters.

For any MOI, considering ion mode and adducts, the theoretical monoisotopic $m/z$, $m_{MOI}$, is computed or taken from a curated database. For this $m/z$, the expected data-dependent FWHM and $\sigma_G$ of a Gaussian envelope centered at $m_{MOI}$ are determined. This Gaussian envelope, scaled to [0,1] intensity, is used as a weighting function for any observed POIs occurring within its effective support ($m_{MOI} \pm 3\sigma_G$), i.e., computing $\sum_{j=1}^{p} w_j i_j$ where $p$ is the number of peaks observed within $m_{MOI} \pm 3\sigma_G$, and $w_j$ is the corresponding Gaussian weight at the *j*-th peak with intensity $i_j$. This serves as a protection against possible proximal background signals by down-weighting them relative to $m_{MOI}$; it does not protect against miscalibrated/misaligned data. Here, we utilized the data-adaptive MSI-recalibration method[48] in such cases.

## Spatial point pattern (SPP) data representation

The difference between SPP and raster image representations resembles the difference between vector-based and raster-based graphics. Both can be used to represent the same spatial distribution, but each provides a unique set of tools and methods that are tailored to the specific nature of each representation. While MOI's SPPs inherit the gridded spatial locations from MSI data, creating a CSR model on a spatial grid would directly violate the randomness criterion of such models. Therefore, in order to facilitate the direct and homogeneous comparison between MOI and a corresponding CSR model, MOI spatial intensities must be converted into an SPP representation. Once MOI's intensities are extracted via the Gaussian mass-window weighting as described in the previous section, the *Spatstat* framework[51] is then used to construct a marked (i.e., intensity-weighted) SPP representation $SPP_{MOI}$ of MSI signals distributed in a spatial 2D contour $\Phi_{tissue}$ representing the tissue section with a spatial point density $\Lambda$, which equals the number of points per unit area, i.e., the average spatial density of all points *n* within $\Phi_{tissue}$ or $n/A_{tissue}$ where $A_{tissue}$ is the total area of $\Phi_{tissue}$.

## Molecular probabilistic map (MPM)

The SPP representation of MOI, $SPP_{MOI}$, within a given tissue contour $\Phi_{tissue}$ enables computation of the corresponding $MPM_{MOI}$. First, a random point pattern $CSR_{MOI}$ is created according to a complete spatial randomness (CSR) model and is used to represent a sample of random events to be considered as an intrinsic control for every MOI case. This CSR process is generated spatially as a uniform Poisson process with a fixed spatial point density of $\Lambda$. Unlike in common CSR generating models[20,52], in the case of MSI, $CSR_{MOI}$ must also carry intensity weights (representing pixel-wise signal intensities) in order to be a valid intrinsic control model for $SPP_{MOI}$. To achieve this, a uniform Poisson spatial point process is created within $\Phi_{tissue}$ with the same spatial point density $\Lambda$, then point intensities of $SPP_{MOI}$ are randomly permuted and assigned to the points just created. These two steps have the effect of a spatial reshuffling of $SPP_{MOI}$ points, until they

assume a spatial uniform Poisson process, thus effectively dissolving any spatial clustering or autocorrelation of signals (Fig. 2). To capture the overall spatial trend of the MOIs' intensities, kernel density estimation (KDE) is applied with an isotropic Gaussian kernel (i.e., no specific spatial "direction" is assumed of the MOIs under study) for both $SPP_{MOI}$ and its corresponding $CSR_{MOI}$ (see "KDE Bandwidth Estimation" section) and is sum-normalized to compute the weighted spatial density functions $\rho_{MOI}(x,y)$ and $\rho_{CSR}(x,y)$, respectively. Let $f_{MOI}(k)$ and $f_{CSR}(k)$ denote the probability density functions of intensities *k* obtained from the resulting $\rho_{MOI}(x,y)$ and $\rho_{CSR}(x,y)$, respectively. As a consequence of the central limit theorem, and as a convenient byproduct of applying KDE on $CSR_{MOI}$, the intensity distribution $f_{CSR}(k)$ converges toward a normal distribution as the bandwidth increases, which in practice can already be observed for low bandwidth values. This does not necessarily apply to $f_{MOI}(k)$ (see Supplementary Fig. 12a, b, respectively). Hence

$$f_{CSR}(k) \cong \frac{1}{\sigma_{CSR}\sqrt{2\pi}} e^{-\frac{1}{2}\left(\frac{k-\mu_{CSR}}{\sigma_{CSR}}\right)^2}, \tag{5}$$

where $\mu_{CSR}$ and $\sigma_{CSR}$ are the mean and standard deviation of $\rho_{CSR}(x,y)$. To identify areas with higher likelihood of showing a significant relative spatial abundance of MOI when compared to a random distribution (i.e., MOI's MPM hotspot; i.e., nonrandom spatial accumulations of MOI intensities) and, on the other hand, areas which have a higher likelihood of showing a significant relative spatial deficiency of MOI (i.e., MOI's MPM coldspot; i.e., nonrandom spatial depletions of MOI intensities), the lower and upper-tail *P*-value is computed for every pixel intensity in $\rho_{MOI}(x,y)$ against the null distribution $f_{CSR}(k)$ resulting in two spatial maps of lower and upper-tail *P*-values $P_{lwr}(x,y)$ and $P_{upr}(x,y)$, respectively. Next, to account for the inherent multiple testing problem, Benjamini–Hochberg *P*-value correction is applied resulting in $P^*_{lwr}(x,y)$ and $P^*_{upr}(x,y)$ (Supplementary Fig. 9). Then null-hypothesis significance testing is carried out by comparing each corrected *P*-value in $P^*_{lwr}(x,y)$ and $P^*_{upr}(x,y)$ against a significance level of $\alpha = 0.05$. Locations that reject the null hypothesis are declared to belong to either an MPM hotspot $(x_{hs}, y_{hs})$ or coldspot $(x_{cs}, y_{cs})$ if

$$(x_{hs}, y_{hs}) \in \left\{ x, y : P^*_{upr}(x_{hs}, y_{hs}) \le \alpha \right\}$$
$$(x_{cs}, y_{cs}) \in \left\{ x, y : P^*_{lwr}(x_{cs}, y_{cs}) \le \alpha \right\} \tag{6}$$

The MOI's molecular probabilistic map, $MPM_{MOI}$ is then defined as a composite representation of MOI spatial density of Gaussian-weighted intensities according to the scheme shown in Fig. 1, with MOI's MPM hotspots and/or coldspots superimposed as polygonal contours identifying areas of MOI significant relative spatial abundance and deficiency, respectively. It is important to note that the above procedure does not affect the visual aspect of the MOI's intensity distribution; the previously mentioned KDE smoothing occurs internally at the level of MPM computation, while the original (Gaussian-weighted) SPP intensities are carried on to the resulting MPM, and individual signal intensities (including sparse or single-pixel signals) are not altered or removed.

## Cross-tissue molecular probabilistic map (CT-MPM)

Ion intensity distributions of metabolites or drugs are compared between test and reference tissues, e.g., those dosed with a drug or carrying certain mutations versus controls, in two steps: First, areas of significant relative spatial abundance/deficiency are computed in the test tissue (testing against the spatial null hypothesis; MPM method described above) as shown in Fig. 2a. Then the signal intensities of the reference tissue are used to infer a nonparametric (distribution-free) empirical cumulative distribution function (eCDF) which acts as an estimator of the underlying cumulative distribution function. All MOI

intensities of the test tissue are then tested against it (i.e., the inferred eCDF) in order to find the likelihood of them (i.e., signal intensities of the test tissue) being drawn from the signal distribution of the reference tissue. More precisely, the lower and upper-tail $P$-value is computed for every MOI intensity in $SPP_{MOI}$ of the test tissue against the inferred eCDF of the reference tissue, and Benjamini−Hochberg correction is applied to account for the inherent multiple testing problem. Similar to the "within-tissue" MPM method described above, the $P$-value threshold beyond which the null hypothesis is rejected is set to $\alpha = 0.05$. Finally, test tissue intensities that reject both the spatial and test-vs-reference intensity distributions null hypotheses are designated as having significant cross-tissue relative spatial abundance/deficiency. In other words, pixel locations of these intensities could be described as areas of the test tissue which exhibit a statistically significant nonrandom spatial MOI abundance/deficiency pattern and contain intensities that are unlikely to belong to the distribution of MOI intensities of the reference tissue. It is important to note, however, that for such comparative cross-tissue analyses, an appropriate experimental design must be observed in order to minimize technical variability and ensure comparability. This could be achieved, for example, by placing tissues on the same slide to be measured in a single measurement[6] and/or by relying on robust intensity normalization methods[10,32].

## Kernel bandwidth estimation

KDE is a key step in the MPM workflow, both for a given MOI's spatial point pattern and the corresponding CSR model: (1) It captures the overall spatial trend of the MOIs' intensities; (2) it forces $f_{CSR}$ to converge to a normal distribution and; (3) and, being a low-pass filter, it has the often-desired outcome of smoothing technical variations and noise fluctuations during the process of hotspot/coldspot estimation which has, in turn, a positive outcome on the method's tolerance to pixel-to-pixel batch effects, and section-to-section/slide-to-slide batch effects[6]. However, smoothing SPPs with high KDE bandwidths while generating MPMs could potentially have an adverse effect of overlooking fine, small, or sparse spatial structures, deeming them insignificant in terms of their relative spatial abundance. To prevent spatial oversmoothing, we ensured that KDE bandwidth ($h_{KDE}$) estimation takes into account the MOI's spatial autocorrelation[19] estimated over a scale-space representation of the MOI intensity image. To achieve this, KDE was applied iteratively with $h_{KDE}$ varying from 1 to 10 (pixels; multiples of $50\,\mu m$ in this study) in 0.5 increments. During each iteration, the global Moran's I statistic, a measure of spatial autocorrelation, is determined (using the *raster* R package; first order Queen's case adjacency with unit weights). The optimal $h_{KDE}$ is then determined by finding the point of maximum curvature, i.e., the "knee" point, via the *Kneedle* method[26] in the Moran's I vs $h_{KDE}$ plot. This is the point, after which an increase in $h_{KDE}$ does not result in a considerable increase in the spatial autocorrelation of the smoothed density image. In other words, the KDE bandwidth at which the Moran's I statistic's rate of change abruptly falls is the scale, at which it is expected that random pixel fluctuations are smoothed away and important spatial structures/features/patterns start dominating the spatial landscape.

## Collective projections probabilistic map (CPPM)

Given a set of MOIs $C \in \{MOI_1, MOI_2, \ldots, MOI_m\}$, in this study queried from the SwissLipids database (https://www.swisslipids.org) and verified against the POI-MOI matching platform METASPACE (https://metaspace2020.eu), for each single $MOI_i$ an SPP representation $SPP_i$ is calculated. Afterward, all individual $SPP_i$ are projected into the same tissue plane $\Phi_{tissue}$ resulting in an SPP for the collective projection, $SPP_C$. Since SPP representations do not restrict the number or location of points in the point pattern, a single SPP can hold any number of points coming from any number of MOIs. Points sharing the same coordinate location (e.g., originating from the same $x,y$-coordinate

location on the MSI raster) can co-exist without the need to sum them up. Since POI-MOI matching usually reports a group of candidate molecules for a single POI (at a given FDR, metabolite database, and mass resolving power), mapping MOIs to POIs could result in duplicated representations of POIs within CPPMs. Here, *moleculaR* provides simple tools to filter out duplicated counts of the same $m/z$ value by incorporating only the intensities of unique masses present in the computed CPPM. Moreover, if two MOIs of the MOI set $C$ overlap due to insufficient resolving power of the mass spectrometer at $m/z$ MOI, the Gaussian mass-window weighting compensates for this, provided that the two MOIs are at least partly resolved (unlike the case in Supplementary Fig. 8).

The workflow then commences with intensity standardization (i.e., $z$-score normalization) applied to the intensities of each individual $SPP_i$ within $SPP_C$ by subtracting its (i.e., $SPP_i$) intensities mean and dividing by the standard deviation. This type of transformation aims to equalize the variance of measured MOI intensities by setting the mean intensity of each MOI equal to zero, thereby adjusting for differences in the offset between MOIs with high- and low-intensity ranges, while, at the same time, setting the standard deviation of intensities equal to one[53]. This is done to (at least partially) compensate for the inherent heteroscedasticity and possible differences in ionization efficiency between the individual $MOI_i$s. Then $CSR_C$ is created, and subsequently, KDE is applied to both $SPP_C$ and $CSR_C$, in order to compute $MPM_C$ (see above), i.e., the resulting collective-projection probabilistic map $CPPM_C$ is equivalent to $MPM_C$. The naming distinction is only made to emphasize that CPPMs are based on the visualization of multiple MOIs at a time.

## Spatial arithmetic expressions

For any number of MOIs, basic arithmetic operations on their SPPs could also be applied. This is useful when a ratio of two MOIs is desired or when a more complex evaluation is of interest. To perform such operations, first, a set of input $SPP_{MOI}$s are converted into pixel-based images with equal pixel grids. Afterwards, the spatial expression is evaluated on a pixel-by-pixel basis. Calculation artifacts such as division by zero (i.e., absence of a peak in that pixel) are computationally dropped during the conversion back to SPP, while low values in the denominator (representing detector baseline or very low peak intensities) are not expected because *moleculaR* works mainly with centroided (SNR $\geq 3$) MSI data. The resulting raster image is then converted back to an SPP whose points are carrying the respective computed pixel intensities. This SPP is then fed into the MPM framework, i.e., no arithmetic operations are applied on the hotspot/coldspot contours. Importantly, even though Gaussian-weighting does improve signal reliability, each ion intensity image will still contain an unknown amount of nonbiological technical variability[6], which could be carried on to the composite image representation. Caution is advised when creating complex composite images involving the division of two variables, as these may be particularly prone to uncertainty propagation.

## Synthetic data generation

In order to test the MPM method against ground truth, SPP data were simulated based on four spatial patterns of simulated hotspots: (1) a single central circle, (2) five equidistantly placed circles of the same size, (3) a ring-like simulated hotspot, and (4) a dominant central circle with four adjacent smaller ones. Each simulated hotspot pattern was placed within a square window of 100 length units denoting the background. Points were distributed within simulated hotspots and background window according to a homogeneous Poisson point process with spatial point densities $\Lambda$ of 0.4 and 0.3 points per unit area for simulated hotspots and background, respectively. Points' intensity values (marks) were sampled from above and below the upper quartile of the empirical intensities of a MALDI-FTICR-MSI

measurement of a human GB tissue sample at $m/z$ 544.3009 (PE(20:1) $[M+Na]^+$; FDR ≤ 0.2) for the simulated hotspot and background, respectively. The difference in spatial point densities and marked intensities accounts for the increased signal intensities and spatial density of peak signals, which is normally observed for areas of high relative abundance of an MOI (i.e., high spatial autocorrelation). We hypothesize this to highlight a biological process spatially localized within a given tissue morphology.

## Artificially added noise

The nature of noise in MSI, especially for FTICR data mainly used here, is still a matter of debate[54]. Here, in order to assess the stability and robustness of MPMs against noise sources observed in MALDI-MSI, we artificially "contaminated" raw data with different noise types: (1) random Gaussian noise, (2) presence of abnormally high-intensity peak artifacts ("intensity artifacts"), and (3) added overlapping peaks $2\sigma_G$ away from MOI $m/z$ ("interference"). For added Gaussian noise, intensities were sampled from a Gaussian distribution $f_{Gaussian}$ with $\mu_{noise} = \mu_{MOI}$ and $\sigma_{noise} = \sigma_{MOI}$ and added to all pixels of the MSI data, where $\mu_{noise}$ and $\sigma_{noise}$ are the mean and standard deviation of $f_{Gaussian}$ and $\mu_{MOI}$ and $\sigma_{MOI}$ are the mean and standard deviation of the MOI intensity distribution $f_{MOI}$. MPMs were tested against a Gaussian noise source with varying $\sigma_{noise} = \sigma_k$ ($k = 0 \ldots 10$), where $\sigma_0 = \sqrt{\mu_{MOI}}$ and $\sigma_k = k\ \sigma_{MOI}$ for $k = 1 \ldots 10$. Note that for $k = 0$, the resulting noise is similar to Poisson noise with $\lambda_{Poisson} = \mu_{MOI} \gg 1000$. The same computational experiment was repeated, this time spiking sampled noise in the vicinity of MOI $m/z$ at MOI $m/z + 2\sigma_G$. For added intensity artifacts, noise intensities were sampled from a uniform rectangular distribution whose range far exceeded the range of the MOI intensity distribution $f_{MOI}$. These were added to $n = 10$ random pixels of the MSI data (Fig. 4) but also iteratively varied for values up to 5000 random pixels (≈20% of the total tissue pixels; Supplementary Fig. 24).

## Transcript expression profiling of TCGA and GTEx datasets

TCGAbiolinks[55] was used to download fragments per kilobase of transcript per million mapped reads (FPKM) and the clinical information of The Cancer Genome Atlas (TCGA) glioblastoma (GB) datasets from Genomic Data Commons (GDC) (https://gdc.cancer.gov). Patient samples characterized as "primary tumor" were retained ($n = 156$). The FPKM values were converted to transcripts per million (TPMs)[56]. TPM data of normal brain tissues ($n = 1671$) were downloaded from the Genotype-Tissue Expression (GTEx) dataset (https://gtexportal.org). All TPM values were log2-transformed. For bioinformatics analysis of TCGA and GTEx data, all pairwise comparisons were performed using Kruskal–Wallis and Wilcoxon rank-sum tests. All analyses were run in R (https://cran.r-project.org) version 4.1, and Bioconductor (https://bioconductor.org) version 3.14. All graphical representations were generated using ggplot2, RcolorBrewer, gridExtra, and ggridges packages.

## Statistics and reproducibility

All statistical tests and graphical depictions of data are defined within the figure legends for the respective data panels. For comparisons between two groups, two-sided Wilcoxon rank-sum tests were performed, as noted within the figure legends. $P < 0.05$ was considered statistically significant. Boxplots indicate median (middle line), 25th, 75th percentile (box), and whiskers which extend to the most extreme data point, which is no more than 1.5 times the length of the box away from the box. The robustness of the proposed methods was tested against several MALDI-MSI data featuring different modalities, data types, and tissue samples. MALDI-MSI experiments were replicated as follows: MSI measurements and analyses of human GB tissues, each measured in positive and negative ion modes, were performed twice on two serial sections (Figs. 3b and 5, Supplementary Figs. 27, 29–34). MSI measurements and analyses of mouse brain tissues were

performed eight times (Fig. 3c, Supplementary Figs. 6–8, 17, 28). MSI measurements and analyses of IDH-mut and IDH-wt human glioma tissues were performed on three different tissue sets (Fig. 3e, Supplementary Fig. 18). Some MALDI-MSI data reused from previously published studies (respectively cited) were not replicated: MALDI-TOF-MSI of the APP NL-G-F Alzheimer's disease mouse model (Fig. 3d and Supplementary Fig. 15), MALDI-FTICR-MSI of imatinib drug dilution series on the porcine liver (Supplementary Fig. 16), MALDI-TOF-MSI of gastrointestinal stromal tumor (GIST) tissue (Supplementary Fig. 20).

## Reporting summary

Further information on research design is available in the Nature Portfolio Reporting Summary linked to this article.

## Data availability

MALDI-MSI Data of human GB and glioma tissue sections, mouse brain tissue sections, and porcine tissue presented in this study are available on Metaspace (metaspace2020.eu) through the following link: https://metaspace2020.eu/project/abusammour-2022. The MALDI-TOF-MSI data of the APP NL-G-F Alzheimer's disease mouse model presented are available via ProteomeXchange with identifier PXD020824. The SwissLipids database is available through the Metaspace portal (metaspace2020.eu) and can also be downloaded as a separate file (swisslipids.org). Transcript Expression Profiles of TCGA and GTEx Datasets are available from Genomic Data Commons (https://gdc.cancer.gov) and Genotype-Tissue Expression dataset (https://gtexportal.org), respectively. Source data are provided with this paper.

## Code availability

The source code is available as a well-documented companion R package that implements the presented framework at https://github.com/CeMOS-Mannheim/molculaR alongside clear introductory sections and example code vignettes. The R package is equipped with a web-based graphical user interface (GUI) and could be deployed and hosted on a centralized server as described in the package link above.

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

## Acknowledgements

We thank T. Alexandrov and S. Mamedov for providing guidance in all matters concerning METASPACE, D. Hofmann for discussions concerning GB tissue, T. Enzlein for providing guidance with respect to *PlaquePicker*, and T. Bausbacher for technical assistance with MSI instruments and software. This work was supported by grants from the Deutsche Forschungsgemeinschaft (DFG, German Research Foundation)—Project-ID 404521405, SFB 1389—UNITE Glioblastoma to T.K., A.v.D., M.P., C.A.O., W.W., and C.H., by the BMBF (German Federal Ministry of Research) as part of the Forschungscampus "Mannheim Molecular Intervention Environment" (M²OLIE), projects M²oBiTE (grant 13GW0091B) and M²OTAN (grant 13GW0388B) to C.H. and I.W., as part of the Innovation Partnership "Multimodal Analytics and Intelligent Sensorics in Health Industry" (M²Aind), project M²OGA (grant 03FH8I02IA) to I.W. and C.H., within the framework FH-Impuls, by German Cancer Aid (70113515; Regulation of tumor immunity through the integrated stress response (ISR) in myeloid cells), and by the Klaus-Tschira Foundation, project MALDISTAR (grant 00.010.2019), to T.B. and C.H. Acquisitions of the solarix 7T XR, of the rapifleX MALDI-TOF MS and of the timsTOFflex MS were supported by DFG (Project 262133997), the Hector Foundation II and the BMBF (grant 161L0212F as part of the MSCorSys research core SMART-CARE) - all to C.H.

## Author contributions

D.A.S. conceived, formulated, developed, and implemented all new computational workflows, implemented the *moleculaR* R package, analyzed and interpreted most MSI data, generated most figures, and wrote the first draft of the manuscript; J.L.C. implemented the GUI and contributed to the *moleculaR* R package and figures; J.C. implemented multimodal image fusion; C.M. and C.R.G. generated and analyzed MSI data; S.S. and S.A.M. contributed the theoretical formulation and computations for FWHM curves; M.F.R. contributed schematic illustrations for the proposed method; T.B. and I.W. contributed to the theoretical formulation of the method; T.K. and W.W. planned the underlying study, contributed GB tissue, provided clinical data and clinical tissue evaluation; M.F. and M.P. contributed GB tissue and provided clinical evaluation; A.v.D. provided histopathological annotation of tissue samples; V.P., A.S., and C.A.O. performed transcript expression analysis, generated figures, and contributed data interpretation; C.H. designed and coordinated the overall work, conceived applications of the computational framework, interpreted data, and wrote manuscript with feedback from all co-authors.

## Funding

## Competing interests

T.B. and C.M. are employees of Bruker Daltonics, a vendor of mass spectrometry imaging equipment. However, the company had no role in this study. The other authors declare no competing interests.
