## [Peer Review File · Nature Communications]

REVIEWER COMMENTS

Reviewer #1 (Remarks to the Author):

The authors report a novel computational framework, called *molecularR*, to spatially investigate small molecule signals gathered from mass spectrometry imaging (MSI). One key novelty of *molecularR* is to rely on a data-driven mass range which is specific to each signal rather than the conventional fixed user-defined mass range for the generation of ion images. *MolecularR* ultimately provides a metabolite probability map (MPM), that can be segmented into regions of high and low analyte abundance. To highlight the robustness of this approach, the authors show high consistency of these segmented areas derived from the MPM of an exemplary sphingomyelin SM(d34:2)+H⁺ when confounded with gaussian noise, spiked artifacts or interfering peaks. Further, multiple metabolites can be combined into a common probability map, which enables arithmetic operations between individual MPMs and the visualization of complex composite measurements such as adenylate energy charge, alkali ion milieu or lipid (un)saturation.

The method is very clever and aims to address important fundamental challenges related to IMS data processing that currently hinders the validity of intensities presented in ion images.

The difference between the POI and the SPP of m/z 544.3033 in FigS1 is striking as finer structures are revealed in the SPP image suggesting a reduction in signal noise. In addition, the results of Figure 1C are encouraging and show high resilience of this method to common confounding factors in IMS. Although these results are encouraging, I believe this study lacks two key aspects that can potentially be answered.

1 - Clearly present the advantage of MPMs over POIs. Simply being data-driven does not suffice in my opinion to justify the value of *molecularR*. Are MPM more robust to analytical noise? What about the inter-sample variability or batch effect? Ultimately, it comes down to asking if the intensities obtained from MPMs are more reliable than the ones collected from POIs.

2 - An experiment that validates the *molecularR* workflow. Ideally a ground truth would be directly compared to the intensities provided by the MPM to quantitatively assess the method.

A calibration curve made of successive spots of internal standards (IS) of known concentration could potentially answer both of these points by providing a ground truth for spatial distribution and intensity to which both MPMs and POIs can be compared against. Also, replicating this experiment, would help investigate how the *molecularR* processing can normalize for the inter- and intra-sample

variability, which could potentially be presented as a key advantage of using SPP and MPMs over POIs. Finally, using IS representing different metabolite classes would also be very informative as their ionization properties differ, which could potentially be at least partially normalized by molecularR.

On the technical side, I have a few questions/ comments:

- The relationship between FWHM and m/z is key to the molecularR workflow but I could not find an explanation of why they are assumed to be in such close correlation. Is this an accepted relationship in the field? If yes, it should be mentioned for non-expert readers. If not, a clear explanation of why the chosen smoothing curve has been used to best to model this relationship should be added in the method section.

- The bandwidth of the gaussian kernel of the KDE has been optimized using Moran's I. As I understand it, increasing the bandwidth does improve the signal-to-noise ratio at the cost of spatial resolution. How is this trade-off accounted for and how much loss of resolution can be afforded? Why has Moran's I been used as a metric and how is it relevant for defining such a crucial value? I do understand the challenge of defining the optimal KDE parameters and Moran's I might be the best option, I would simply suggest the authors add their rationale on these points to the main text or supplementary if possible.

- Using spatially reshuffled intensity values of the SPP map to generate the CSR is very clever although I believe there are a few combinations to achieve CSR. How does the resulting MPM vary for different CSR generated from the same tissue section?

- Similarly, in 1F, using the WT tissue to generate the CSR is a very creative idea but it raises questions as it involves inter-sample and inter-individual variations that could potentially confound the observations made here. For example, would it be possible to evaluate how different sections of IDH-wt would impact the areas of significant Trp presence?

- In 2B, the authors claim the Adenylate Energy Charge and Adenylate Kinase Mass Action Ratio scores are indicative of viable tumor (VT). I am not sure to understand how it is the case or how they came to that conclusion. Is it by colocalization evaluation? If anything, these scores indeed share some locations with the VT areas but are in now way indicative of them. I believe such conclusions should be either re-evaluated or better justified.

Reviewer #2 (Remarks to the Author):

Recommendation: Major Revisions

This work focuses on mass spectrometry imaging (MSI) as an important approach for studying metabolites in tissue. I agree with the authors when they point out data analysis of MSI measurements as an important area for development, and support their efforts in providing new software to accomplish this. I appreciated the manuscript's move away from traditional ion images, and the introduction of probabilistic mapping and spatial point patterns (SPPs) as novel approaches for MSI data. However, several important questions and issues would need to be resolved before publication can be considered. The comments can be grouped as follows:

- Comments on potential method artefacts and result reliability: The manuscript needs a more stringent (quantitative) assessment of whether the computational methods (and assumptions they make) introduce artefacts or remove potential biological information. Examples: Can kernel density estimation potentially reduce or eliminate secondary hotspots (that are smaller than the main hotspot) when the bandwidth estimation is optimized for the dominating hotspot and is applied tissue wide?; The manuscript does not provide much discussion on whether raw intensity values are reporting biology only or still contain technical variation as well, and what the effect is when translating these values into MPMs and CPPMs. Since ion images commonly include non-biological variation and preprocessing is not perfect, what is the residual effect of non-biological variation in calculating probabilities for MPMs and CPPMs on the basis of those intensity values (do they reduce or amplify their effect), and what is the additional effect of follow-up arithmetic operations using those potentially perturbed probabilities? Also, what is the potential for calculation anomalies (e.g. in ratio calculations)? Given the statistical significance claims made in the manuscript, a quantitative assessment of MPM and CPPM reliability seems appropriate.

- Comments on method motivations, assumptions, and choices made: Several assumptions about the data and model/parameter choices are made with insufficient motivation for why these assumptions/choices are valid and with insufficient guidance for the reader on how to make these choices for their own MSI data. Example: Which evidence supports using a Gaussian shaped weighting scheme along the m/z axis for establishing an MOI intensity from multiple matching POI intensities?

- Comments on precision in wording, nomenclature, and mathematical definitions, and the need for additional clarity: The precision in wording, nomenclature, and mathematical definitions is not sufficient to avoid ambiguity. Examples: The manuscript seems to state it assesses statistically "significant presence" of an MOI. However, the comparison to the CSR checks for spatial autocorrelation and the original intensity values are e.g. changed/spatially smoothed using KDE. Significant spatial autocorrelation (even with an intensity component) is not the same as significant presence. Ions can be significantly present in tissue without concentrating spatially, while it seems the method presented here enriches only for the latter. More precise wording would be helpful here.; The manuscript is not very clear about its definition of sigma in the context of Gaussian distributions. At some locations sigma is implied to be the standard deviation. At other locations sigma is stated as being set equal to the FWHM of a peak, while the FWHM of a Gaussian shaped peak would be $>2\sigma$?

Detailed comments for each group are provided below.

Finally, since the manuscript covers a software package meant for broader application and in order to assess whether the observations in the manuscript are dataset-specific or method-specific, a demonstration on different MSI data types (e.g. non-FTICR) and on additional MSI case studies using different tissue types seems needed.

Comments on potential method artifacts and result reliability

- "a complete spatial randomness (CSR) model of that MOI is created by random spatial permutations of MOI points" and "the intensity distribution of the CSR density image is expected and observed to converge towards a normal distribution (Suppl. Fig. 3)" -> If the CSR is a spatially reshuffled version of the corresponding MOI, the same intensity values would be present in the CSR as in the MOI, but in different locations, so the histograms of intensities would be the same for both at this level. Then, the same KDE is applied to both the CSR and MOI, smoothing the intensity distributions using spatial location and the bandwidth parameter controlling the level of smoothing. The KDE effectively functions as a filter that eliminates/reduces hotspots (by smoothing them out) that are smaller than the bandwidth parameter lets through, in both the CSR and MOI cases. Is it fair to eliminate smaller hotspots by smoothing them? In the MOI case particularly, this seems to hold risk in the case of ion distributions that have several different sized hotspots. Only the larger most

dominant hotspot (which ends up influencing the bandwidth parameter most) would seem to survive this process (without getting reduced in spatial size or being eliminated entirely). Please explain how this method would avoid eliminating or reducing smaller (but biological and therefore relevant) features in an MOI distribution. Since it would impact the biological distribution that ends up in the MPM, the manuscript needs a more exhaustive assessment of this. Suppl. Figure 3 shows, particularly for m/z 701.5592 and 666.4340, how as the bandwidth increases, a bimodal or trimodal distribution is smoothed into a unimodal form. The question remains whether it is fair to remove those spatially smaller pockets of ion intensity (or even reduce them in size) since spatial area alone cannot determine if an ion distribution is biological in nature or noise of some form, and so it seems there is the possibility for biological patterns to get substantially modified or removed in the process. Please provide guidance.

- (Figure 1) "are robust against noise and common signal artifacts" -> This statement seems somewhat broad. There are several types of noise, commonly present in MSI data, against which this method would not be robust. The manuscript demonstrates adding Gaussian noise to ion intensities, which is not a very common noise type in mass spectrometry data (e.g. Poisson noise is more common), particularly in FTICR data where the intensity range being reported is large. Additionally, robustness to noise is usually a matter of how strong the noise patterns are. The demonstration seems to apply Gaussian noise using a single variance value. Please show robustness for several Gaussian distribution widths so it can be assessed at what level of noise the method's robustness breaks down. The same argument holds for the spiked artifacts and interfering peaks cases, in that those probably also have parameters in function of which robustness should be reported. Please provide a narrow and precise definition of what type of noise the method is robust against.

- "enabling true spatial probabilistic cross-tissue comparisons" -> Figure 1F is given as an example of how comparisons can be made between two tissue samples. It is not entirely clear what the reader should take away from the panel. The statement "Significant Trp presence" seems imprecise. The statistical test locates areas where there is more than random spatial clustering it seems. Whether the amount or intensity is significant is hard to assess since the intensities (that the probabilities are based on) get changed quite a bit in the process (e.g. are the intensities directly comparable to begin with? -> see preprocessing question; the intensities can furthermore be weighted by Gaussian weights and potentially transformed with a square root transformation it seems, etc.). The statement "Significant Trp presence" seems too broad to capture that nuance. Please provide a more precise comparison example of what there can be learned from panel 1F. How should the MPM contour be utilized in the reader's datasets. Please provide some guidance on what this allows in terms of comparison between tissues and what it does not allow.

- "permits spatial evaluation of composite numeric scores obtained by applying basic arithmetic operations on spatial point patterns of multiple MOIs" and "adenylate kinase mass action ratio" -> Although it is technically possible to use the MOI values for calculation, the real question is how much uncertainty they are accompanied with and how well they represent actual biological

abundance. While this is already the case for the raw intensity values present in standard ion images, the authors should address whether their subsequent processing and transforming steps to get to MPMs and CPPMs do not add additional uncertainty on top of the biological and technical uncertainty present in the raw measurements. Please provide a quantitative assessment. Furthermore, please address issues with taking a ratio, which is used in the examples of the manuscript and is somewhat susceptible to introducing unforeseen calculation artifacts when uncertainty is not controlled for. For example, zero or low values in the denominator can lead to respectively NaNs or exaggerated values in the calculated ratio result. The manuscript states that NaNs are removed, but that does not address the underlying issue. Please address how unintended effects of direct calculation with intensity values can be discovered and countered.

- Were the results of the FWHM estimation (on top of Friedman's super smoother) assessed for quality? If so, how does this translate to other MSI datasets?

- "For any number of MOIs, basic arithmetic operations on the spatial point patterns of MOIs could also be applied. This is useful when a ratio of two MOIs is desired (ex. Fig. 2B bottom row) or when a more complex evaluation is of interest (ex. Fig. 2B top right and bottom right)." -> The direct use of the intensity values in arithmetic operations implies that the intensity values only report tissue content and do not report technical variation or noise. Is this the case? Is noise effectively removed leading up to this point in the process? How does the square root transformation affect these arithmetic operations? Please explain why this is valid, and what preprocessing procedures were applied to this data such that technical noise has been sufficiently removed to make direct calculation reliable.

- "Possible divisions by zero are computationally dropped." -> Since divide by zero leads to NaNs, their removal seems evident. However, dividing by very small (nonzero) numbers has a similar exaggerating effect (very large numbers that overrepresent the true values, despite not being NaNs). How are these handled by the software? How do the users know their end result of the arithmetic operation they applied does not have this instability?

- (Suppl. Fig. 4) The comparison using the Dice coefficient shows a substantial number of MOIs for which the DS drops below 0.75. This suggests that the biological distribution of the MOI gets substantially changed by the method presented. Is this not an issue? How should the reader assess whether the MPM they get at the end is reliable? In this comparison, known noise sources were added to the original data and so there is something to compare against. How should this reliability be assessed when the true noise in the data is not known, which would be the common situation for most MSI measured MOIs. Please elaborate.

Comments on method motivations, assumptions, and choices made:

- The authors make the case for approaching a MSI dataset as a spatial point pattern (SPP) and doing molecular spatial probability mapping on that SPP. However, since the regular grid of locations in an ion image is implicitly an SPP (with every pixel representing a point, and with potentially low intensity values still included), what is the major reason MSI analysis should move to SPPs? For example, MOIs, CPPMs, and hotspot contours could also be calculated directly on ion images it seems. Please explain more precisely why SPPs are needed for the process and why they are beneficial to the MSI user.

- "arithmetic operations within the same MS image" -> The method seems to assume that the peak intensity of POIs reflects the abundance of a species directly. Since this is not always the case and preprocessing is often not perfect, using the weighted peak height for consecutive calculation carries the potential for amplifying the non-biological variation in the intensity distribution. Please assess this quantitatively and acknowledge it in the manuscript.

- "use the sum of ion intensities of all peaks present in a user-defined mass range centered on the POI m/z instead (Suppl. Figure 1A)." -> The manuscript refers to a user-defined mass range for integrating the ion intensity around a specific m/z value. However, most peak picking algorithms, also elsewhere in mass spectrometry and spectroscopy, automatically determine a (m/z) window around each found peak and provide it for integration. The argument for human bias being introduced by manually specifying a mass window around a peak therefore does not seem to fit here, but specifying it around an MOI's m/z taken from a database might fit? However, in that case, the question becomes why the problem is approached from the perspective of the MOI (for which a window needs to be specified) instead of the measured POIs (for which the windows can be measured from the full profile spectra directly). Please clarify what the exact type of data is that is being started from and/or why there is a MOI-centric approach being taken, so that it's clear why there are multiple separately detected POIs that need to be summed together in a weighted fashion.

- "we introduce an MOI-centric biomedical perspective that systematically analyzes and visualizes if biologically relevant MOIs have a statistically validated spatial presence across tissue morphologies."
-> There are often going to be more MOIs present in databases to match to than there are POIs present in the measured dataset. This seems to imply that approaching the problem from an MOI

perspective could lead to more false positive matches to POIs than when trying to match a POI to potential MOIs. Please elaborate on why an MOI-centric approach is better. Also, we know for sure a POI was present and detected in the dataset. We do not know if a particular MOI is present in the dataset. How does the method address this one-to-many mapping (in terms of overlap and double-counting of POIs, and in terms of computational scalability) and how does it address the uncertainty of such a POI to MOI assignment?

- "we propose molecular probability maps (MPMs) for rigorous user-independent spatial statistical testing that are based on the assumption that for any given MOI spatial autocorrelations exist that mirror the biological interaction between neighboring tissue morphologies" -> Please explain why it's fair to assume that any MOI's spatial distribution demonstrates spatial autocorrelation. There are examples of biologically relevant molecular species that do not show strong spatial autocorrelation, yet are important. Does this approach not work for these species? Are they filtered out as not statistically significant? Should the user remove such ion distributions beforehand? Please specify in the manuscript.

- "are based on the assumption that for any given MOI spatial autocorrelations exist that mirror the biological interaction between neighboring tissue morphologies" -> An additional implicit assumption seems to be that an MOI is present in the data and that there's no uncertainty related to the POI-MOI match. What happens if the MOI is not present in the data, but a set of peaks close to that MOI's theoretical mass happen to show spatial autocorrelation? Does that count as a significant match for the statistical test? Please elaborate.

- (Suppl. Figure 1) In panel 1b, a Gaussian weight profile is multiplied with individual peaks. How does the method know it is dealing with a single Gaussian peak and not a mixture? If this is an assumption, what is the potential effect of this assumption when it does not fit reality. One example is shown in panel b with quite a high peak to the right of the Gaussian profile getting severely down-weighted. This might be correct to do, but the the manuscript should explain why that is the case, and how we can tell that that's the right thing to do.

- (Suppl. Figure 1) "A user-selected full mass spectrum is used to compute full width half maximum (fwhm) values of its peaks across the m/z axis." -> How should the user select a mass spectrum to accomplish this? Also, would that not require that the chosen spectrum is representative for the entire dataset, which can contain tissue subareas that are molecularly distinct from the location where the reference mass spectrum is pulled? Please elaborate.

- (Suppl. Figure 1) "A curve is fitted to describe the continuous relation of fwhm as a function of m/z." -> Please elaborate on why it is fair to assume that there is a linear relationship between

FWMH and the m/z value? In complex mixtures, this seems like it would be a hard argument to make.

- (Suppl. Figure 1) "A Gaussian envelope, whose sigma is inferred from the fwhm model at MOI, is centered on MOI. The Gaussian is then used as a weighting factor to protect against proximal background signals; the further the measured m/z (= POI) from the theoretical m/z (= MOI), the lower the weight it receives in the SPP representation." -> It seems like the method assumes that the MOI is present. What happens if the match is incorrect and the MOI is not present? Are the POIs still summed (in a Gaussian weighted way) and made to represent the MOI? Please explain in the manuscript how the uncertainty of the POI-MOI match is taken into account.

- (Suppl. Figure 1) "An exemplary comparison between standard ion image and SPP representation of [Heme+K]⁺." -> Please explain what the fundamental difference is between representing ion presence as a regular grid (where absence of a molecular species is depicted as a zero value) and a spatial point pattern (where absence of a molecular species is depicted as an absent data point). Furthermore, since both approaches can represent absence, the manuscript should make clear what the advantages and disadvantages are of representing data either way.

- (Figure 1) "curve-fitted to describe FWHM as a function of m/z " -> Please explain why a linear relationship is an appropriate assumption. Also, in panel a, please provide the actual m/z values along the m/z axis.

- (Figure 1) In panel b, kernel density estimation is referred to, which comes with choices to be made (e.g. the choice of kernel used). Please explain why the choice for a (isotropic) Gaussian kernel is the right one here? What are the implications if this choice is not right? How should a user that wants to employ this method on her own data choose? Please acknowledge method choices in the manuscript and explain the reasons for the choices made here.

- (Figure 1) "added overlapping peaks 2 sigma away" -> Why specifically 2 sigma? Why is that the right location to put an overlapping peak? Would 1 sigma not be a more overlapping peak? Please explain the parameter choices, and motivate why these are the right values to evaluate at.

- "(MALDI) MSI raw data of any given MOI m/z is first transformed into an SPP representation" -> The formulation of this sentence implies that the raw data measured in the MSI experiment "belongs" to the MOI. However, the link between the raw data and the MOI is based on a numerical matching of m/z values (with an uncertainty window and an additional user-defined Gaussian weighting), there is no definitive link between the two until traditional mass spectral identification

has confirmed the link. How does the method take the tentative nature of the POI-MOI link into account? What happens if the match turns out to be incorrect? Is that a responsibility for the user? Please elaborate and have the tentative nature of this link reflected in the manuscript text.

- "which then forms the basis for inferring intensity cutoffs beyond which the intensities of MOI's density image are unlikely to occur if generated by a random spatial process (see Online Methods)" -
> Since the KDE has a smoothing effect that can filter out or reduce potential biological hotspots smaller than the largest hotspot, is this determination of cutoffs fair? Please motivate.

- "statistically-validated significant localization of immunosuppression-associated tryptophan in IDH-mutant- compared to IDH wild-type glioblastoma" -> Since the CSR model of the test tissue is inferred from signal intensities of the reference tissue, it implies that the signal intensities between the mutant and wild type measurements were directly comparable. How was this accomplished? How was the preprocessing done to ensure that the intensities between the two samples were on the same scale, without the potential for skewing e.g. by the number of pixels in each sample or e.g. the substantial amount of different tissue structure between mutant and wild type? Please explain this aspect and acknowledge in the manuscript.

- "Cropped images of both modalities were resampled" -> What type of resampling was used? Please report the type of resampling used, the parameters involved, and motivate why these choices are appropriate in this case.

- "with a resolution of $7.5 \mu\text{m}^2$ per pixel" -> Assuming isotropic resampling, do the authors mean that the resampled pixel describes a square of $2.739 \mu\text{m}$ ($=\sqrt{7.5}$) by $2.739 \mu\text{m}$? Why this choice? Regardless of the specific pixel area resampled to, why this target value? How should a reader decide what area to resample to?

- "down-sampling factors of 4,2,1" and "Advanced Mattes Mutual Information metric" and "250 iterations"-> Please provide motivation for these parameter choices, and guidance for the reader to make their own choices for their datasets.

- "interpolation using third-order B-Splines" and "750 iterations" -> Same comment as above.

- "manually defined control points were added at corresponding locations in both modalities to support the deformable registration" -> How many locations? Does this manual step influence any analysis results later on? If not, has this been quantitatively verified?

- "Matt's Mutual Information metric and the Corresponding Points Euclidean Distance metric (equally weighted)" -> Why is equal weighting the right mixture? Why was the Euclidean Distance metric added, while the previous step only used the other metric? Was there an issue with the mutual information metric that the Euclidean Distance metric solves? Please explain.

- "to control registration parameters" -> Were these set automatically, or manually using the M2aia application? Please explain.

- "Peaks that occurred in less than 1% of the pixels were filtered out" -> Please explain why 1%, and not for example 5%? How should the reader make this assessment for their own data?

- "after being binned to a relative tolerance" -> Please elaborate on the specifics of this procedure. Are the bins 12 ppm wide? Are they sequential or can they overlap? How are the bin centers decided?

- "a relative tolerance of 12 ppm (i.e. the maximal relative deviation of a peak position to be considered as identical)" -> Please elaborate on why 12 ppm is an acceptable range for this mass range.

- "effective support ($mMOI \pm 3\sigma$)" -> Why is the Gaussian profile the appropriate weight profile to use? Why not another profile that weighs down peaks that are farther away? Was the influence of this choice assessed on the results?

- "Given a set of MOIs" -> How is this list determined? Is it provided by the researchers? Provided by Metascape? Please elaborate. What happens if MOIs overlap in terms of which POIs they connect to? How is the overlap handled if there's no one-to-one match?

- "allowing for coordinate duplication" -> Please explain the coordinate duplication referred to here. Is maybe overlap meant?

- "results in accumulation of the corresponding signal intensities" -> Are the MOIs guaranteed to be non-overlapping in terms of which POIs they connect to (even using lower weights of the Gaussian

distribution)? If not, why is it fair to add the SPPs of the MOIs together since that could mean certain POIs are counted multiple times? Please explain in the manuscript.

- "The workflow then commences with the square root transformation of intensities to compensate for the inherent heteroskedasticity and possible differences in ionization efficiency between the individual MOIs" -> Why does the manuscript consider square root downscaling an appropriate way of reducing the effect of ions with high ionization response versus ions with low ionization response? Since this is FTICR data, the ionization intensity response can be orders of magnitude different between low and high intensity ions. Square root transformation would not remove that effect necessarily, and the CPPMs could still be dominated by high intensity ions. Please elaborate.

- "converted into pixel-based images with equal pixel grids" and "The resulting raster image is then converted back to an SPP whose points are carrying the respective computed pixel intensities"-> The manuscript suggests that SPPs are an improved way of representing of MSI data, and describes the process from grid to SPP. Here, the process is reversed to perform arithmetic operations on a grid representation. Why not do the arithmetic operations on the values in the SPPs directly?

- "with analyte hotspots and/or analyte coldspots superimposed as polygonal contours identifying areas of MOI significant abundance and deficiency, respectively." -> The manuscript seems to assume that MOIs have to spatially autocorrelate. What happens for MOIs that do not (e.g. because their spatial distribution pattern is homogeneous)?

Comments on precision in wording, nomenclature, and mathematical definitions, and the need for additional clarity

- "poorly defined ion images for data interpretation" -> Please explain in what way ion images are "poorly defined".

- "systematic probabilistic mapping" -> What makes this "systematic"?

- "Ion images currently used in MSI do not represent ion intensity of a single observed POI in a user-unbiased way." -> Please elaborate on what is meant exactly when making this statement. The generation of ion images is done using different approaches. For example, both the peak height of a centroided peak and integration over a peak in full profile spectra are common ways of translating spectra into intensity values in an ion image. Does the bias affect both? Other approaches are in use as well, so if the manuscript wants to make a statement in terms of bias, it is necessary to be precise on what type of bias is being referring to. The sentences following this one are insufficient in narrowing this down (see also my comment dedicated to specifying the peak picking method the manuscript starts from).

- "They rather neglect mass accuracy and resolving power at the POI m/z " -> More clarity is needed here. The manuscript seems to refer to the case where peak picking is done and then an ion image is generated for a specific peak that was found. Most peak picking algorithms report not only the m/z value of the peak centroid, but also directly report some measure of peak width such as its FWHM. Such a peak width measure reports a heuristic for resolving power, and this information is often used to determine what m/z window to integrate over and thus ends up affecting the ion image directly, so it's not clear what the manuscript means when it says that mass accuracy and resolving power is neglected, at least not in the general case.

- (Figure 1) "Full-width-at-half-maximum (FWHM) values are computed for all peaks of a user-selected full mass spectrum" -> A traditional full profile mass spectrum would describe a peak using multiple neighboring intensity values, covering the peak from left to right. The figures show spectra that look like centroided spectra. It would be good to bring clarity to which type of data is in use here, also because peak picking algorithms on traditional full profile spectra tend to provide FWHM estimations directly, and thus would not require an estimate of FWHM values afterwards as seems to be the case here. Please clarify and provide additional guidance.

- (Figure 1) "For any MOI m/z (dashed blue line), a Gaussian envelop is computed with sigma equaling the estimated FWHM at MOI m/z " -> Please motivate why the decrease in weight should follow a Gaussian model. Also, if a Gaussian is the right model, please clarify the definition of sigma in this context. If the manuscript intends to have sigma represent the Gaussian curve's standard deviation (a common notation), it does not seem to make sense to equal sigma to the FWHM. The FWHM of a Gaussian curve is not equal to one standard deviation, in fact it is not exactly equal to 2 standard deviations even, so there seems to be a miscommunication here. Does the sigma in the manuscript mean something else than the standard deviation? Please provide a precise set of definitions and nomenclature to avoid ambiguity.

- (Figure 1) "to estimate areas of significant abundance" -> The statistical test against a CSR seems to test whether there is spatial autocorrelation in the distribution pattern, not necessarily whether the abundance/intensity is significant. There can be an element of intensity included in the test by using

the KDE smoothing, but that does not change the experimental design of testing against a CSR model. Using "areas of significant abundance" seems to imply an intensity argument, which does not seem to be supported by the nature of the test? Please state more clearly in the manuscript what property the significance of which is being tested, and what the implications are for interpretation of the results, for example in Figure 1F.

- (Figure 1) "(MOI "hotspots"; red/white contours) or deficiency (MOI "coldspots"; blue/white contours)" -> What threshold was determined as the border of significance for both of these? Please state them in the caption as well.

- (Figure 1) "raw, noise-free (green)" -> The manuscript states the raw version to be noise-free. Is the raw intensity data noise-free to begin with? How can a user assess this? Please elaborate.

- Suppl. Figure 2's caption states "statistically significant". -> As mentioned in other comments, please elaborate on what property exactly the significance is tested of, and state the statistical significance thresholds employed.

- "with an POI-specific bandwidth estimation" -> The values in the SPP are stated to be MOI-specific. When doing the kernel estimation, why go back to a POI-based estimation of the kernel parameter?

- "spatial probabilistic mapping of analytes aids in outlining the significant presence or absence of analytes relative to vital tumor regions" -> This statement seems somewhat imprecise and generalizing. The approach presented reports whether there is spatially local concentration of an analyte (by comparing it to a CSR distribution that has random spatial distribution), it does not report whether the intensity (or absence/presence) of an analyte is significant. Analytes can be significantly present in terms of intensity without their spatial distribution concentrating in certain spots (and thus being spatially autocorrelated). The method presented here only seems to establish the significance of whether spatial autocorrelation occurs. One example would be an ion that is present across the entire tissue being measured. It can be significantly present in the sample, in that it shows up in amounts and with intensities that are significant and that cannot be easily confused with intensity noise levels. The method presented here would only seem to judge whether the ion's distribution spatially autocorrelates. Since inferential statistics and the determination of 'significance' carry a lot of scientific weight, please elaborate on this in the manuscript. Also, since the current statements could lead to misunderstandings, please reformulate wording such as "significant presence or absence" to more clearly separate out for the reader the concepts being tested.

- "statistically-validated significant localization" -> See other comments on this. Please be precise on what property is being statistically validated. 'Localization' is insufficiently precise to differentiate between presence/intensity and spatial clustering/autocorrelation.

- "maps of entire lipid classes" -> How are these molecular ensembles defined? Are they groups of molecules that are defined by software on the basis of the measurements in the MSI data? If so, please provide how that works and what procedure was used to obtain the ensembles used in the manuscript. Are these groups of molecules manually curated? If so, please provide how it is avoided that POIs are counted more than once when they happen to match to more than one of the MOIs in the ensemble. Overall, please provide more clarity on the ensemble work.

- "5 μm^2 per pixel" -> Do the authors mean a pixel of 5 μm by 5 μm , or really 2.236 μm by 2.236 μm , which amounts to 5 μm^2 per pixel?

- "0.5 μm^2 per pixel" -> Same question as above.

- "Matt's Mutual Information metric" -> Earlier in the manuscript, the "Advanced Mattes Mutual Information metric" is mentioned. Is this the same metric? If so, please keep the naming consistent. If not, please explain the difference.

- "The centroided MALDI FTICR dataset" -> The statements in the supplementary "MALDI Mass Spectrometry Imaging" section suggest full profile data, with low intensity values set to zero. That is not commonly considered centroided data. Is there maybe a step missing between the description in "MALDI Mass Spectrometry Imaging" and "Data Preprocessing"? Please provide clarity on the conditions of the data, and show a raw example in a graph. Particularly given the manuscript's work towards summarizing peaks together using a Gaussian weighting scheme, which does not seem to be required for a full profile spectrum, but would make sense for centroided data, it is important that the manuscript makes clear where the challenge being solved enters the data.

- Given the resolving powers involved and the use of FTICR measurements, how does the method address isotopes in the data?

- "One pixel representing a full mass spectrum was randomly chosen" -> Does the manuscript mean 'full' in terms of covering the complete m/z axis, or 'full' in terms of "full profile"?

- "To minimize data load, data was saved as Profile Spectrum with a Data Reduction Factor of 97%." -
> The "Data Preprocessing" section in the manuscript states "The centroided MALDI FTICR dataset", while this statement seems to suggest full profile data, at least for the more abundant peaks. Please add more precision about what the conditions are of the data from which the method starts.

- "compute the expected data-dependent fwhm, which is then used to determine the sigma of a Gaussian envelop" -> The wording and the display in Figure 1 seem to indicate that sigma is matched to the FWHM. The sigma in most common definitions of the Gaussian distribution represents the standard deviation, which is not equal to the Gaussian peak width at half maximum. Are the authors using an alternative definition of the Gaussian distribution in which sigma is not the standard deviation? If so, please be precise on the definition of sigma. If the sigma does represent the standard deviation (which seems to be the case later on), further explanation is needed since the FWHM of a Gaussian distribution is roughly two-fold off from the standard deviation.

- (Suppl. Fig. 4) "contaminated data" -> Why call this 'contaminated'? Do the authors mean data changed by their algorithm? Please use a more descriptive label.

Minor comments & typos

- (Figure 1) "(green mesh)" -> The green mesh is hard to see in the image. Please adjust.

- "a mass resolution of 85k" -> Shouldn't this be resolving power? It's listed as a unitless value in the text.

- Please rewrite sentences such as "m/z values of observed POI and biologically relevant MOI like the potassium adduct of heme ([Heme + K]⁺), i.e., database entries with corresponding theoretical masses, typically differ."

- "MPMs enables"

- Suppl. Figure 2. lists "infliction = 2.4". Should this be 'inflection'?

- "which equals to the number of points"

- "two exemplary MOIs" -> "two example MOIs"

Point-for-Point Reply to Reviewers

Contents

1. Reviewer #1.....	3
1.1. Comment	3
1.2. Advantages of MPM over peak-of-interest (POI) ion images	4
1.3. Ground-truth.....	7
1.4. FWHM estimation	8
1.5. KDE Gaussian bandwidth	8
1.6. Permutations of CSR	9
1.7. Inter-sample variations.....	10
1.8. Colocalization indicative of VT	11
2. Reviewer #2.....	11
2.1. Secondary spots.....	11
2.2. Non-biological variations	14
2.3. Effect of arithmetic operations.....	16
2.4. Gaussian weighting	17
2.5. Significant presence	18
2.6. Definition of Sigma.....	18
2.7. Demonstration of different data types.....	18
2.8. Compromising smaller hotspots by KDE.....	19
2.9. Artificial noise	22
2.10. Test vs control tissues	24
2.11. Arithmetic operations and uncertainty.	25
2.12. FWHM estimation	27
2.13. Effect of non-biological variations	28
2.14. Division by zero or small values	28
2.15. Artificial noise and Dice coefficient	29
2.16. Reason of using SPP representation	29
2.17. Gaussian weighting and non-biological variations	30
2.18. Gaussian weighting and MOI	30
2.19. MOI-centric approach and double counting.....	32
2.20. Spatial autocorrelation and biological relevance	32
2.21. POI-MOI matching.....	33
2.22. Gaussian weighting	34

2.23.	FWHM estimation full spectrum.....	34
2.24.	FWHM curve fitting.....	34
2.25.	What if MOI is not present?.....	35
2.26.	SPP representation.....	35
2.27.	FWHM curve fitting.....	36
2.28.	KDE choice of kernel	36
2.29.	Artificial noise overlapping peaks	36
2.30.	POI-MOI matching.....	37
2.31.	Smaller hotspots	37
2.32.	Test vs control tissues	37
2.33.	Registration	38
2.34.	Registration	38
2.35.	Registration	39
2.36.	Registration	39
2.37.	Registration	40
2.38.	Registration	40
2.39.	Peaks filtering.....	41
2.40.	Binning	41
2.41.	Binning	42
2.42.	Gaussian weighting	42
2.43.	MOI list.....	42
2.44.	Coordinate duplication	42
2.45.	Double counting.....	43
2.46.	Sqrt transform and Heteroscedasticity.....	43
2.47.	SPP representation.....	44
2.48.	Spatial autocorrelation and biological relevance	45
2.49.	Ion images	45
2.50.	Comment on the word “systematic”	45
2.51.	Bias in ion images	46
2.52.	Peak picking and FWHM	46
2.53.	FWHM full profile spectrum	47
2.54.	Gaussian weighting sigma	47
2.55.	Significant abundance	48
2.56.	Coldspot/hotspot thresholds.....	48
2.57.	Artificial noise	48

2.58.	Significance	48
2.59.	POI-specific bandwidth	49
2.60.	Spatial autocorrelation and biological relevance	49
2.61.	Significant localization	49
2.62.	Maps of lipid classes	50
2.63.	Pixel resolution - registration.....	50
2.64.	Pixel resolution - registration section	50
2.65.	Registration.....	50
2.66.	Centroided data	51
2.67.	Isotopes.....	51
2.68.	Full mass spectrum	51
2.69.	Profile vs centroided	51
2.70.	Sigma and FWHM.....	52
2.71.	Artificially contaminated data.....	52
2.72.	Typos.....	52

1. Reviewer #1

1.1. Comment

The authors report a novel computational framework, called `moleculaR`, to spatially investigate small molecule signals gathered from mass spectrometry imaging (MSI). One key novelty of `moleculaR` is to rely on a data-driven mass range which is specific to each signal rather than the conventional fixed user-defined mass range for the generation of ion images. `MoleculaR` ultimately provides a metabolite probability map (MPM), that can be segmented into regions of high and low analyte abundance. To highlight the robustness of this approach, the authors show high consistency of these segmented areas derived from the MPM of an exemplary sphingomyelin $SM(d34:2)+H^+$ when confounded with gaussian noise, spiked artifacts or interfering peaks. Further, multiple metabolites can be combined into a common probability map, which enables arithmetic operations between individual MPMs and the visualization of complex composite measurements such as adenylate energy charge, alkali ion milieu or lipid (un)saturation.

The method is very clever and aims to address important fundamental challenges related to IMS data processing that currently hinders the validity of intensities presented in ion images. The difference between the POI and the SPP of m/z 544.3033 in FigS1 is striking as finer structures are revealed in the SPP image suggesting a reduction

in signal noise. In addition, the results of Figure 1C are encouraging and show high resilience of this method to common confounding factors in IMS. Although these results are encouraging, I believe this study lacks two key aspects that can potentially be answered.

We thank the reviewer for considering our molecular approach as “clever”, addressing “fundamental challenges related to IMS data processing” and for qualifying “results ... encouraging”. We also thank the reviewer for pointing out that “this study lacks two key aspects that can potentially be answered”. Please, find our answers below.

1.2. Advantages of MPM over peak-of-interest (POI) ion images

1 - Clearly present the advantage of MPMs over POIs. Simply being data-driven does not suffice in my opinion to justify the value of molecular. Are MPM more robust to analytical noise? What about the inter-sample variability or batch effect? Ultimately, it comes down to asking if the intensities obtained from MPMs are more reliable than the ones collected from POIs.

1.2.1. Are MPM more robust to analytical noise?

In the original manuscript we had shown that MPMs but not ion images of exemplary SM(d34:2)+H⁺ are robust against noise and common signal artefacts, e.g., random Gaussian noise, presence of abnormally-high-intensity peak artefacts and added interference peaks $2\sigma_G$ away from the MOIs m/z . MPM hotspots in raw and artificially/computationally „contaminated” data had high overlaps, as judged by their Dice similarity coefficients (DSC) (**Fig. 1C** in the **original** manuscript). We had also computed DSC for areas of significance (MOI “hotspots”) between raw- against artificially/ computationally “contaminated” MSI data of 142 MPMs of MOIs (**Suppl. Fig. 4** in the **original** manuscript).

To address the reviewer’s question, we have now done several additional experiments and data analyses summarized in the **new Fig. 4** and **new Suppl. Fig. 16, 17 & 18**. **Former Fig. 1C** is **now Fig. 4C**, to which new parts **A** and **B** have been added for better illustration and clarification. **Former Suppl. Fig. 4** is now **Suppl. Fig. 16**. During the revision we observed that the previously used strict Bonferroni P-value correction had a negative effect on the stability of the computed hotspot/coldspot contours. Therefore, we switched to Benjamini-Hochberg P-value correction, which had a positive impact on the stability and reproducibility of the method, as demonstrated by higher median DSCs of the **new Suppl. Fig. 16** compared to the **original Suppl. Fig. 4**. We have therefore updated the method and used the new P-value correction approach throughout the revised manuscript. Most importantly, **we ran many more computations for different degrees of synthetically added noise in new Suppl. Figures 17 and 18**. Even at extreme (and unlikely) artificial noise “contamination” with obvious visual degradation and high dissimilarity between raw and artificially contaminated (see normalized sum of squared differences (SSD) curve) corresponding ion images, MPMs were still able to detect and localize parts of the raw hotspot contours (**new Suppl. Fig. 17** and **18**). A detailed analysis of noise sources for various MSI platforms goes beyond the scope of this study. The origin of noise in MSI (e.g. Gaussian,

Poisson, other noise) has mainly been analyzed in more detail in the literature for TOF mass analyzers¹. For FTICR data that was primarily used here, and which is not based on a counter but on ion cyclotron frequencies instead, the situation is less clear. It has been stated by Zhurov et al.: “The major noise component is assumed to be thermal noise from the ion detection electronic circuit and may be modeled as the band-limited Gaussian noise.”² In our updated Method Section on Artificially added noise we also outlined that the noise added in some cases was similar to Poisson noise.

1.2.2. What about the inter-sample variability or batch effect?

To test how *moleculaR* performs in inter-sample variability scenarios, we had looked into MPMs of three different lipids in two serial sections of human GB tissues in the **original** manuscript. These were perhaps a bit hidden as **Suppl. Fig. 5 and 7**.

To address the reviewer’s question, we **performed additional computation on three exemplary lipids and also include now an analysis and discussion of different normalization methods (TIC, RMS; new Suppl. Fig. 20, 22, 23) in the revised manuscript**. We also provide a new experiment as an additional example showing the proposed Gaussian mass-window weighting and the corresponding MPMs of an exemplary lipid on six mouse brain MSI replicates (**Suppl. Fig. 21**). The observed results indicated good agreement across serial sections and across intensity normalization methods, as indicated by the similar localizations for the lipids’ hotspot and coldspot contours. This is expected, since MPMs are designed to statistically model the characteristics of each measurement and sample individually.

1.2.3. Are intensities obtained from MPMs more reliable than the ones collected from POIs?

We are not entirely sure if we understand the reviewer’s question about “reliability” of intensities correctly. We understand it as follows: Are interferences that change or obscure ion intensities effectively removed, and do we get a statistically valid statement of intensities. In revision of this manuscript, we have **added many more experiments as examples and validity tests** to emphasize this understanding of “reliable intensities”:

The *moleculaR* framework presented here introduces three novel and complementary concepts to the MSI field:

- 1) Data-driven MOI-specific and FWHM-specific Gaussian mass-window down-weighting of interfering peaks for increased reliability of MOI intensities,**
- 2) Metabolite probability maps (MPMs) for statistically validated relative presence of a given MOI as indicated by hotspot and coldspot contour maps, and**

¹ Incl. Trede, D. et al. On the importance of mathematical methods for analysis of MALDI-imaging mass spectrometry data. *J Integr Bioinform* 9, 189, (2012); Alexandrov, T. et al. Spatial segmentation of imaging mass spectrometry data with edge-preserving image denoising and clustering. *J Proteome Res* 9, 6535-6546, (2010); Thacker, N. A. et al. The statistical properties of raw and preprocessed ToF mass spectra. *Int. J. of Mass Spectrometry* 428, 62-70, (2018); Deepaisarn, S. et al. Quantifying biological samples using Linear Poisson Independent Component Analysis for MALDI-ToF mass spectra. *Bioinformatics* 34, 1001-1008, (2018).

² Zhurov, K. O., Kozhinov, A. N., Fornelli, L. & Tsybin, Y. O. Distinguishing analyte from noise components in mass spectra of complex samples: where to cut the noise? *Anal Chem* 86, 3308-3316, doi:10.1021/ac403278t (2014).

3) Collective projection probability maps (CPPMs) for spatial visualization and analysis of user-defined metabolite ensembles.

In our view especially the first concept does indeed lead to more reliable intensities, whereas MPMs and CPPMs enable innovative displays and analyses of statistically validated probability contours and molecular ensembles, respectively, all based on more reliable intensities.

We hypothesize that the proposed Gaussian mass-window weighting provides more realistic MOI signal intensities, because i) it takes into consideration the mass resolving power of the mass spectrometer at m/z of MOI with a direct relation between σ_G of the Gaussian weighting envelop and the corresponding FWHM at m/z of MOI; and ii) because of the inherent characteristics of the Gaussian curve, which effectively helps in down-weighting proximal background signals (noise or other interfering analytes). To better illustrate this idea, we have provided a **new Fig. 1, which makes a stronger effort to explain this concept** in addition to now **Suppl. Fig. 1** (part of former Fig. 1). Very importantly, FWHM is a non-linear function of m/z (**new Suppl. Fig. 2**), and the function differs considerable for FTICR, timsTOF and TOF MSI platforms, a comparison of which we have now included in the revised manuscript (**new Suppl. Fig. 3**), in order to present *moleculaR* as a platform-independent tool.

In essence, for ion intensities, intensities for all detected peaks (i.e. POI plus whatever unrelated signals) in the user-defined (at least by defining parameters in a computer software) mass window, are summed up. In contrast, *moleculaR* for the first time provides a data-driven filter that adjusts filtering across MSI instrument platforms to settings in individual experiments like the actual resolving power (that for FTICR will depend on transient times and other parameters). **We argue that applying a Gaussian data-driven filter versus no filter makes intensities more reliable.** To demonstrate this, **we performed an additional extensive MSI experimental series in mouse sagittal brain sections that were in parallel imaged on a solarix FTICR-MSI, timsTOFflex orthogonal TOF-MSI and conventional rapiflex linear MALDI-TOF-MSI (Suppl. Fig. 5-7).** Our data suggests that *moleculaR* can help to resolve MOI ambiguities and that it may even be more useful in high resolution timsTOF-MSI than in ultra-high resolution MSI.

In addition, we now provide a new **Suppl. Fig. 4** to show the advantage of applying the proposed Gaussian mass-window weighting on MSI data that was artificially/computationally “contaminated” by spiked-in Gaussian noise in the vicinity of MOI m/z (**Suppl. Fig. 4AB**). These interfering noise signals were added at mass intervals within the mass resolving power of the mass spectrometer at MOI. The computed normalized sum of squared differences (SSD) between the raw MOI representation and the artificially/computationally “contaminated” one is greatly reduced compared to the uniform mass-window weighting, even when the spiked noise signals occur within the mass resolving window of MOI (**Suppl. Fig. 4B**). This is directly mirrored in the improved visual quality of the artificially/computationally “contaminated” MOI (**Suppl. Fig. 4C**). Based on the above, we postulate that the proposed Gaussian mass-window weighting can produce more reliable intensities compared to classical uniform mass window weighting when interfering signals are present at close proximity to MOI m/z .

On the other hand, another possible definition for “reliable” intensities could be thought of as intensities that are unlikely to occur as a consequence of a random spatial event. Here, *moleculaR* introduces MPMs. These maps represent a new MOI visualization method that adds statistical validation to the classical “ion images”, which lack it. MPMs provide statically validated relative spatial distribution of analyte intensities (shown as hotspot/coldspot contours superimposed on a rastered image). In other words, MPMs not only show the spatial distribution of analytes within a certain tissue,

but in addition they provide a user-independent and unbiased indication of where analytes' intensities have significant relative spatial abundance (or deficiency) that is unlikely to be seen if these signals were generated due to a random event (**Fig. 2A**).

1.3. Ground-truth

2 - An experiment that validates the molecular workflow. Ideally a ground-truth would be directly compared to the intensities provided by the MPM to quantitatively assess the method.

A calibration curve made of successive spots of internal standards (IS) of known concentration could potentially answer both of these points by providing a ground truth for spatial distribution and intensity to which both MPMs and POIs can be compared against. Also, replicating this experiment, would help investigate how the molecular processing can normalize for the inter- and intra-sample variability, which could potentially be presented as a key advantage of using SPP and MPMs over POIs. Finally, using IS representing different metabolite classes would also be very informative as their ionization properties differ, which could potentially be at least partially normalized by molecular.

To the best of our knowledge, the concept of ground-truth in MALDI MSI is lacking. This would require knowledge of true amounts or concentrations of analytes in the tissue – and all that in a spatially resolved manner. Defining and evaluating quantitative ground-truth in MSI would be a huge and very important area of research, as we and others have pointed out (e.g., Balluff et al (2021). Batch Effects in MALDI Mass Spectrometry Imaging. *J Am Soc Mass Spectrom* 32(3):628-635. doi: 10.1021/jasms.0c00393). But it is also beyond the scope of this study.

Nevertheless, we thank the reviewer for the suggestion to explore calibration curves in so-called quantitative MSI (qMSI) as a possible surrogate of a ground-truth: As is widely accepted in the field, qMSI refers to single or small numbers of MOI and requires stable-isotope labelled internal standards and calibration curves based on dilution series of target compound spotted on tissue or spiked into tissue homogenates (for example: Schulz S et al. Advanced MALDI mass spectrometry imaging in pharmaceutical research and drug development. *Curr Opin Biotechnol.* 2019; 55:51-59. doi: 10.1016/j.copbio.2018.08.003). Here, **we have re-analyzed the calibration data from Abu-Sammour et al, 2019** (Quantitative Mass Spectrometry Imaging Reveals Mutation Status-independent Lack of Imatinib in Liver Metastases of Gastrointestinal Stromal Tumors. *Sci Rep.* 9(1):10698. doi: 10.1038/s41598-019-47089-5), and **added a new Supp. Fig. 14** that suggests that calibration curves in qMSI may also benefit from the MPM concept: In this example, *molecular* was tested on an **imatinib drug dilution series measured on porcine liver with MALDI-FTICR-MSI, normalized to the deuterated internal standard imatinib-D8**. We compared linear and nonlinear calibration curve fitting on all detected signal intensities (in the respective measurement spots) of the drug imatinib versus the signal intensities obtained from the proposed Gaussian mass-window weighting which, importantly, were localized within a detected MPM hotspot. We observed that the fitted calibration curves exhibited higher linearity for the MPM case, as demonstrated by higher R^2 and x -exponent of the nonlinear fit even for curves that without it required non-linear fitting (**Suppl. Fig. 14**). Clearly, this initial encouraging finding needs to be further validated in subsequent in-depth studies.

To address the reviewer's rightfully raised concern, **we have also performed a substantial number of additional computational experiments that investigate the ground-truth topic using simulated data.** To this end, we generated artificial data that mimics a hypothetical analyte distribution according to a predefined ground-truth. Uniform Poisson SPPs of four different patterns of local hotspots were simulated (**New Fig. 2BC and Suppl. Fig. 8**). Their Intensity values were sampled from above and below the upper quartile of the empirical intensities of a MALDI-FTICR-MSI measurement of a human glioblastoma (GB) tissue sample (see revised **Methods**). *moleculaR* was able to reliably localize all simulated hotspots (green contours in the SPP plot **Fig. 2BC, Suppl. Fig. 8**) and identify points exhibiting significant relative spatial abundance (green points on density and surface plots in **Fig. 2BC, Suppl. Fig. 8**) with high overlap (as judged by high Dice similarity coefficient (DSC)) between the detected hotspots and ground-truth shapes (**Suppl. Fig. 10**).

1.4. FWHM estimation

On the technical side, I have a few questions/ comments:

- The relationship between FWHM and m/z is key to the moleculaR workflow but I could not find an explanation of why they are assumed to be in such close correlation. Is this an accepted relationship in the field? If yes, it should be mentioned for non-expert readers. If not, a clear explanation of why the chosen smoothing curve has been used to best to model this relationship should be added in the method section.

The relation between FWHM and m/z (or m) is known in the field, especially for FTICR MS, but it is currently not used in data analysis. Therefore, we thank the reviewer for asking us to strengthen this argument as one of the key innovative aspects of this study. In *moleculaR*, the Gaussian mass-window weighting is dependent on the estimated FWHM for a given MOI by the relation $\text{FWHM} = 2\sqrt{2\ln 2} \sigma_G$, where σ_G is the standard deviation of the MOI-specific Gaussian weighting envelope. On the other hand, FWHM is also directly related to the resolving power of the mass spectrometer, which is a characteristic of the instrument and the specific user-chosen measurement parameters. This is why we reasoned that for a reliable investigation of an MOI, the mass resolving power at m/z MOI should be (but currently isn't) an important consideration with a direct effect on the used mass-window and hence on the calculated signal intensities.

To better explain the point above, we have added New Fig.1, new Suppl. Fig. 2 and 3 and the "FWHM Model Fitting" section in Methods. Suppl. Fig. 3 also highlights the fact that the relationship between mass and FWHM is instrument-dependent.

1.5. KDE Gaussian bandwidth

- The bandwidth of the gaussian kernel of the KDE has been optimized using Moran's I. As I understand it, increasing the bandwidth does improve the signal-to-noise ratio at the cost of spatial resolution. How is this trade-off accounted for and how much loss of resolution can be afforded? Why has Moran's I been used as a metric and how is it relevant for defining such a crucial value? I do understand the

challenge of defining the optimal KDE parameters and Moran's I might be the best option, I would simply suggest the authors add their rationale on these points to the main text or supplementary if possible.

Again, we thank the reviewer for asking us to provide more details on rationale and practice of KDE used for *moleculaR*. Again, **we performed a substantial number of additional computational experiments, some using simulated data again, and added several new (suppl.) Figures.**

The bandwidth h_{KDE} for the chosen Gaussian kernel of the KDE was computed taking into account the previously reported inherent spatial autocorrelation of MSI measurements³ by finding the “knee” point in the Moran's I vs h_{KDE} plot, at which Moran's I levels-off, i.e. after which an increase in h_{KDE} does not result in a considerable increase in the spatial autocorrelation of the smoothed density image (**New Suppl. Fig. 9**). In other words, the smoothing bandwidth, at which Moran's I statistic rate of change abruptly falls, is taken as the unit length below which the MSI signals exhibit high spatial autocorrelation, thus effectively defining the scale of the spatial co-dependency of the signals. To further test this, **the KDE bandwidth estimation method was evaluated on the simulated data of New Fig. 2B and New Suppl. Fig. 8.** As illustrated in **New Suppl. Fig. 10**, for each of the simulated SPP types, an MPM was generated with the proposed KDE bandwidth estimation method described above, and the DSC (overlap) was computed between the calculated hotspot window and the ground-truth (red and green in **Suppl. Fig. 10**, respectively) based on which the simulated SPP was created. DSCs showed high degrees of overlap at 0.96, 0.89, 0.94 and 0.91 for all four local ground-truth/MPM hotspot patterns. Moreover, MPM was iteratively estimated for every simulated SPP type using assigned KDE Gaussian bandwidth h_{KDE} , which was varied from values of 0 to 10 length units in 0.1 incremental steps. At each iteration, DSC was computed between the estimated hotspot and the ground-truth shape used for generating the simulated SPP. **The resulting curves suggest that the KDE bandwidth estimation method of MPM coincided with the highest (i.e. the optimum) DSC for all simulated SPPs. This in itself was an unexpected finding that underlines the strength of this approach.**

1.6. Permutations of CSR

- Using spatially reshuffled intensity values of the SPP map to generate the CSR is very clever although I believe there are a few combinations to achieve CSR. How does the resulting MPM vary for different CSR generated from the same tissue section?

Since the internally generated MOI-specific CSR model is unique for every MPM evaluation, as rightfully mentioned by the reviewer, and since the resulting MPM directly depends on the generated CSR model, **we performed new computational experiments and tested MPM stability across many runs by repeating the same evaluation 100 times each with different CSR permutations.** We

³ Cassese, Alberto, et al. "Spatial autocorrelation in mass spectrometry imaging." *Analytical chemistry* 88.11 (2016): 5871-5878.

observed that the hotspot and coldspot areas relative to the total tissue area were stable across all iterations (mean and standard deviation of 0.3234 and 0.0045, respectively, for hotspots and 0.3771 and 0.0036, respectively, for coldspots) with high DSC overlap (mean and standard deviation of 0.9883 and 0.0069, respectively, for hotspots and 0.9914 and 0.0046, respectively, for coldspots) between the hotspot and coldspot areas of each experimental run relative to the hotspot and coldspot areas of the first iteration(**New Suppl. Fig. 19**).

1.7. Inter-sample variations

- Similarly, in 1F, using the WT tissue to generate the CSR is a very creative idea but it raises questions as it involves inter-sample and inter-individual variations that could potentially confound the observations made here. For example, would it be possible to evaluate how different sections of IDH-wt would impact the areas of significant Trp presence?

We thank the reviewer for appreciating this idea. The case of comparing signal intensities of a particular MOI in two tissues of different conditions (tumor vs normal, dosed vs control, etc.) is a typical one in the MSI field. In such cases, it is customary to compute comparative box/violin plots of bulk MOI intensities/concentrations (from tissue extracts in HPLC-MS or **pooled pixel intensities in MSI**), thereby **disregarding any, sometimes important, information on the spatial distribution of the MOI**. This is exemplified by the violin plot of the new panel in **Fig. 3E** (substantially improved over former **Fig. 1F**) which shows (i.e. the violin plot) a spatially-unaware comparison between [Trp-H]⁻ in IDH-mutant vs IDH-wild type glioblastoma (GB)⁴.

Here, molecularR has the added benefit of providing the spatial dimension to such comparisons by enabling spatial probabilistic cross-tissue comparisons. To help the reader to better understand this comparison, the violin plot was added into **Fig. 3E**, and a new **Suppl. Fig. 13** was added showing two additional comparisons of GB tissue sets².

However, in such experiments it is customary, and rightfully important, to approach this problem with an experimental design, which deliberately minimizes technical variation between the two samples by placing them on the same slide to be measured simultaneously. Even though such measures do not necessarily guarantee the absence of technical variability, they, together with standard intensity normalization methods (ex. TIC, RMS), produce reliable (enough) results and are considered to be the current norm.

For investigating this further, additional computational experimentation would be needed that involved multiple slides or measurements where robust intensity normalization becomes crucial. This, however, is beyond the scope of the proposed method, as it would require extensive studies of entire (patient) cohorts of samples, which were not available here.

⁴ Friedrich, Mirco, et al. "Tryptophan metabolism drives dynamic immunosuppressive myeloid states in IDH-mutant gliomas." *Nature Cancer* 2.7 (2021): 723-740.

1.8. Colocalization indicative of VT

- In 2B, the authors claim the Adenylate Energy Charge and Adenylate Kinase Mass Action Ratio scores are indicative of viable tumor (VT). I am not sure to understand how it is the case or how they came to that conclusion. Is it by colocalization evaluation? If anything, these scores indeed share some locations with the VT areas but are in now way indicative of them. I believe such conclusions should be either re-evaluated or better justified.

A rightful concern. We have tuned down the language in the manuscript.

2. Reviewer #2

Recommendation: Major Revisions

This work focuses on mass spectrometry imaging (MSI) as an important approach for studying metabolites in tissue. I agree with the authors when they point out data analysis of MSI measurements as an important area for development, and support their efforts in providing new software to accomplish this. I appreciated the manuscript's move away from traditional ion images, and the introduction of probabilistic mapping and spatial point patterns (SPPs) as novel approaches for MSI data. However, several important questions and issues would need to be resolved before publication can be considered.

We thank the reviewer for appreciating our study's "move away from traditional ion images, and the introduction of probabilistic mapping and spatial point patterns (SPPs) as novel approaches for MSI data".

The comments can be grouped as follows:

2.1. Secondary spots

- Comments on potential method artefacts and result reliability: The manuscript needs a more stringent (quantitative) assessment of whether the computational methods (and assumptions they make) introduce artefacts or remove potential biological information. Examples: Can kernel density estimation potentially reduce or eliminate secondary hotspots (that are smaller than the main hotspot) when the bandwidth estimation is optimized for the dominating hotspot and is applied tissue wide?

We agree with the reviewer that this is a rightful concern. In the MPM method workflow, an important step is the application of kernel density estimation (KDE) for both the spatial point pattern (SPP) of a

given MOI and its corresponding complete spatial randomness (CSR) model. This step is crucial, because it captures the overall spatial trend of the analyte's intensities. But perhaps more importantly (see **current Suppl. Fig. 11**, i.e., former **Suppl. Fig. 3**), this step conveniently causes the CSR intensity distribution f_{CSR} to converge towards a normal distribution, which then forms the basis for inferring the upper and lower intensity quantile cut-offs for hotspots and coldspots, respectively. To better illustrate this point, we edited original **Fig. 1B** to create current **Fig. 2A**).

Theoretical considerations:

Generally speaking, it is true that increasing the KDE bandwidth could have the effect of smoothing out signal intensities overall. However, it is important to state here that the mentioned smoothing does NOT occur before the MPM pipeline, and that it consequently does NOT affect the SPP representation/visualization. Everything is still shown. Smoothing occurs at the level of MPM computation, i.e. the level of spatial probability testing to highlight areas of "significant relative abundance/deficiency of MOI" that are indicated by the hotspot/coldspot contours. Once these contours are estimated, the original non-smoothed (but Gaussian-weighted) SPP intensities are carried on to the resulting MPM, and the hotspot/coldspot contours are superimposed. This essentially means that the visual aspect of the analyte's intensity distribution is not affected and that sparsely distributed signals including single-pixel signals are not altered or removed in the process. Therefore, the smoothing the reviewer refers to does NOT eliminate signals but is only an internal mechanism that directly affects estimated hotspots/coldspot areas.

For computation of hotspot/coldspot contours in a given MOI's MPM, importantly, the same KDE procedure and thus the same degree of smoothing is applied for both SPP_{MOI} and CSR_{MOI} , whose signal intensities are identical to the intensities of SPP_{MOI} at start. This causes the CSR_{MOI} intensity distribution to shrink with increasing KDE bandwidth, as observed in **Suppl. Fig. 11B** (former **Suppl. Fig. 3B**). Consequently, while signal intensities (of secondary as well as primary hotspots) are reduced with increased smoothing bandwidth, the quantile intensity cut-offs, which are used as thresholds for hotspots and coldspots, shrink towards the mean accordingly (**Suppl. Fig. 11B**).

Since smoothing SPPs (and CSRs) with high KDE bandwidths could result in underappreciating fine spatial structures or secondary MPM hotspots, this effect must be balanced against the desired effect of improved signal reliability. To this end, we have put forth a mechanism for KDE bandwidth estimation that prevents spatial over-smoothing based on the iterative evaluation of spatial autocorrelation as a function of KDE bandwidth (**new Suppl. Fig. 9**, which is related to former **Suppl. Fig. 2**, but **employs** an updated method for "knee" point detection). This is discussed in **comment 2.8** (as well as in **comment 1.5** of reviewer #1) in more detail. Importantly, this our updated method for "knee" point detection identifies bandwidth parameters that lead to optimal DSCs, as evidenced in **new Suppl. Fig. 10**.

Additional simulated data and wet lab experiments:

From an experimental perspective, to address this important concern in revision, we have developed a computational method for generating artificial data that mimics a hypothetical analyte distribution according to a predefined spatial ground-truth: Uniform Poisson SPPs of four different patterns of simulated ground-truth hotspots with varying complexity were simulated (**new Fig. 2BC** and **new Suppl. Fig. 8**). Their Intensity values were sampled from above and below the upper quartile of the empirical intensities of a MALDI-FTICR-MSI measurement of a human glioblastoma (GB) tissue sample at m/z 544.3009 (putative PE(20:1)+Na⁺; FDR \leq 0.2; see **revised Methods**). *moleculaR* was able to reliably localize all simulated hotspots (green contours in the SPP plot in **Fig. 2BC**, **Suppl. Fig. 8**) and to

identify points exhibiting significant relative spatial abundance (green points on density and surface plots in **Fig. 2BC, Suppl. Fig. 8**). It is important to note, that the **simulated SPP of Fig. 1C was specifically created to address the concern raised in this comment**. Here, the local hotspots are arranged into a central bigger circle of 20 length units radius and four adjacent smaller circles of 5 length units. Despite the big difference in ground-truth circle areas (smaller circles are 16 times smaller than the central one), *moleculaR* was able to correctly localize all four smaller local hotspots, as seen in the same figure. But **more importantly, we have developed an experiment for testing the effect of KDE bandwidth on the resulting hotspot relative to the respective ground-truth**. As illustrated in **New Suppl. Fig. 10**, for each of the simulated SPP types, an MPM was generated with the proposed KDE bandwidth estimation method (see **comments 1.5, 2.8** and **revised Methods**) and the DSC (overlap) was computed between the calculated hotspot window and the ground-truth (red and green in **Suppl. Fig. 10**, respectively) based on which the simulated SPP was created. DSCs showed high degree of overlap at 0.96, 0.89, 0.94 and 0.91 for all four local hotspot patterns, respectively. The resulting curves suggest that the KDE bandwidth estimation method of MPM coincided with the optimum DSC for all simulated SPPs, which in itself was not an expected finding.

In addition to the developed computational methods with simulated data above, **we have generated MPMs for additional biological use cases and tissue examples**. For instance, for the mouse brain MALDI FTICR MSI data depicted in the **new Fig. 3C** and the serial sections of the same tissue for N=6 technical replicates in **Suppl. Fig. 21**, even with the presence of bigger primary hotspots (brain stem area), MPM can still identify secondary hotspots much smaller in area relative to the primary ones (ex. Small circular at the base of the longitudinal fiber tract region).

We have also applied *moleculaR*'s MPM workflow to a previously published MALDI-TOF-MSI scenario of amyloid peptide signals in an APP NL-G-F Alzheimer's disease mouse model brain. This is an excellent example of sparsely distributed features and fine structures. We evaluated *moleculaR*'s MPM workflow's ability to detect $A\beta_{1-38}$ (m/z 4060.5) in so-called amyloid plaques and compared the calculated MPM hotspots to the results with the previously reported *Plaque Picker* method⁵. Out of the box, MPM hotspots (red/white contours of **new Fig. 3D**) correctly localized pockets of $A\beta_{1-38}$ plaques referenced by *Plaque Picker* (pixels highlighted in red in **new Fig. 3D**). One notable distinction was that *moleculaR* disregarded subsets of single pixels with high intensity assumed to be plaques by *Plaque Picker* which could be attributed to the fact that, unlike *moleculaR*, *Plaque Picker* strictly relies on image-global intensity thresholding irrespective of the spatial co-dependence of analyte signals which could be a useful criterion for filtering out spurious outlier single-pixel signals. After all, whether the single-pixel signals labelled as plaques by *Plaque Picker* but disregarded by *moleculaR* are in fact $A\beta_{1-38}$ plaques would have to be a subject for additional experimentation with orthogonal methods. Nevertheless, *moleculaR* allows the user to manually set the KDE bandwidth (or to rely on the "knee" point method), which could be either fine-tuned against an orthogonal (e.g. optical) method or simply inferred from what is theoretically expected for the object being imaged (if known, e.g. the minimum theoretical plaque diameter as shown in the **new Suppl. Fig. 12** for the data above).

The above has been acknowledged in various parts of the revised manuscript.

⁵ Enzlein, Thomas, et al. "Computational analysis of Alzheimer amyloid plaque composition in 2D-and elastically reconstructed 3D-MALDI MS images." *Analytical Chemistry* 92.21 (2020): 14484-14493.

2.2. Non-biological variations

The manuscript does not provide much discussion on whether raw intensity values are reporting biology only or still contain technical variation as well, and what the effect is when translating these values into MPMs and CPPMs. Since ion images commonly include non-biological variation and preprocessing is not perfect, what is the residual effect of non-biological variation in calculating probabilities for MPMs and CPPMs on the basis of those intensity values (do they reduce or amplify their effect).

2.2.1. The manuscript does not provide much discussion on whether raw intensity values are reporting biology only or still contain technical variation as well, and what the effect is when translating these values into MPMs and CPPMs

As in the case of ion images *molecularR* does its calculations based on ion/signal intensities generated by the MALDI ionization process, which invariably adds technical variability to biology-related variability. Hence, *molecularR* assumes the presence of both. Generally speaking, we do not present *molecularR* as a bona-fide MSI data denoising method, because it is not primarily designed to detect and filter out technical variability. However, we do claim that the proposed method will produce more “reliable” representations of the analytes’ distributions. For two reasons and on two levels both not primarily related to technical variability:

1. We hypothesize that the proposed Gaussian mass-window weighting provides more reliable MOI signal intensities because i) it takes into consideration the mass resolving power of the mass spectrometer with a direct relation between σ_G of the Gaussian weighting envelop and the corresponding FWHM at m/z MOI and ii) because of the inherent characteristic of the Gaussian curve, which effectively helps to down-weight proximal background signals (noise or other interfering analytes). To better illustrate this idea, we have provided a **new schematic Fig. 1, which makes a stronger effort to explain this concept** in addition to **Suppl. Fig. 1**. Moreover, new **Suppl. Fig. 5-7, which are based on an additional extensive MSI experimental series in mouse sagittal brain sections that were in parallel imaged on a solarix FTICR-MSI, timsTOFflex orthogonal TOF-MSI and conventional linear MALDI-TOF-MSI**, illustrate real life situations where *molecularR* can help to resolve MOI ambiguities. (These examples even suggest that *molecularR* may be even more useful in high resolution MSI than in ultra-high resolution MSI). Finally, we have provided a new **Suppl. Fig. 4** to show the advantage of applying the proposed Gaussian mass-window weighting on artificially/computationally “contaminated” MSI data spiked with interfering Gaussian noise at the vicinity of m/z MOI (**Suppl. Fig. 4AB**). These interfering noise signals are added to mass intervals within the resolving power of the mass spectrometer at MOI. The computed normalized sum of squared differences (SSD) between the raw MOI representation and artificially/computationally “contaminated” one is greatly reduced compared to the uniform mass-window weighting even when the spiked noise signals occur within the mass resolving window of MOI (**Suppl. Fig. 4B**), which is directly mirrored in the improved visual quality of the artificially/computationally “contaminated” MOI (**Suppl. Fig. 4C**). Based on the above, we can postulate that the proposed Gaussian mass-window weighting can produce more reliable

intensities compared to classical uniform mass window weighting when interfering signals are present at close proximity of m/z MOI.

2. On the other hand, another possible definition for “reliable” intensities would be intensities that are unlikely to occur due to a random spatial event (ex. noise). Here, *moleculaR* introduces MPMs. These represent a new MOI visualization method that adds statistical validation to the classical “ion images”, which lack it. MPMs provide statically validated *relative* spatial distribution of analyte intensities (shown as hotspot/coldspot contours superimposed on a rastered image). It is important to note that MPMs do not alter analyte intensities (calculated by the Gaussian mass-window weighting in point 1 above) in any way or form, they, however, provide a user-independent and unbiased indication of where analytes’ intensities have significant relative spatial abundance (or deficiency) that is unlikely to be seen if these signals were generated due to a random event (**Fig. 2A**). As mentioned above, in our additional mouse brain MALDI FTICR MSI experiment with N=6 technical replicates (=adjacent tissue sections) in **New Suppl. Fig. 21**, MPM hotspot contours were very comparable. All considerations for MPMs, in principle, also apply for CPPMs.

We have added a discussion of this topic to the manuscript.

2.2.2. Since ion images commonly include non-biological variation and preprocessing is not perfect, what is the residual effect of non-biological variation in calculating probabilities for MPMs and CPPMs on the basis of those intensity values (do they reduce or amplify their effect)

While the proposed Gaussian mass-window weighting might be able to tune down interfering peaks proximal to the m/z of MOI that may obstruct “classical” ion images, we cannot assume the absence of non-biological variations. Nevertheless, we hypothesize that because MPMs and CPPMs incorporate KDE at its core, it is expected that technical variations and noise fluctuations are conveniently reduced during the process of hotspot/coldspot estimation (see above). This could be clearly seen in the **improved Fig. 4** (former **Fig. 1C**) where MPMs were tested against different types of computationally added noise, as evidenced by Dice similarity coefficients (DSC) of 0.85, 0.97 and 0.98 for comparisons of MPMs based on raw data versus artificially/ computationally added Gaussian noise, intensity artefacts (isolated very high-intensity pixels) or interference peaks placed in the proximity of the MOI, respectively. Applying the same testing procedure to 142 MPMs of MOIs (FDR ≤ 0.2) in positive ion mode revealed median Dice similarity coefficients of 0.91, 0.98 and 0.98 for these three types of added noise, respectively (**improved Suppl. Fig. 16**; former **Suppl. Fig. 4**; the apparent difference between these Figures is due to the updated P-value correction method, Benjamini-Hochberg instead of Bonferroni, as described in **comment 1.2**). We also performed rigorous *moleculaR* vs noise testing by varying the standard deviation of the sampled noise (**new Suppl. Fig. 17 and 18**), for a detailed description please see **comment 2.9**. Based on these results, we can postulate that MPMs and CPPMs are relatively robust against different noise sources.

Additionally, to test how *moleculaR* performs in inter-sample and inter-measurement variability scenarios, we looked into MPMs of **three exemplary lipids** in two serial sections of human GB tissues (**new Suppl. Fig. 20**) while also looking at their corresponding MPMs before and after intensity normalization (**new Suppl. Fig. 22 and 23**). Perhaps most importantly, we also provided an additional

experiment showing MPMs of an exemplary lipid on six mouse brain replicates (**new Suppl. Fig. 21**). Despite considerable differences in TIC-normalized ion intensity images and Gaussian-weighted SPPs, the MPM hotspot contours displayed remarkable similarity and, hence, robustness against technical variability suggesting that the residual effect of non-biological variation was largely mitigated during the evaluation. Indeed, the observed results showed relative agreement across serial sections and intensity normalization methods indicated by the similar localizations for the lipids' hotspots and coldspot contours, which is expected since MPMs are designed to statistically model the characteristics of each measurement and sample individually. In other words, if we consider technical variation resulting from pixel-to-pixel, section-to-section and slide-to-slide batch effects as described in Balluff et al.⁶ then the computational experiments above indicate the *moleculaR* is relatively invariant for all three. However, it should be stated that separating biological and non-biological variability was not the focus of this study. Hence, the preliminary findings described above need to be further explored in follow-up studies.

The considerations above have been acknowledged in various parts of the revised manuscript.

2.3. Effect of arithmetic operations

and what is the additional effect of follow-up arithmetic operations using those potentially perturbed probabilities? Also, what is the potential for calculation anomalies (e.g. in ratio calculations)? Given the statistical significance claims made in the manuscript, a quantitative assessment of MPM and CPPM reliability seems appropriate.

2.3.1. What is the additional effect of follow-up arithmetic operations using those potentially perturbed probabilities?

It is important to note that MPMs (and CPPMs) do not alter analyte intensities (which are calculated by the proposed Gaussian mass-window weighting) in any way or form. This means that the pixel intensities seen on the resulting MPMs are in fact carried on from the original SPP intensities and do not hold probabilities (only the hotspot/coldspot contours are indicative of the probabilistic distribution of analytes). Arithmetic operations are carried out on the SPP intensities and only after such arithmetic operations are completed, MPMs or CPPMs are calculated. In other words, all follow-up arithmetic operations are evaluated upstream on SPP intensities, the result of which is then fed into the MPM pipeline to compute the probability contours. There are no arithmetic operations applied on probability contours. We have updated the corresponding section in **new Methods** to make it clearer for the reader.

2.3.2. What is the potential for calculation anomalies (e.g. in ratio calculations)? Given the statistical significance claims made in the manuscript, a quantitative assessment of MPM and CPPM reliability seems appropriate

⁶ Balluff, Benjamin, et al. "Batch effects in MALDI mass spectrometry imaging." *Journal of the American Society for Mass Spectrometry* 32.3 (2021): 628-635.

A valid concern. We direct the editor and reviewers to our answers for **comment 2.11** where the reviewer kindly made clearer points on this question.

2.4. Gaussian weighting

- Comments on method motivations, assumptions, and choices made: Several assumptions about the data and model/parameter choices are made with insufficient motivation for why these assumptions/choices are valid and with insufficient guidance for the reader on how to make these choices for their own MSI data. Example: Which evidence supports using a Gaussian shaped weighting scheme along the m/z axis for establishing an MOI intensity from multiple matching POI intensities?

This is also a valid concern. Why did we choose a Gaussian shape weighting scheme?

There are numerous examples in the MSI literature where Gaussians have been used (mostly for TOF mass analyzers), for example, to model peak shapes.^{7,8,9} The rationale behind using a Gaussian mass-window weighting was to incorporate the resolving power of the mass spectrometer at m/z MOI, which we have shown to be nonlinear (see new sections “FWHM Model Fitting” and “Gaussian Mass-Window Weighting” in **Methods** and **new Suppl. Fig. 3**), into the calculation of MOI intensities. By doing so we can make use of the convenient relations i) between the mass resolving power and the calculated peaks’ FWHM and second ii) between FWHM and σ_G of the weighting Gaussian envelop ($\text{FWHM} = 2\sqrt{2\ln 2} \sigma_G$), both of which enable data-driven calculation of the mass-window width taken as $m_{MOI} \pm 3\sigma_G$ (i.e. the span of the erected Gaussian-window) independent of the user, measurement device and - parameters (**Fig. 1**). At the same time, we did not want to introduce different mechanisms for FWHM-dependent weighting for different instruments, even though we understand that for FTICR data, for example, the peak shape is given by the Fourier transform of a sinusoidal decay curve and thus can be explained mathematically. Assuming no decay within the measurement time, it should be a $\text{sinc}(x)^2$.

To illustrate the desirable outcome of the proposed Gaussian mass-window weighting, we provide **new Suppl. Fig. 4**, which illustrates its advantage over the “classical” uniform mass-window weighting on artificially contaminated MSI data spiking Gaussian noise at the vicinity of m/z MOI (**Suppl. Fig. 4AB**).

Moreover, we performed additional extensive MSI experimental series in mouse sagittal brain sections that were in parallel imaged on a solarix FTICR-MSI, timsTOFflex orthogonal TOF-MSI and conventional linear MALDI-TOF-MSI (**new Suppl. Fig. 5-7**). Our data suggests that the proposed Gaussian mass-window weighting can help to resolve MOI ambiguities in a data-driven and user-unbiased way taking into account the mass resolving power at the analyte under study which can vary

⁷ Djambazova, Katerina V., et al. "Resolving the complexity of spatial lipidomics using MALDI TMS imaging mass spectrometry." *Analytical Chemistry* 92.19 (2020): 13290-13297.

⁸ Boskamp, Tobias, et al. "Using the chemical noise background in MALDI mass spectrometry imaging for mass alignment and calibration." *Analytical Chemistry* 92.1 (2019): 1301-1308.

⁹ Källback, Patrik, et al. "msiQuant—quantitation software for mass spectrometry imaging enabling fast access, visualization, and analysis of large data sets." *Analytical chemistry* 88.8 (2016): 4346-4353.

depending on m/z of the analyte, measurement parameters and the imaging platform being used. We refer editor and reviewers to detailed discussions of new **Suppl. Fig. 4-7** above.

The considerations above have been acknowledged in various parts of the revised manuscript.

2.5. Significant presence

- Comments on precision in wording, nomenclature, and mathematical definitions, and the need for additional clarity: The precision in wording, nomenclature, and mathematical definitions is not sufficient to avoid ambiguity. Examples: The manuscript seems to state it assesses statistically "significant presence" of an MOI. However, the comparison to the CSR checks for spatial autocorrelation and the original intensity values are e.g. changed/spatially smoothed using KDE. Significant spatial autocorrelation (even with an intensity component) is not the same as significant presence. Ions can be significantly present in tissue without concentrating spatially, while it seems the method presented here enriches only for the latter.

A valid concern. This is indeed an imprecise statement. We have changed the wording to “**significant metabolite relative spatial abundance/deficiency**” for analyte hotspots and coldspots, respectively.

2.6. Definition of Sigma

More precise wording would be helpful here. The manuscript is not very clear about its definition of sigma in the context of Gaussian distributions. At some locations sigma is implied to be the standard deviation. At other locations sigma is stated as being set equal to the FWHM of a peak, while the FWHM of a Gaussian shaped peak would be $>2\sigma$?

Detailed comments for each group are provided below.

A rightful concern. **We have revised the manuscript accordingly and unified the sigma definitions.** For instance, σ_G represents the standard deviation of the Gaussian mass-window weighting envelop, σ_{CSR} is the standard deviation of the intensities of the CSR model, σ_{noise} is the standard deviation of the added artificial noise and σ_{MOI} is the standard deviation of the signal intensities of m/z MOI. Moreover, we have made a clear statement for the relation between FWHM and σ_G as $FWHM = 2\sqrt{2\ln 2} \sigma_G$, while the span of the Gaussian envelop is taken as $m_{MOI} \pm 3\sigma_G$.

2.7. Demonstration of different data types

Finally, since the manuscript covers a software package meant for broader application and in order to assess whether the observations

in the manuscript are dataset-specific or method-specific, a demonstration on different MSI data types (e.g. non-FTICR) and on additional MSI case studies using different tissue types seems needed

To address this important comment and demonstrate broader application of our proposed solution for a life science as well as biomedical readership, we have **performed a substantial number of additional experiments on three different MSI platforms and examining additional use cases**. As a consequence, we tested *moleculaR* on several **additional** MSI data sets and accordingly now **provide several new (Suppl.) Figures**:

1. MALDI-FTICR-MSI data of mouse brain tissues (**Fig. 3C, Suppl. Fig. 5, Suppl. Fig. 15, Suppl. Fig. 21**).
2. MALDI-FTICR-MSI data of IDH-mutant and -wildtype human GB tissues (**Suppl. Fig. 13**).
3. MALDI-FTICT-quantification MSI data of porcine liver tissue (**Suppl. Fig. 14**).
4. MALDI-timsTOF-MSI data of mouse brain tissue (**Suppl. Fig. 6, Suppl. Fig. 15**).
5. MALDI-TOF-MSI data of mouse brain tissue (**Suppl. Fig. 7, Suppl. Fig. 15**).
6. MALDI-TOF-MSI data of an APP NL-G-F Alzheimer's disease mouse model (**Fig. 3D, Suppl. Fig. 12**).

2.8. Compromising smaller hotspots by KDE

Comments on potential method artifacts and result reliability

*- "a complete spatial randomness (CSR) model of that MOI is created by random spatial permutations of MOI points" and "the intensity distribution of the CSR density image is expected and observed to converge towards a normal distribution (Suppl. Fig. 3)" -> If the CSR is a spatially reshuffled version of the corresponding MOI, the same intensity values would be present in the CSR as in the MOI, but in different locations, so the histograms of intensities would be the same for both at this level. Then, the same KDE is applied to both the CSR and MOI, smoothing the intensity distributions using spatial location and the bandwidth parameter controlling the level of smoothing. The KDE effectively functions as a filter that eliminates/reduces hotspots (by smoothing them out) that are smaller than the bandwidth parameter lets through, in both the CSR and MOI cases. **Is it fair to eliminate smaller hotspots by smoothing them?** In the MOI case particularly, this seems to hold risk in the case of ion distributions that have several different sized hotspots. Only the larger most dominant hotspot (which ends up influencing the bandwidth parameter most) would seem to survive this process (without getting reduced in spatial size or being eliminated entirely). Please explain how this method would avoid eliminating or reducing smaller (but biological and therefore relevant) features in an MOI distribution. Since it would impact the biological distribution that ends up in the MPM, the manuscript needs a more exhaustive assessment of this. Suppl. Figure 3 shows, particularly*

for m/z 701.5592 and 666.4340, how as the bandwidth increases, a bimodal or trimodal distribution is smoothed into a unimodal form. The question remains whether it is fair to remove those spatially smaller pockets of ion intensity (or even reduce them in size) since spatial area alone cannot determine if an ion distribution is biological in nature or noise of some form, and so **it seems there is the possibility for biological patterns to get substantially modified or removed in the process.** Please provide guidance.

We thank the reviewer for reiterating this important aspect. We'd like to respond to this in two ways – a fundamental mathematical-technical and a biological-pragmatic way.

Biological-pragmatic consideration:

It is important to state, as also described in **Comment 2.1**, that the mentioned smoothing is not reflected on resulting intensities in visualized SPPs/MPMs, thus ensuring that sparsely distributed signals including single-pixel signals are not altered (after applying the Gaussian mass-window weighting) or removed. It is only at the level of MPM computation, i.e., the level of probability testing that single pixel signals tend to be disregarded as “significant metabolite relative spatial abundance/ of MOI”. Therefore, the KDE-based smoothing the reviewer refers to does not eliminate signals, but it serves as internal mechanism that directly affects the computed hotspot/coldspot probability contours for a user-unbiased evaluation of analytes' spatial distributions. It has several convenient characteristics, one of which is the reduction of the impact of technical noise fluctuations on these calculations, a direct result of this smoothing (see **Theoretical consideration** paragraph below). Another beneficial aspect of this internal smoothing comes from imposing an additional constraint on signals to be labelled as “significant abundance”; in addition to high intensity, they also have to have a certain degree of spatial correlation, which in many biological use cases will lead to a more conservative statement of spatial distributions. Such a rather conservative approach is actually desired in most use cases and rather serves to promote quality science. However, there are some use cases of MSI where rather sparse or single-pixel signals are biologically relevant, e.g., amyloid peptide deposits, so called plaques, in Alzheimer's brain.

The **new Fig. 3D** (Alzheimer plaques in mouse brain) therefore provides an important point in case: Several single pixel signals that are not filtered out in SPPs and that would be identified as plaque using the *plaque picker* method¹⁰ are not considered as MPM hotspots (i.e. not considered as unlikely to happen by chance only) by *moleculaR* with the data-estimated KDE bandwidth. This indicates that the default behavior of *moleculaR* tends to be more on the conservative side, especially when it comes to single-pixel signals. Nevertheless, as described in **Comment 2.1** and shown in **Suppl. Fig. 12**, molecular also permits the user to manually set the KDE bandwidth against a theoretically acceptable value which tends to reduce the spatial constraint described above. It must be noted, however, that this is one of the very rare cases, where a user has access to something like a ground-truth: single peaks per pixel that represent the amyloid peptide 1-38, which are absent in most pixels of the mutant mouse brains and completely absent in brains of control mice. In most biomedical use cases of MSI today, on the other hand, such a ground-truth is missing and current MSI users have no way of knowing whether single pixel signals have biological meaning or not or if they may be sparsely distributed outliers. There

¹⁰ Enzlein, Thomas, et al. "Computational analysis of Alzheimer amyloid plaque composition in 2D-and elastically reconstructed 3D-MALDI MS images." *Analytical Chemistry* 92.21 (2020): 14484-14493.

is no valid single-pixel MS/MS on tissue and co-registration of adjacent sections (with immunohistochemistry) is typically not precise enough, immunohistochemistry on the same section after MSI is often too fuzzy. Despite considerable recent success to use a lot of sophistication to utilize laser ablation marks for precise mapping of single cells¹¹, this is currently challenging in routine MSI on tissue.

Theoretical consideration:

As discussed in **Comment 2.1 and 2.2**, the application of KDE possess three main advantages: i) it captures the overall spatial trend of the analytes' intensities, ii) it forces f_{CSR} to converge to a normal distribution (**Suppl. Fig. 11**; former **Suppl. Fig. 3**) and, being a low-pass filter as indicated by the reviewer, iii) it has the often-desired outcome of smoothing technical variations and noise fluctuations during the process of hotspot/coldspot estimation which has in turn a positive outcome on the method's tolerance to pixel-to-pixel batch effects (**Fig. 4, Suppl. Fig. 16, Suppl. Fig. 17, Suppl. Fig. 18**), and section-to-section/slide-to-slide batch effects¹² (**Suppl. Fig. 20-23**).

However, as pointed out by the reviewer in this comment and as discussed in **Comment 2.1**, smoothing SPPs with high KDE bandwidths while generating MPMs has the potential adverse effect of overlooking fine, small or sparse spatial structures as small secondary (probability) hotspots.

To address this issue, we have put forth a mechanism for KDE bandwidth estimation based on spatial autocorrelation that prevents spatial over-smoothing. To elaborate on this, the KDE bandwidth h_{KDE} is computed taking into account the previously reported inherent spatial autocorrelation of MSI measurements¹³ by finding the "knee" point in the Moran's I vs h_{KDE} plot, at which Moran's I levels-off, i.e. after which an increase in h_{KDE} does not result in a considerable increase in the spatial autocorrelation of the smoothed density image (**New Suppl. Fig. 9**; former **Suppl. Fig. 2**; difference is due to an updated method for "knee" point detection). In other words, the smoothing bandwidth, at which Moran's I statistic rate of change abruptly falls, is taken as the unit length below which the MSI signals exhibit high spatial autocorrelation, thus effectively defining the scale of the spatial co-dependency of the signals. To further test this, the **KDE bandwidth estimation method was evaluated on the simulated data shown in Fig. 2B and New Suppl. Fig. 8** and discussed in **Comment 2.1**. As illustrated in new **Suppl. Fig. 10**, for each of the simulated SPP types, an MPM was generated with the proposed KDE bandwidth estimation method as described above and the DSC (overlap) was computed between the calculated hotspot window and the ground-truth (red and green in **Suppl. Fig. 10**, respectively) based on which the simulated SPP was created. DSCs showed high degree of overlap at 0.96, 0.89, 0.94 and 0.91 for all four local hotspot patterns, respectively. More importantly, MPM was iteratively estimated for every simulated SPP type using assigned KDE Gaussian bandwidth h_{KDE} , which was varied from values of 0 to 10 length units in 0.1 incremental steps. At each iteration DSC was computed between the estimated hotspot and the ground-truth shape used for generating the simulated SPP. The resulting curves demonstrated that the KDE bandwidth estimation method of MPM coincided with the highest (i.e. the optimum) DSC for all simulated SPPs. Here, we draw the reviewer's attention to the simulation case shown in **Suppl. Fig. 10C**, which precisely mimics the situation of primary and secondary hotspots as raised in this comment.

¹¹ Rappez, Luca, et al. "SpaceM reveals metabolic states of single cells." *Nature methods* 18.7 (2021): 799-805.

¹² Balluff, Benjamin, et al. "Batch effects in MALDI mass spectrometry imaging." *Journal of the American Society for Mass Spectrometry* 32.3 (2021): 628-635.

¹³ Cassese, Alberto, et al. "Spatial autocorrelation in mass spectrometry imaging." *Analytical chemistry* 88.11 (2016): 5871-5878.

In addition to the anti-over-smoothing mechanism described above, we would like to bring to the reviewer's attention, as pointed out in **Comment 2.1**, that the same degree of smoothing is applied for both SPP_{MOI} and CSR_{MOI} retaining their comparability and causing CSR_{MOI} intensity distribution f_{CSR} to shrink with increasing KDE bandwidth as observed in **Suppl. Fig. 11B** (former **Suppl. Fig. 3B** mentioned by the reviewer) affecting thereby the quantile thresholds of the hotspots and coldspots. Here, for m/z 701.5592 and 666.4340, even when one of the observed maxima in f_{MOI} is smoothed out (ex. local maximum at 0.0005 intensity), the corresponding f_{CSR} shrinks, thus reducing the quantile threshold for declaring an intensity as part of the hotspot. This means that while pixel intensities (of secondary as well as primary hotspots) are reduced with increased smoothing bandwidth, the quantile intensity cutoffs, which are used as thresholds for hotspots and coldspots, shrink towards the mean accordingly (**Suppl. Fig. 11B**).

In addition to the above, in **Comment 2.1** we have also provided evidence based on real data exemplified by the mouse brain MALDI FTICR MSI data depicted in the new **Fig. 3C** and the six serial sections of the same tissue in **Suppl. Fig. 21**. This data shows that even with the presence of bigger primary hotspots (brain stem area), MPM can still identify secondary hotspots much smaller in area relative to the primary ones (ex. small circular at the base of the longitudinal fibre tract region). In **Comment 2.1** and in the **Biological Consideration** above, we have also discussed the example of A β ₁₋₃₈ amyloid plaques detection in a MALDI-TOF-MSI measurement of an APP NL-G-F Alzheimer's disease mouse model and compared *molecularR*'s performance against the previously published *Plaque Picker* method.

To this end, while the point raised by the reviewer is an important one, based on the evidence presented (simulated and real data), we think that i) with the proposed method of KDE bandwidth estimation, ii) the methodology of applying comparative KDE smoothing on both SPP_{MOI} CSR_{MOI} and iii) giving the user the ability to set the KDE bandwidth, possibly based on a theoretically expected value (see **Comment 2.1**), the method is able to avoid the overlooking of smaller secondary hotspots. All of that being said, it is essential to reiterate that MPMs do not affect pixel intensities in any way or form, they rather compute the hotspot/coldspot contours which are then superimposed on the pixel intensities computed by the Gaussian mass-window weighting. This essentially means that MPMs do not *reduce* or *eliminate* pixel intensities seen in the final MPM.

2.9. Artificial noise

- (Figure 1) "are robust against noise and common signal artifacts"
 -> This statement seems somewhat broad. There are several types of noise, commonly present in MSI data, against which this method would not be robust. The manuscript demonstrates adding Gaussian noise to ion intensities, which is not a very common noise type in mass spectrometry data (e.g. Poisson noise is more common), particularly in FTICR data where the intensity range being reported is large. Additionally, robustness to noise is usually a matter of how strong the noise patterns are. The demonstration seems to apply Gaussian noise using a single variance value. Please show robustness for several Gaussian distribution widths so it can be assessed at what level of noise the method's robustness breaks down. The same

argument holds for the spiked artifacts and interfering peaks cases, in that those probably also have parameters in function of which robustness should be reported. Please provide a narrow and precise definition of what type of noise the method is robust against.

2.9.1. *This statement seems somewhat broad. There are several types of noise, commonly present in MSI data, against which this method would not be robust. As discussed in **Comments 1.2** and **2.2.2**, *moleculaR* has been shown to be relatively robust against several types of artificially added noise (**Fig. 4**, **Suppl. Fig. 16-18**) as well as against section-to-section and slide-to-slide batch effects (**Suppl. Fig. 20-23**). The reviewer did not provide examples of types of noise that *moleculaR* may not be robust against (except for mentioning Poisson noise in **Comment 2.9.2** below). We nevertheless chose to tune-down the above statement in the manuscript; “MPMs are robust against various forms of artificially added noise and signal artefacts”.*

2.9.2. *The manuscript demonstrates adding Gaussian noise to ion intensities, which is not a very common noise type in mass spectrometry data (e.g. Poisson noise is more common), particularly in FTICR data where the intensity range being reported is large.*

A rightful concern. To better illustrate this idea we have provided an updated **Fig. 4**, where the performance of MPMs is tested against different types of computationally added Gaussian noise, intensity artefacts (isolated very high-intensity pixels) and interference peaks placed in the proximity of the MOI. Moreover, to address the comment raised here, we have added a new **Suppl. Fig. 17**, which provides more insight into how MPMs are affected by Gaussian noise of increasing variance. Since the intensity range for MALDI-FTICR-MSI is normally large, as rightfully pointed by the reviewer, a Poisson noise distribution could be approximated as a special case of the Gaussian distribution where $\lambda_{\text{Poisson}} = \mu_{\text{MOI}} \gg 1000$. We added the following text to the methods: MPMs were tested against a Gaussian noise source with varying $\sigma_{\text{noise}} = \sigma_k$ ($k = 0 \dots 10$), where $\sigma_0 = \sqrt{\mu_{\text{MOI}}}$ and $\sigma_k = k \sigma_{\text{MOI}}$ for $k = 1 \dots 10$. Note that for $k = 0$, the resulting noise is similar to Poisson noise with $\lambda_{\text{Poisson}} = \mu_{\text{MOI}} \gg 1000$.

2.9.3. *Additionally, robustness to noise is usually a matter of how strong the noise patterns are. The demonstration seems to apply Gaussian noise using a single variance value. Please show robustness for several Gaussian distribution widths so it can be assessed at what level of noise the method's robustness breaks down.*

To address this rightfully raised concern, we performed *moleculaR* vs noise testing by varying the standard deviation of the sampled Gaussian noise starting with a rather low noise dispersion resembling that of a Poisson noise (**Comment 2.9.2**) all the way up to 10 times the standard deviation of the raw MOI signal. We found that *moleculaR* is able to withstand Gaussian noise sources up to 4

times the standard deviation of the raw MOI signal retaining hotspot overlap of above 0.75 DSC, all this while observing noticeable visual image degradation (**Suppl. Fig. 17C**).

2.9.4. *The same argument holds for the spiked artifacts and interfering peaks cases, in that those probably also have parameters in function of which robustness should be reported. Please provide a narrow and precise definition of what type of noise the method is robust against*

A precise description of the noise types which *moleculaR* was tested against was lacking in the original submission. To better clarify this concept, we have added **Fig. 4AB** and added the section “**Artificially Added Noise**” in **Methods**. For the case of interfering peaks, as in the case of Gaussian noise, noise intensities are sampled from a Gaussian distribution (but spiked at m/z MOI + $2\sigma_G$) and is therefore expected to have a milder impact on the resulting MPMs as the Gaussian mass-window weighting provides excellent protection against this type of noise (see new **Suppl. Fig. 18**).

2.10. Test vs control tissues

- "enabling true spatial probabilistic cross-tissue comparisons" -> Figure 1F is given as an example of how comparisons can be made between two tissue samples. It is not entirely clear what the reader should take away from the panel. The statement "Significant Trp presence" seems imprecise. The statistical test locates areas where there is more than random spatial clustering it seems. Whether the amount or intensity is significant is hard to assess since the intensities (that the probabilities are based on) get changed quite a bit in the process (e.g. are the intensities directly comparable to begin with? -> see preprocessing question; the intensities can furthermore be weighted by Gaussian weights and potentially transformed with a square root transformation it seems, etc.). The statement "Significant Trp presence" seems too broad to capture that nuance. Please provide a more precise comparison example of what there can be learned from panel 1F. How should the MPM contour be utilized in the reader's datasets. Please provide some guidance on what this allows in terms of comparison between tissues and what it does not allow.

We thank the reviewer for appreciating this idea. To start with, as suggested by the reviewer in **Comment 2.5**, we have replaced the statement of “significant analyte presence/absence” with “significant relative spatial abundance/deficiency” throughout the manuscript. We think that this is a more appropriate description.

The use case of comparing signal intensities of a particular analyte in two tissues representing different conditions (tumor vs normal, dosed vs control, etc.) is frequently seen in the MSI literature. In such cases, it is customary to do comparative box/violin plots (with or without statistical inference) of bulk MOI intensities/concentrations (e.g. based on tissue extracts in HPLC-MS or **pooled pixel intensities**

in MSI) disregarding any, sometimes important, information on the spatial distribution of the MOI. This is exemplified by the violin plot of the new panel in **Fig. 3E** (improved over former **Fig. 1F**) which (i.e. the violin plot) shows a spatially-*unaware* comparison between [Trp-H]⁻ in IDH-mutant vs IDH-wild type glioblastoma (GB)¹⁴. Here, *moleculaR* has the added benefit of providing the spatial dimension to such comparisons by enabling spatial probabilistic cross-tissue comparisons. To help the reader to better understand this comparison, the violin plot was added into **Fig. 3E**, and a new **Suppl. Fig. 12** was added showing two additional comparisons of GB tissue sets⁸.

It is important to note that in such experiments it is customary, and rightfully important, to approach this problem with an experimental design, which deliberately minimizes technical variation between the two samples by placing them on the same slide to be measured in a single measurement. Even though such measures do not necessarily guarantee the absence of technical variability, they, together with standard intensity normalization methods (ex. TIC, RMS), generally produce reliable results and are considered to be the norm. To this end, all GB tissue sets of **Fig. 3E** and **Suppl. Fig. 12**, were placed on the same slide, measured within the same MSI run and captured using the proposed Gaussian mass-window weighting (see **Comment 2.2.1** and **2.4** for a discussion on its reliability) with no square-root transformation. Therefore, we conclude that [Trp-H]⁻ intensities, which formed the basis for the comparison, were indeed comparable.

For situations where tissues intended for such comparisons originate from different slides or MSI measurements, the same caution should be practiced as when comparisons are made with classical box/violin plots. Here, robust intensity normalization would become crucial, and additional computational experimentation would be needed involving multiple slides or measurements to further explore this. This, however, is beyond the scope of this manuscript and the proposed method, as it would require extensive studies of entire (patient) cohorts of samples, which were not available here.

2.11. Arithmetic operations and uncertainty.

2.11.1 "permits spatial evaluation of composite numeric scores obtained by applying basic arithmetic operations on spatial point patterns of multiple MOIs" and "adenylate kinase mass action ratio" -> Although it is technically possible to use the MOI values for calculation, the real question is how much uncertainty they are accompanied with and how well they represent actual biological abundance. While this is already the case for the raw intensity values present in standard ion images, the authors should address whether their subsequent processing and transforming steps to get to MPMs and CPPMs do not add additional uncertainty on top of the biological and technical uncertainty present in the raw measurements. Please provide a quantitative assessment.

¹⁴ Friedrich, Mirco, et al. "Tryptophan metabolism drives dynamic immunosuppressive myeloid states in IDH-mutant gliomas." *Nature Cancer* 2.7 (2021): 723-740.

It is widely accepted in the MSI field and also stated by the reviewer that ion intensities in MSI are accompanied by much uncertainty, which require sophisticated on-tissue or tissue-mimetic model-based calibration curves and the use of SIL internal standards for quantitative MSI (Schulz et al., 2019, *Curr. Op. Biotechnol.* and many others). In our recent review¹⁵, a thorough discussion was presented on the fact that biological information is concealed by technical variation arising from various sources of the MALDI-MSI process and several suggestions has been proposed to limit such adverse effects. Hence, this is a valid concern which we think is really a question to the entire community of MSI.

This challenge, however, has not stopped the use of ion images as well as their ratios and derived composite scores as an investigative tool in MSI in highly regarded journals. For instance intensity ratio-based MS images of key metabolites were used to study esophageal cancer human tissue samples¹⁶ while MSI-based spatial distribution of energy charge index has been previously shown on patient-derived xenografts of IDH-mutant and -wild type gliomas¹⁷.

In comparison with this state-of-the-art in excellent journals, we have argued above that *moleculaR*, while not designed as a denoising tool per se, can be expected to reduce variability by reducing interferences for MOIs by Gaussian weighting and by KDE smoothing (at the expense perhaps of disregarding some single-pixel signals as insignificant). Also in arithmetic operations this should lead to a reduced propagated error, because of reduced input error. According to the laws of error propagation (linear, Gaussian, other) arithmetic operations may always add (relative or absolute) uncertainty. However, we expect them to have less impact on the resulting image intensities of MPMs (because of the Gaussian mass-window weighting; **comments 2.2.1** and **2.4**) and on the hotspot/coldspot contours (because of the internal KDE smoothing; **comment 2.2.2**) when compared to state-of-the-art ion image procedures.

A quantitative assessment of this effect, however, would require orthogonal measurements on an entire tissue cohort and ideally a ground-truth scenario, which is beyond the scope of this study. This is especially true, since not even the nature of error propagation is clear for MSI. Is it linear or perhaps Gaussian? More research is needed in this arena. Nevertheless, a word of caution has been added to the revised Methods (last paragraph of "**Spatial Arithmetic Expressions**" section).

2.11.2 Furthermore, please address issues with taking a ratio, which is used in the examples of the manuscript and is somewhat susceptible to introducing unforeseen calculation artifacts when uncertainty is not controlled for. For example, zero or low values in the denominator can lead to respectively NaNs or exaggerated values in the calculated ratio result. The manuscript states that NaNs are removed, but that does not address

¹⁵ Balluff, Benjamin, et al. "Batch effects in MALDI mass spectrometry imaging." *Journal of the American Society for Mass Spectrometry* 32.3 (2021): 628-635.

¹⁶ Sun, Chenglong, et al. "Spatially resolved metabolomics to discover tumor-associated metabolic alterations." *Proceedings of the National Academy of Sciences* 116.1 (2019): 52-57.

¹⁷ Fack, Fred, et al. "Altered metabolic landscape in IDH-mutant gliomas affects phospholipid, energy, and oxidative stress pathways." *EMBO molecular medicine* 9.12 (2017): 1681-1695.

the underlying issue. Please address how unintended effects of direct calculation with intensity values can be discovered and countered.

Since *molecularR* relies only on centroided data (except for a single full profile spectrum used for FWHM estimation), the SPP intensities, which were used in ratio computations (and everything else), are peak intensities (with SNR ≥ 3) and could be assumed to be devoid of (detector) baseline intensities (“low values”). For any arithmetic operation to take place, SPPs are converted to rastered images with dimensions identical to the original MSI raster (as described in revised **Methods**) where the absence of a peak signal in a certain pixel is encoded as a zero intensity. During pixel-wise ratio calculations the only anomalies that could arise are dividing an empty pixel by an empty pixel (i.e. $0/0 = \text{NaN}$) and dividing a peak signal by an empty pixel (i.e. $\text{peak}/0 = \text{Inf}$). For the arithmetic operations defined in *molecularR*, the two situations above are considered as invalid, and all NaN or Inf pixels are dropped (by simply passing these through *is.finite* command in R) when converting back to SPP representation for further analysis using the MPM pipeline. In other words, zero values in the denominator are dealt with computationally while low values (baseline or no signal) in the denominator are not expected due the fact that *molecularR* works mainly with centroided data. The description of these operations has been made clearer in the revised **Methods**.

2.12. FWHM estimation

- Were the results of the FWHM estimation (on top of Friedman's super smoother) assessed for quality? If so, how does this translate to other MSI datasets?

A valid concern, we acknowledge that the issue of FWHM estimation and model fitting was not clearly described in the original manuscript, and we thank the reviewer for pointing this out. To address this, we have provided additional theoretical insight into how FWHM varies as a function of mass for MALDI-FTICR, -timsTOF and -TOF mass spectrometers (see new section “**FWHM Model Fitting**” in revised **Methods**) and provided the **new Suppl. Fig. 3**, which nicely correlates the empirical data of this study with the theoretical equations governing this relationship. Moreover, we have dropped the use of Friedman’s super smoother (see original manuscript) in favor of locally estimated scatterplot smoothing (LOESS) for fitting a smoothing curve into the FWHM vs m/z data points. The latter showed excellent agreement with the expected theoretical response (green and light red curves of **Suppl. Fig. 3A**, respectively). To test this further, we have included a **new Suppl. Fig. 2** which compares the fitted LOESS curve for FWHM vs m/z data points obtained from a single (cyan in **Suppl. Fig. 2AB**) vs 100 randomly chosen spectra (light red) for both positive and negative ion mode of the glioblastoma tissue samples presented in the manuscript. Evidently, using a single randomly chosen spectrum could be representative of the FWHM vs m/z axis relationship for the entire measurement (see **Comment 2.23**). We have also compared the fitted curves on the same glioblastoma tissues for both ionization modes and observed high similarity for all four cases (**Suppl. Fig. 2C**). Finally, we have compared the FWHM estimates based on the fitted LOESS model proposed in this study against the ones calculated by the mass spectrometry vendor software (DataAnalysis version 5.3; Bruker Daltonics) of six random peaks of the single spectra taken from the measurements mentioned above to find high agreement between the two with differences being in the fourth decimal digit (**Suppl. Fig. 2D**).

2.13. Effect of non-biological variations

- "For any number of MOIs, basic arithmetic operations on the spatial point patterns of MOIs could also be applied. This is useful when a ratio of two MOIs is desired (ex. Fig. 2B bottom row) or when a more complex evaluation is of interest (ex. Fig. 2B top right and bottom right)." -> The direct use of the intensity values in arithmetic operations implies that the intensity values only report tissue content and do not report technical variation or noise. Is this the case? Is noise effectively removed leading up to this point in the process? How does the square root transformation affect these arithmetic operations? Please explain why this is valid, and what preprocessing procedures were applied to this data such that technical noise has been sufficiently removed to make direct calculation reliable.

2.13.1 The direct use of the intensity values in arithmetic operations implies that the intensity values only report tissue content and do not report technical variation or noise. Is this the case? Is noise effectively removed leading up to this point in the process? Please explain why this is valid, and what preprocessing procedures were applied to this data such that technical noise has been sufficiently removed to make direct calculation reliable.

Please refer to **Comment 2.11** for a more complete discussion on this matter.

2.13.2 How does the square root transformation affect these arithmetic operations?

Please refer to **Comments 2.46** for a thorough discussion on this matter.

2.14. Division by zero or small values

- "Possible divisions by zero are computationally dropped." -> Since divide by zero leads to NaNs, their removal seems evident. However, dividing by very small (nonzero) numbers has a similar exaggerating effect (very large numbers that overrepresent the true values, despite not being NaNs). How are these handled by the software? How do the users know their end result of the arithmetic operation they applied does not have this instability?

We believe that this part has been thoroughly answered in **Comment 2.11.2**.

2.15. Artificial noise and Dice coefficient

- (Suppl. Fig. 4) The comparison using the Dice coefficient shows a substantial number of MOIs for which the DS drops below 0.75. This suggests that the biological distribution of the MOI gets substantially changed by the method presented. Is this not an issue? How should the reader assess whether the MPM they get at the end is reliable? In this comparison, known noise sources were added to the original data and so there is something to compare against. How should this reliability be assessed when the true noise in the data is not known, which would be the common situation for most MSI measured MOIs. Please elaborate.

A rightfully raised concern. During the revision of the manuscript we observed that the previously used strict Bonferroni P-value correction (within the MPM pipeline; **Methods**) had a negative effect on the stability of the computed hotspots/coldspots. Switching to the Benjamini-Hochberg P-value correction, which is frequently used in proteomics, had a positive impact on the stability and reproducibility of the method, as demonstrated by higher median DSCs of the **new Suppl. Fig. 16** compared to the **former Suppl. Fig. 4**. We have therefore updated the method and used the Benjamini-Hochberg P-value correction approach throughout the revised manuscript.

Since noise is artificially and randomly added to the MSI data, we expect that the distribution of biologically relevant MOI signals does not change, but that MOIs rather get concealed by the added noise. The degree by which the raw signals are affected by the added noise depends on their raw intensity and spatial distributions; MOIs that have sparse spatial distribution with low dynamic range of intensities are expected to be affected more by the added noise than MOIs that have strongly clustered spatial distribution and high dynamic range. However, it is important to note that the artificially added Gaussian noise discussed in the manuscript is rather extreme as rightfully indicated by the reviewer in **Comment 2.9.2**. This could be seen in the **new Suppl. Fig. 17** and **18** where the standard deviation for the Gaussian noise was chosen to be equal to the standard deviation of the raw intensities and therefore has a higher impact on the calculated DSCs compared to Poisson noise.

Regarding the reliability of MPMs and the impact of noise on them, we believe that this was thoroughly discussed in **Comment 2.2.2**.

2.16. Reason of using SPP representation

Comments on method motivations, assumptions, and choices made:

- The authors make the case for approaching a MSI dataset as a spatial point pattern (SPP) and doing molecular spatial probability mapping on that SPP. However, since the regular grid of locations in an ion image is implicitly an SPP (with every pixel representing a point, and with potentially low intensity values still included), what is the major reason MSI analysis should move to SPPs? For example, MOIs, CPPMs, and hotspot contours could also be calculated directly on ion images it seems. Please explain more precisely why SPPs are needed for the process and why they are beneficial to the MSI user.

Generally speaking, the difference between SPP and raster image representations could be described as the difference between vector-based and raster-based graphics; they are equivalent representations of the same spatial distribution, but each provides a unique set of tools and methods that are tailored to the specific nature of each representation. While transforming raster-based MSI data into SPP representations may not seem to add much (visual) benefit, SPP representation enables new types of operations with data, e.g., analysis of localized events and advanced statistical and homogeneity analysis of spatial data. More specifically, as described in the manuscript, the MPM (and CPPM) pipeline relies on a direct comparison between the SPPs of MOI and the corresponding CSR model. While MOI SPPs inherit the gridded spatial locations from MSI data, creating a CSR model on a spatial grid would directly violate the randomness criterion of such models. Therefore, in order to facilitate the direct and homogeneous comparison between MOI and a corresponding CSR model, MOI spatial intensities must be converted into an SPP representation. In other words, this representation is necessary to carry on with the MPM evaluation. Having said that, we could still confidently propose that MPMs, but not necessarily SPPs, shall replace or complement ion images for the spatial analysis of MOI because of its valuable benefit of enabling localization of significant relative spatial abundance/deficiency of MOI.

2.17. Gaussian weighting and non-biological variations

- "arithmetic operations within the same MS image" -> The method seems to assume that the peak intensity of POIs reflects the abundance of a species directly. Since this is not always the case and preprocessing is often not perfect, using the weighted peak height for consecutive calculation carries the potential for amplifying the non-biological variation in the intensity distribution. Please assess this quantitatively and acknowledge it in the manuscript.

We believe that this part has been thoroughly answered in **Comments 2.2, 2.3, 2.4 and 2.11. We also acknowledged this possibility in the manuscript.**

2.18. Gaussian weighting and MOI

- "use the sum of ion intensities of all peaks present in a user-defined mass range centered on the POI m/z instead (Suppl. Figure 1A)." -> The manuscript refers to a user-defined mass range for integrating the ion intensity around a specific m/z value. However, most peak picking algorithms, also elsewhere in mass spectrometry and spectroscopy, automatically determine a (m/z) window around each found peak and provide it for integration. The argument for human bias being introduced by manually specifying a mass window around a peak therefore does not seem to fit here, but specifying it around an MOI's m/z taken from a database might fit? However, in that case, the question becomes why the problem is approached from the perspective of the MOI (for which a window needs to be specified) instead of the measured POIs (for which the windows can be measured from the full profile spectra directly). Please clarify what the exact type of data is that is being started from and/or why

there is a MOI-centric approach being taken, so that it's clear why there are multiple separately detected POIs that need to be summed together in a weighted fashion.

First, let us clarify why an MOI-centric approach is taken. We see two use cases: 1. An MOI-centric approach introduces or focuses on a different perspective, namely the perspective of a non-MS-proficient user, e.g., in the medical or life science fields. *moleculaR* permits this non-MS-proficient user to analyze and visualize ensembles of MOI that he/she is interested in. Therefore, we call it an "MOI-centric approach". 2. *moleculaR* complements existing POI annotation engines for mass spectrometrists by adding an analysis/visualization tool for statistically validated relative spatial abundances of POIs (externally, e.g., via Metaspace) matched to certain MOIs across tissue morphologies.

Therefore, the key aspect here is NOT that windows around POIs may not be a good choice, whereas windows for MOIs are. Instead, a focus on MOI versus POI accomplish different things, and the Gaussian down-weighting could also be applied to POI if so desired. Exploring use cases for the latter is, however, not the focus of this study.

Regarding the type of data, a distinction should be made between peak intensity calculations in full profile MS(I) data compared to the uniform mass-window weighting, which is used to render ion images. For the former, typically when peak detection is performed on a mass spectrum and a peak is detected, the intensity is assigned either as the maximum intensity of the peak profile or area under the curve of that peak profile. This is only applied on full profile MS(I) data. This, however, is different from the uniform mass-window weighting, which could be performed on full profile as well as centroided MSI data, to generate ion images of analytes. In the latter case, the user normally defines the extent of the mass-window where (peak or profile) signals are to be integrated either as an absolute value in Daltons or a relative value in ppm (relative to m/z MOI). This could be seen in all major software platforms dedicated to analysis and visualization of MSI data (ex. FlexImaging, SCiLS LAB; Bruker Daltonics) as well bioinformatics packages (ex. Cardinal, MALDIquant).

As in the case of the major MSI data analysis platforms, *moleculaR* uses an MOI-centric approach for the visualization of analytes, i.e., a given molecule of interest (drug, metabolite, etc), and addresses the question of what the underlying ion distribution as seen by MALDI MSI is. This approach is beneficial for three reasons: i) it is computationally efficient, ii) it is robust against slight inter-pixel peak misalignments (which are normally observed in MSI) and iii) it works well with spatially sparse peaks (the absence of the MOI signal in a given pixel does not cause any computational trouble). Most importantly, this approach is the only approach which is suitable for the storage-efficient centroided (peaks only) MSI data. Because *moleculaR* works mainly with this kind of data (centroided), the peak profile information is not available during evaluation. Instead *moleculaR* relies on a single, user-provided, randomly chosen full profile spectrum based on which FWHM data is estimated for the entire m/z axes and used for the determination of the extent of the mass-window as described in **Comment 2.4**. The approach described above is, therefore, different from the workflow: "given a detected peak, what does this peak represent i.e. which molecular entity is that?" which is a different question that has been addressed elsewhere.^{18,19} The respective statement has been updated in the

¹⁸ Palmer, Andrew, et al. "FDR-controlled metabolite annotation for high-resolution imaging mass spectrometry." *Nature methods* 14.1 (2017): 57-60.

¹⁹ <https://metaspace2020.eu/>

revised manuscript to clearly indicate what sort of data *molecularR* deals with (**Methods**; “MALDI Mass Spectrometry Imaging” and “Data Preprocessing” sections).

2.19. MOI-centric approach and double counting

- "we introduce an MOI-centric biomedical perspective that systematically analyzes and visualizes if biologically relevant MOIs have a statistically validated spatial presence across tissue morphologies." -> There are often going to be more MOIs present in databases to match to than there are POIs present in the measured dataset. This seems to imply that approaching the problem from an MOI perspective could lead to more false positive matches to POIs than when trying to match a POI to potential MOIs. Please elaborate on why an MOI-centric approach is better. Also, we know for sure a POI was present and detected in the dataset. We do not know if a particular MOI is present in the dataset. How does the method address this one-to-many mapping (in terms of overlap and double-counting of POIs, and in terms of computational scalability) and how does it address the uncertainty of such a POI to MOI assignment?

In no way are we claiming that “an MOI-centric approach is better”. It is rather complementary, as it introduces or focuses on a different perspective, namely the perspective of a non-MS-proficient user, e.g., in the medical or life science fields. Therefore, it is important to clarify that *molecularR* does not provide a framework for testing if POI=MOI. It rather permits this non-MS-proficient user to analyse and visualize ensembles of MOI that he/she is interested in. Therefore, we call it an “MOI-centric approach”. In addition, *molecularR* complements existing POI annotation engines for mass spectrometrists by adding an analysis/visualization tool for statistically validated relative spatial abundances of POIs (externally, e.g., via Metaspaces) matched to certain MOIs across tissue morphologies. For MPMs, the rationale for using an MOI-centric approach is outlined in **Comment 2.18** above. Whether the intensity distribution observed is, in fact, the desired MOI (drug, metabolite, etc.) is a question that must be validated by orthogonal analytical approaches such as MS/MS or tims-MS. This is equally true for existing complementary POI-to-MOI pipelines. Nevertheless, we acknowledge that the expression “MOI-centric” could be misleading for the reader. We have therefore clarified this in the manuscript.

Although, we do agree with the reviewer that there are more database MOIs to match to than there are POIs present in the measured dataset, mapping MOIs into POIs does not create more false positives than mapping POIs into MOIs, as for a given FDR the database usually reports a group of candidate molecules for a single POI all of which could be the detected POI. This is a known problem and is currently limited by the measurement technology capabilities (mass resolution, on-tissue MS/MS, on-tissue ion mobility MS). The problem of double counting is a rightful concern. Here, and especially for CPPMs, *molecularR* provides simple tools to filter out duplicated counts of the same *m/z* value by incorporating only the intensities of unique masses present in the computed CPPM.

2.20. Spatial autocorrelation and biological relevance

- "we propose molecular probability maps (MPMs) for rigorous user-independent spatial statistical testing that are based on the assumption that for any given MOI spatial autocorrelations exist that mirror the biological interaction between neighboring tissue morphologies" -> Please explain why it's fair to assume that any MOI's spatial distribution demonstrates spatial autocorrelation. There are examples of biologically relevant molecular species that do not show strong spatial autocorrelation, yet are important. Does this approach not work for these species? Are they filtered out as not statistically significant? Should the user remove such ion distributions beforehand? Please specify in the manuscript.

We thank the reviewer of pointing out this statement, which appears to be imprecise. The intended assumption is that the observed increase in spatial autocorrelation (systematic spatial variations) of a given observed POI in a tissue section could be a surrogate for defining similar biological states or an underlying tissue morphology linked directly or indirectly to this POI. We have updated this statement in the revised text.

2.21. POI-MOI matching

- "are based on the assumption that for any given MOI spatial autocorrelations exist that mirror the biological interaction between neighboring tissue morphologies" -> An additional implicit assumption seems to be that an MOI is present in the data and that there's no uncertainty related to the POI-MOI match. What happens if the MOI is not present in the data, but a set of peaks close to that MOI's theoretical mass happen to show spatial autocorrelation? Does that count as a significant match for the statistical test? Please elaborate.

In general, *moleculaR* provides two mechanisms to avoid scenarios such as the one described above:

1. It uses Gaussian mass-window weighting to down-weight other interfering signals that could be present in the vicinity of m/z MOI.
2. It provides a statistically validated spatial testing on the detected signals and filters out spatially random signals by considering them of non-significant spatial relative abundance/deficiency.

In each of the steps above, there is of course some degree of uncertainty. If a false positive POI-MOI match (through an external method) may occur for an interfering peak that is observed in a close proximity of the theoretical m/z MOI which also happen to have a degree of spatial autocorrelation, then *moleculaR* will have no other way of protecting against such cases and will provide an MPM that shows hotspots/coldspots for that particular POI. It is important to note, however, that ion images, which is still the current norm in studying spatial distribution of analytes in MSI, will also fail in this case. Unlike *moleculaR*, ion images, which are the benchmark against which *moleculaR* is compared, do not possess any of the points mentioned above.

2.22. Gaussian weighting

- (Suppl. Figure 1) In panel 1b, a Gaussian weight profile is multiplied with individual peaks. How does the method know it is dealing with a single Gaussian peak and not a mixture? If this is an assumption, what is the potential effect of this assumption when it does not fit reality. One example is shown in panel b with quite a high peak to the right of the Gaussian profile getting severely down-weighted. This might be correct to do, but the the manuscript should explain why that is the case, and how we can tell that that's the right thing to do.

To better illustrate the workflow of Gaussian mass-window weighting, we have provided a new **Fig. 1**, which makes a stronger effort to explain this concept in addition to **Suppl. Fig. 1**. Since *molecularR* is designed to work with centroided data (see **Comment 2.66**), the vertical bars shown in the schematic **Fig. 1** hypothetically indicate (separate) peaks detected at that mass interval. The new **Fig. 1** also does a better job in explaining the nature of the observed peaks differentiating three different kinds; a theoretical m/z MOI (vertical dashed black bar), an observed m/z POI (solid vertical blue bar) and an interfering peak (solid vertical orange bar). In this case, the severe down-weighting of the interference is a desired outcome. **Comments 2.2** and **2.4** provided evidence on the effectiveness of this scheme of signal weighting.

2.23. FWHM estimation full spectrum

- (Suppl. Figure 1) "A user-selected full mass spectrum is used to compute full width half maximum (fwhm) values of its peaks across the m/z axis." -> How should the user select a mass spectrum to accomplish this? Also, would that not require that the chosen spectrum is representative for the entire dataset, which can contain tissue subareas that are molecularly distinct from the location where the reference mass spectrum is pulled? Please elaborate.

We thank the reviewer for pointing this out. What we meant to say is this: At least one full profile spectrum has to be made available for FWHM estimation, whereas the rest of the analysis in *molecularR* uses all centroided data. The location that the chosen full profile spectrum is taken from is irrelevant, as is the identities of the peaks present in it. FWHM estimation builds a model based on the FWHM values of its peaks, which would later represent FWHM at any theoretical peak. This is because it is assumed (and observed; **Suppl. Fig. 2AB**) that the dataset is measured with the same measurement parameters, and therefore the mass resolution as a function of m/z should be similar across the tissue. The only requirement is that the user doesn't select a corrupt spectrum (a spectrum that only contains noise), which should be clear. For more information we kindly ask the reviewer to refer to **Comment 2.12**. This has been acknowledged in the revised manuscript.

2.24. FWHM curve fitting

- (Suppl. Figure 1) "A curve is fitted to describe the continuous relation of fwhm as a function of m/z ." -> Please elaborate on why it is fair to assume that there is a linear relationship between

FWHM and the m/z value? In complex mixtures, this seems like it would be a hard argument to make.

We acknowledge that the issue of FWHM estimation and model fitting was not clearly described in the original manuscript and we thank the reviewer of pointing this out. Indeed, the relationship is not linear, and we consider it an achievement of the *moleculaR* framework to take this nonlinear relationship into account for Gaussian weighting. To clarify, this we have provided additional theoretical insight into how FWHM varies as a function of mass for MALDI-FTICR, -timsTOF and -TOF mass spectrometers (see new sections “FWHM Model Fitting” and “Gaussian Mass-Window weighting” in revised **Methods**) and provided the new **Suppl. Fig. 2** and **3**, which nicely correlate the empirical data of this study with the theoretical equations governing this relationship. As it turns out, estimated FWHM vs m/z axis are observed to be nonlinear. Please refer to **Comment 2.12** for more details.

2.25. What if MOI is not present?

- (Suppl. Figure 1) "A Gaussian envelope, whose sigma is inferred from the fwhm model at MOI, is centered on MOI. The Gaussian is then used as a weighting factor to protect against proximal background signals; the further the measured m/z (= POI) from the theoretical m/z (= MOI), the lower the weight it receives in the SPP representation." -> It seems like the method assumes that the MOI is present. What happens if the match is incorrect and the MOI is not present? Are the POIs still summed (in a Gaussian weighted way) and made to represent the MOI? Please explain in the manuscript how the uncertainty of the POI-MOI match is taken into account.

We believe that this concern has been addressed in **Comment 2.21**. This was now now phrased more carefully in the manuscript.

2.26. SPP representation

- (Suppl. Figure 1) "An exemplary comparison between standard ion image and SPP representation of [Heme+K]+." -> Please explain what the fundamental difference is between representing ion presence as a regular grid (where absence of a molecular species is depicted as a zero value) and a spatial point pattern (where absence of a molecular species is depicted as an absent data point). Furthermore, since both approaches can represent absence, the manuscript should make clear what the advantages and disadvantages are of representing data either way.

We believe that this question has been thoroughly discussed in **Comment 2.16**. This was now made clear in the manuscript.

2.27. FWHM curve fitting

- (Figure 1) "curve-fitted to describe FWHM as a function of m/z " -> Please explain why a linear relationship is an appropriate assumption. Also, in panel a, please provide the actual m/z values along the m/z axis.

We believe that this question has been thoroughly discussed in **Comments 2.12, 2.23 and 2.24**. New dedicated Figures (**Fig. 1, Suppl. Fig. 2 and 3**) and method section "**FWHM Model Fitting**" and "**Gaussian Mass-Window weighting**" have been provided in the revised manuscript. Since the new **Fig. 1** is a schematic figure, there are no real m/z values to be shown. However, these has been provided in the corresponding Supplementary figures incl. former Fig. 1 that was moved to the supplement.

2.28. KDE choice of kernel

- (Figure 1) In panel b, kernel density estimation is referred to, which comes with choices to be made (e.g. the choice of kernel used). Please explain why the choice for a (isotropic) Gaussian kernel is the right one here? What are the implications if this choice is not right? How should a user that wants to employ this method on her own data choose? Please acknowledge method choices in the manuscript and explain the reasons for the choices made here.

Anisotropic Gaussian-kernel density estimation is widely adopted in the scientific community when there are no assumptions or preference regarding the spatial pattern to be smoothed. Within the MSI field, studies involving unsupervised clustering of MSI data also relied internally on spatial weighting using Gaussian kernels^{20,21}. *moleculaR* does not assume any specific spatial "direction" of the analytes under study and, therefore, an isotropic Gaussian kernel is an appropriate choice in this case. **This was now made clear in the manuscript.**

2.29. Artificial noise overlapping peaks

- (Figure 1) "added overlapping peaks 2 sigma away" -> Why specifically 2 sigma? Why is that the right location to put an overlapping peak? Would 1 sigma not be a more overlapping peak? Please explain the parameter choices, and motivate why these are the right values to evaluate at.

Here, the interfering noise was put **arbitrarily** at m/z $MOI + 2\sigma_G$. We have also provided an additional **Suppl. Fig. 4** which shows the effect of an artificially added Gaussian noise on the uniform and Gaussian mass-window weighting spiked at increasing m/z locations measured in multiples of σ_G away

²⁰ Alexandrov, Theodore, and Jan Hendrik Kobarg. "Efficient spatial segmentation of large imaging mass spectrometry datasets with spatially aware clustering." *Bioinformatics* 27.13 (2011): i230-i238.

²¹ Bemis, Kyle D., et al. "Probabilistic segmentation of mass spectrometry (MS) images helps select important ions and characterize confidence in the resulting segments." *Molecular & Cellular Proteomics* 15.5 (2016): 1761-1772.

from m/z MOI starting from m/z MOI + $0(\sigma_G)$ to MOI + $3(\sigma_G)$. **This was now made clear in the Suppl. Fig. 4 caption.**

2.30. POI-MOI matching

- "(MALDI) MSI raw data of any given MOI m/z is first transformed into an SPP representation" -> The formulation of this sentence implies that the raw data measured in the MSI experiment "belongs" to the MOI. However, the link between the raw data and the MOI is based on a numerical matching of m/z values (with an uncertainty window and an additional user-defined Gaussian weighting), there is no definitive link between the two until traditional mass spectral identification has confirmed the link. How does the method take the tentative nature of the POI-MOI link into account? What happens if the match turns out to be incorrect? Is that a responsibility for the user? Please elaborate and have the tentative nature of this link reflected in the manuscript text.

We believe that this question has been thoroughly discussed in **Comments 2.19 and 2.21**. In any case will POI-MOI links in MSI be tentative. They require the use of orthogonal analytical techniques and measures such as MS/MS fragmentation (at least in accompanying LC-MS/MS experiments), CCS values in ion mobility and/or accurate mass and sum formula determination at very high resolving power. **This was now made clear in the manuscript.**

2.31. Smaller hotspots

- "which then forms the basis for inferring intensity cutoffs beyond which the intensities of MOI's density image are unlikely to occur if generated by a random spatial process (see Online Methods)" -> Since the KDE has a smoothing effect that can filter out or reduce potential biological hotspots smaller than the largest hotspot, is this determination of cutoffs fair? Please motivate.

We believe that this question has been thoroughly discussed in **Comments 2.1 and 2.8** and acknowledged in the revised manuscript.

2.32. Test vs control tissues

- "statistically-validated significant localization of immunosuppression-associated tryptophan in IDH-mutant- compared to IDH wild-type glioblastoma" -> Since the CSR model of the test tissue is inferred from signal intensities of the reference tissue, it implies that the signal intensities between the mutant and wild type measurements were directly comparable. How was this accomplished? How was the preprocessing done to ensure that the

intensities between the two samples were on the same scale, without the potential for skewing e.g. by the number of pixels in each sample or e.g. the substantial amount of different tissue structure between mutant and wild type? Please explain this aspect and acknowledge in the manuscript.

We believe that this question has been addressed in **Comments 2.10**, and appropriate recommendations have been added to the manuscript.

2.33. Registration

- "Cropped images of both modalities were resampled" -> What type of resampling was used? Please report the type of resampling used, the parameters involved, and motivate why these choices are appropriate in this case.

Resampling of spatial resolution was done by a bilinear interpolation. Pixel spacing was adapted according to the provided values. A motivation for the choice of the new pixel size can be found in **Comment 2.34**.

2.34. Registration

- "with a resolution of 7.5 μm^2 per pixel" -> Assuming isotropic resampling, do the authors mean that the resampled pixel describes a square of 2.739 μm ($=\sqrt{7.5}$) by 2.739 μm ? Why this choice? Regardless of the specific pixel area resampled to, why this target value? How should a reader decide what area to resample to?

2.34.1 Assuming isotropic resampling, do the authors mean that the resampled pixel describes a square of 2.739 μm ($=\sqrt{7.5}$) by 2.739 μm ?

We thank the reviewer for pointing this out. It is actually a pixel size of $(7.5 \mu\text{m})^2 \Rightarrow 7.5\mu\text{m} * 7.5\mu\text{m}$. This was corrected.

2.34.2 Why this choice? Regardless of the specific pixel area resampled to, why this target value?

The focus in this multimodal registration procedure was the larger morphological structures of the tissue sections, which can be seen in both the optical and H&E stained images. Choosing the aforementioned parameters, helped orienting the registration procedure on the morphological structures in the H&E.

2.34.3 Why this choice? Regardless of the specific pixel area resampled to, why this target value?

Choosing the right parameters depends on the experimental setup. The resolution for resampling should be chosen so that the larger morphological structures are clearly visible in the fixed and moving images.

2.35. Registration

- "down-sampling factors of 4,2,1" and "Advanced Mattes Mutual Information metric" and "250 iterations"-> Please provide motivation for these parameter choices, and guidance for the reader to make their own choices for their datasets.

High factors in the pyramidal approach are used for coarse alignment. Choosing successively lower factors helps capture finer details. If the overall alignment fails, higher factors can be added (creating a pyramid with additional levels), or the existing factors can be changed.

Advanced Mattes Mutual Information (MI) is a standard metric for multimodal image registration problems. If optical images and adjacent images are from the same imaging modality, monomodal image metrics, e.g. the Mean of Squared Errors or Normalized Cross-Correlation, can be used.

In this study, 250 iterations for each pyramidal resolution for the rigid registration was successful. For more robustness this could be increased up to values of 1000-2000. **Guidance is now provided in the manuscript.**

2.36. Registration

- "interpolation using third-order B-Splines" and "750 iterations" -> Same comment as above.

B-Splines of 3rd order are commonly chosen for deformable registration problems. In this study, 750 iterations for each pyramidal resolution for the deformable registration was successful. For other studies and data, it may be necessary to increase it up to values of 1000-2000.

- "manually defined control points were added at corresponding locations in both modalities to support the deformable registration" -> How many locations? Does this manual step influence any analysis results later on? If not, has this been quantitatively verified?

Additional points were successively added at locations where structures were not sufficiently aligned. The number of points is related to the severity of the morphological distortion. The transferred annotations (vital tumor only) are used to highlight the co-localization between vital tumor regions (as annotated by an expert pathologist) and identified hotspot/coldspot areas of MPMs/CPPMs (new **Fig. 3AB, Fig. 5BCD**). Since the analysis results are for demonstration purposes only, a qualitative visual evaluation was considered sufficient.

2.37. Registration

- *"Matt's Mutual Information metric and the Corresponding Points Euclidean Distance metric (equally weighted)" -> Why is equal weighting the right mixture? Why was the Euclidean Distance metric added, while the previous step only used the other metric? Was there an issue with the mutual information metric that the Euclidean Distance metric solves? Please explain.*

2.37.1 Why was the Euclidean Distance metric added, while the previous step only used the other metric? Was there an issue with the mutual information metric that the Euclidean Distance metric solves?

The preparation of adjacent sections may introduce morphological deformations during cutting, slide placement, washing and staining. In some cases, it was necessary to force the registration to a good fit by using pairwise corresponding point annotations in the fixed and moving images. In other words, if either of these registration steps did not work automatically, corresponding point annotation pairs were manually added to the fixed and moving images. These point annotations were included in the rigid and deformable registration steps. Failures were identified by qualitative assessment. The word "deformable" in the respective sentence was misleading and was therefore removed. We have updated this paragraph in the **Methods** section.

2.37.2 Why is equal weighting the right mixture?

Equal weighting yielded good results. Since the point annotations were carefully placed, we wanted a strong match of those corresponding point annotations. Other ratios can be selected for looser alignment of corresponding point annotations and higher Mattes Mutual Information influence.

2.38. Registration

- *"to control registration parameters" -> Were these set automatically, or manually using the M2aia application? Please explain.*

We thank the reviewer for pointing this out. The M²aia²² (RRID: SCR_019324; <https://www.github.com/jtfcordes/m2aia>) desktop application was used to view registration results and to interactively define pairwise corresponding control points within both image modalities. Ultimately, the default settings provided by M²aia were used. We have updated the respective paragraph in the **Methods** section.

²² Cordes, Jonas, et al. "M2aia—Interactive, fast, and memory-efficient analysis of 2D and 3D multi-modal mass spectrometry imaging data." *GigaScience* 10.7 (2021): giab049.

2.39. Peaks filtering

- "Peaks that occurred in less than 1% of the pixels were filtered out" -> Please explain why 1%, and not for example 5%? How should the reader make this assessment for their own data?

Such peak filtering is a norm in MALDI MSI experiments and is usually done to limit the presence of spurious random peaks occurring in an MSI experiment.^{23,24} The choice of the percentage threshold is rather a rule of thumb. We have chosen 1% to be more on the conservative side.

2.40. Binning

- "after being binned to a relative tolerance" -> Please elaborate on the specifics of this procedure. Are the bins 12 ppm wide? Are they sequential or can they overlap? How are the bin centers decided?

Peak binning is an essential preprocessing step in MALDI MSI data analysis. It is usually done to reduce the effect of the slight inter-pixel peak misalignments which are normally observed in this kind of data due to interpolation errors during mass calibration. As described in **Methods**, this binning is not part of *molecularR*, it is rather handled by an R package called *MALDIquant*²⁵ and has already been applied in many leading studies²⁶. Briefly, the observed peak masses of the entire MSI data (all peaks of all pixels) are grouped and sorted into a single vector and the difference between each neighboring pair is computed. Then a series of iterative bisecting is applied on the mass vector, each time at the largest difference, until all peaks within each bin fulfill the criterion $|peak_{ij} - \mu_j|/\mu_j < tolerance$; where $peak_{ij}$ the mass of the i -th peak at the j -th bin, μ_j is the mean mass of all peaks present in the j -th bin (bin center; the new peak position) and *tolerance* is the maximal relative peak deviation ($\Delta m/m$) of peak positions to be considered as identical which, for this study, was set to 12 ppm ($= \Delta m/m$; $\Delta m = FWHM$ at m/z 400 ≈ 0.0048 Da; $m = m/z$ 400). Since the focus of this study was on the lipidome, m/z 400 was chosen as it the typical starting mass range at which lipids signals are observed. For more information on this procedure please refer to the published study of Gibb et al.²⁷ and the accompanying package documentation (*binPeaks* command)²⁸. We have updated the statement on peak binning in the revised manuscript (Methods; **Data Preprocessing** section).

²³ Ly, Alice, et al. "High-mass-resolution MALDI mass spectrometry imaging of metabolites from formalin-fixed paraffin-embedded tissue." *Nature protocols* 11.8 (2016): 1428-1443.

²⁴ Janda, Moritz, et al. "Determination of abundant metabolite matrix adducts illuminates the dark metabolome of MALDI-mass spectrometry imaging datasets." *Analytical chemistry* 93.24 (2021): 8399-8407.

²⁵ Gibb, Sebastian, and Korbinian Strimmer. "MALDIquant: a versatile R package for the analysis of mass spectrometry data." *Bioinformatics* 28.17 (2012): 2270-2271.

²⁶ Nachtigall, Fabiane M., et al. "Detection of SARS-CoV-2 in nasal swabs using MALDI-MS." *Nature biotechnology* 38.10 (2020): 1168-1173.

²⁷ Gibb, Sebastian, and Korbinian Strimmer. "MALDIquant: a versatile R package for the analysis of mass spectrometry data." *Bioinformatics* 28.17 (2012): 2270-2271.

²⁸ <https://cran.r-project.org/web/packages/MALDIquant/index.html>

2.41. Binning

- "a relative tolerance of 12 ppm (i.e. the maximal relative deviation of a peak position to be considered as identical)" -> Please elaborate on why 12 ppm is an acceptable range for this mass range.

Please refer to **Comment 2.40** above.

2.42. Gaussian weighting

- "effective support ($mMOI \pm 3\sigma$)" -> Why is the Gaussian profile the appropriate weight profile to use? Why not another profile that weighs down peaks that are farther away? Was the influence of this choice assessed on the results?

We believe that this question has been thoroughly discussed in **Comments 2.4**.

2.43. MOI list

- "Given a set of MOIs" -> How is this list determined? Is it provided by the researchers? Provided by Metascape? Please elaborate. What happens if MOIs overlap in terms of which POIs they connect to? How is the overlap handled if there's no one-to-one match?

The main intended use case for *moleculaR* is indeed that ensembles of MOI be provided by researchers. In order to test and validate *moleculaR* with as many sensible MOIs as possible for a single glioblastoma specimen, however, in this study, the MOI list was obtained from the annotations provided by Metaspace against the SwissLipids database. The respective statement has been updated in the revised text (**Methods; Collective Projections Probability Map** section). Regarding the MOI-to-POI mismatch and overlap, please refer to **Comments 19** and **21**, where this has been thoroughly discussed.

2.44. Coordinate duplication

- "allowing for coordinate duplication" -> Please explain the coordinate duplication referred to here. Is maybe overlap meant?

Unlike raster images where the number of pixels is limited by their dimensions, SPPs do not put any restriction on the number or location of points to be part of the point pattern. Therefore, a single SPP can hold any number of points coming from any number of MOIs. Points sharing the same coordinate location (ex. originating from the same x,y -coordinate location on the MSI raster) can co-exist without the need to sum them up, as, for example, the case of raster images of fixed dimensions. The

respective statement has been updated in the revised text (**Methods; Collective Projections Probability Map** section).

2.45. Double counting

- *"results in accumulation of the corresponding signal intensities"*
-> Are the MOIs guaranteed to be non-overlapping in terms of which POIs they connect to (even using lower weights of the Gaussian distribution)? If not, why is it fair to add the SPPs of the MOIs together since that could mean certain POIs are counted multiple times? Please explain in the manuscript.

This question has already been thoroughly discussed in **Comments 2.19** and **2.21**. Moreover, the window used for Gaussian mass-window weighting in *molecularR* is inferred from the FWHM at m/z MOI, which is directly related to the resolving power of the mass spectrometer. When two neighboring MOIs share observed POIs, there is no way to tell if a certain POI is part of one MOI's Gaussian envelope or the others. This is because "sharing" POIs would essentially mean that these two MOIs are not resolvable (either partially or completely). Therefore, scenarios such as the one described above, hit the limit of the mass resolving power at m/z MOI. Nevertheless, as described in **Comments 2.2.1** and **2.4** and shown in **Suppl. Fig. 5** and **6** of two example neighboring MOIs, the Gaussian mass-window weighting proves to be effective in resolving even overlapping MOIs provided that MOIs are at least partially resolvable. In this sense, POI are not counted multiple times, but they may be used multiple times, as is the case in current uniform intensity weighting. The respective statement has been updated in the revised text (**Methods; Collective Projections Probability Map** section).

2.46. Sqrt transform and Heteroscedasticity

- *"The workflow then commences with the square root transformation of intensities to compensate for the inherent heteroskedasticity and possible differences in ionization efficiency between the individual MOIs"* -> Why does the manuscript consider square root downscaling an appropriate way of reducing the effect of ions with high ionization response versus ions with low ionization response? Since this is FTICR data, the ionization intensity response can be orders of magnitude different between low and high intensity ions. Square root transformation would not remove that effect necessarily, and the CPPMs could still be dominated by high intensity ions. Please elaborate.

The use of square root and logarithmic intensity transformation for stabilizing variance in the presence of heteroscedasticity in MSI data was previously reported in various studies.^{29,30,31} In the original submission, square root transformation has been applied in the context of CPPMs to compensate for the inherent heteroscedasticity and (at least partially) for possible differences in ionization efficiency between the individual MOIs within a CPPM. We agree with the reviewer that the type of intensity transformation does not necessarily completely eradicate differences in the ionization response. For this reason, we have chosen in this revision to apply even more rigorous intensity transformation by **performing z-score intensity normalization** (i.e. standardization; subtracting mean and dividing by standard deviation) which has been used in more recent MSI applications^{32,33} and is generally widely used outside of the MSI context^{34,35}. This type of transformation aims to equalize the variance of measured MOI intensities by setting the mean intensity of each MOI equal to zero, thereby adjusting for differences in the offset between MOIs with high and low intensity ranges, while, at the same time, setting the standard deviation of intensities equal to one.³⁶ Within *molecularR*, this method has been used exclusively with CPPMs and CPPM ratios, applying this transformation on the intensities of each MOI individually. Although a systematic evaluation of the z-score intensity normalization against the commonly used square-root and logarithmic ones is lacking in the context of MSI data, it has been widely adopted in non-MSI applications, not mentioning the increased popularity of this type of intensity transformation specifically in machine learning applications in MSI and elsewhere is an indication of its effectiveness, especially because such applications are the most sensitive to intensity scale inhomogeneity. The respective statement has been updated in the revised text (**Methods; Collective Projections Probability Map** section).

2.47. SPP representation

- *"converted into pixel-based images with equal pixel grids" and "The resulting raster image is then converted back to an SPP whose points are carrying the respective computed pixel intensities"*-> *The manuscript suggests that SPPs are an improved way of representing of MSI data, and describes the process from grid to SPP. Here, the process is reversed to perform arithmetic operations on a grid*

²⁹ Deininger, Sören-Oliver, et al. "Normalization in MALDI-TOF imaging datasets of proteins: practical considerations." *Analytical and bioanalytical chemistry* 401.1 (2011): 167-181.

³⁰ Wehrl, Patrick M., et al. "Chemometric Strategies for Sensitive Annotation and Validation of Anatomical Regions of Interest in Complex Imaging Mass Spectrometry Data." *Journal of The American Society for Mass Spectrometry* 30.11 (2019): 2278-2288.

³¹ Veselkov, Kirill A., et al. "Chemo-informatic strategy for imaging mass spectrometry-based hyperspectral profiling of lipid signatures in colorectal cancer." *Proceedings of the National Academy of Sciences* 111.3 (2014): 1216-1221.

³² Song, Xiaowei, et al. "Virtual calibration quantitative mass spectrometry imaging for accurately mapping analytes across heterogeneous tissue." *Analytical chemistry* 91.4 (2019): 2838-2846.

³³ Dewez, Frédéric, et al. "Precise co-registration of mass spectrometry imaging, histology, and laser microdissection-based omics." *Analytical and bioanalytical chemistry* 411.22 (2019): 5647-5653.

³⁴ Carré, Alexandre, et al. "Standardization of brain MR images across machines and protocols: bridging the gap for MRI-based radiomics." *Scientific reports* 10.1 (2020): 1-15.

³⁵ Antonakoudis, Athanasios, et al. "The era of big data: Genome-scale modelling meets machine learning." *Computational and Structural Biotechnology Journal* 18 (2020): 3287-3300.

³⁶ van den Berg, Robert A., et al. "Centering, scaling, and transformations: improving the biological information content of metabolomics data." *BMC genomics* 7.1 (2006): 1-15.

representation. Why not do the arithmetic operations on the values in the SPPs directly?

Arithmetic operations on SPPs directly are computationally less efficient than performing that on equivalent raster images. Please refer to **Comments 2.16** where this matter has been thoroughly discussed.

2.48. Spatial autocorrelation and biological relevance

- "with analyte hotspots and/or analyte coldspots superimposed as polygonal contours identifying areas of MOI significant abundance and deficiency, respectively." -> The manuscript seems to assume that MOIs have to spatially autocorrelate. What happens for MOIs that do not (e.g. because their spatial distribution pattern is homogeneous)?

We thank the reviewer for making this point. Indeed, because of its reliance on spatial autocorrelations, *moleculaR* focuses on metabolites with non-homogeneous distributions. These are the ones, for which spatially resolved methods provide an advantage compared to LC-MS work with bulk extracts, which will always feature superior sensitivity. We included this aspect in the discussion of the revised manuscript.

2.49. Ion images

Comments on precision in wording, nomenclature, and mathematical definitions, and the need for additional clarity

- "poorly defined ion images for data interpretation" -> Please explain in what way ion images are "poorly defined".

Instead of using the scientifically not sound statement "poorly defined", we have re-written the abstract and used "traditional" instead. . Also see **comment 2.51**.

2.50. Comment on the word "systematic"

- "systematic probabilistic mapping" -> What makes this "systematic"?

Systematic as in "*methodical in procedure or plan*" (Merriam-Webster). This word has been changed to a more appropriate alternative.

2.51. Bias in ion images

- *"Ion images currently used in MSI do not represent ion intensity of a single observed POI in a user-unbiased way." -> Please elaborate on what is meant exactly when making this statement. The generation of ion images is done using different approaches. For example, both the peak height of a centroided peak and integration over a peak in full profile spectra are common ways of translating spectra into intensity values in an ion image. Does the bias affect both? Other approaches are in use as well, so if the manuscript wants to make a statement in terms of bias, it is necessary to be precise on what type of bias is being referring to. The sentences following this one are insufficient in narrowing this down (see also my comment dedicated to specifying the peak picking method the manuscript starts from).*

There are three types of user-bias involved in MSI ion images presented in the manuscript, two of which are briefly described in the first paragraph of the manuscript with their appropriate citations: *"Ion images, i.e., false color renderings of m/z intervals containing an unassigned peak-of-interest (POI), can be prone to technical artifacts³⁷ and user perception-bias³⁸."* Throughout the manuscript we provide numerous evidence how *molecularR*'s MPMs are more robust against technical artifacts than their counterparts (see **Comments 2.2, 2.4, 2.9, 2.15**). We also explain how MPMs provide the user with the hotspot/coldspot contours that objectively evaluate the spatial relative abundance/deficiency comparing that to the "classical" user perception-biased way of interpreting analytes distribution by visual inspection. The third type is mentioned directly after the statement above and involves the choice of the mass-window for creating an ion image: *"Ion images currently used in MSI do not represent ion intensity of a single observed POI in a user-unbiased way. They rather neglect mass accuracy and resolving power at the POI m/z and use the sum (i.e. uniform weighting) of ion intensities of all peaks present in a user-defined mass range centered on the POI m/z instead"*.

Regarding the generation of the ion images, uniform mass-window weighting and peak picking, please refer to **Comment 2.18**.

2.52. Peak picking and FWHM

- *"They rather neglect mass accuracy and resolving power at the POI m/z" -> More clarity is needed here. The manuscript seems to refer to the case where peak picking is done and then an ion image is generated for a specific peak that was found. Most peak picking algorithms report not only the m/z value of the peak centroid, but also directly report some measure of peak width such as its FWHM. Such a peak width measure reports a heuristic for resolving power, and this information is often used to determine what m/z window to integrate over and thus ends up affecting the ion image directly, so*

³⁷ Balluff, Benjamin, et al. "Batch effects in MALDI mass spectrometry imaging." *Journal of the American Society for Mass Spectrometry* 32.3 (2021): 628-635.

³⁸ Race, Alan M., and Josephine Bunch. "Optimisation of colour schemes to accurately display mass spectrometry imaging data based on human colour perception." *Analytical and bioanalytical chemistry* 407.8 (2015): 2047-2054.

it's not clear what the manuscript means when it says that mass accuracy and resolving power is neglected, at least not in the general case.

We believe that this has been thoroughly discussed in **Comment 2.18**

2.53. FWHM full profile spectrum

- (Figure 1) "Full-width-at-half-maximum (FWHM) values are computed for all peaks of a user-selected full mass spectrum" -> A traditional full profile mass spectrum would describe a peak using multiple neighboring intensity values, covering the peak from left to right. The figures show spectra that look like centroided spectra. It would be good to bring clarity to which type of data is in use here, also because peak picking algorithms on traditional full profile spectra tend to provide FWHM estimations directly, and thus would not require an estimate of FWHM values afterwards as seems to be the case here. Please clarify and provide additional guidance.

*moleculaR works mainly with centroided data, the peak profile information is not available during evaluation. Instead moleculaR relies on a single, user-provided continuous profile spectrum based on which FWHM data is estimated for the entire m/z axis and used for the determination of the extent of the mass-window (i.e. the Gaussian mass-window weighting). For a complete discussion on this matter, please refer to **Comments 2.11.2** and **2.18**. **Comment 2.23** provides a discussion on why a single randomly chosen profile spectrum is enough to build a model of FWHM as a function of the entire m/z axis. The respective statement has been updated in the revised manuscript to clearly indicate what sort of data moleculaR deals with (**Methods**; "MALDI Mass Spectrometry Imaging" and "Data Preprocessing" sections).*

2.54. Gaussian weighting sigma

- (Figure 1) "For any MOI m/z (dashed blue line), a Gaussian envelop is computed with sigma equaling the estimated FWHM at MOI m/z " -> Please motivate why the decrease in weight should follow a Gaussian model. Also, if a Gaussian is the right model, please clarify the definition of sigma in this context. If the manuscript intends to have sigma represent the Gaussian curve's standard deviation (a common notation), it does not seem to make sense to equal sigma to the FWHM. The FWHM of a Gaussian curve is not equal to one standard deviation, in fact it is not exactly equal to 2 standard deviations even, so there seems to be a miscommunication here. Does the sigma in the manuscript mean something else than the standard deviation? Please provide a precise set of definitions and nomenclature to avoid ambiguity.

There was in fact a miscommunication that has been addressed in the manuscript and discussed thoroughly in **Comments 2.4** and **2.6**.

2.55. Significant abundance

- (Figure 1) "to estimate areas of significant abundance" -> The statistical test against a CSR seems to test whether there is spatial autocorrelation in the distribution pattern, not necessarily whether the abundance/intensity is significant. There can be an element of intensity included in the test by using the KDE smoothing, but that does not change the experimental design of testing against a CSR model. Using "areas of significant abundance" seems to imply an intensity argument, which does not seem to be supported by the nature of the test? Please state more clearly in the manuscript what property the significance of which is being tested, and what the implications are for interpretation of the results, for example in Figure 1F.

We believe that this has been thoroughly discussed in **Comment 2.5** and addressed in the revised manuscript.

2.56. Coldspot/hotspot thresholds

- (Figure 1) "(MOI "hotspots"; red/white contours) or deficiency (MOI "coldspots"; blue/white contours)" -> What threshold was determined as the border of significance for both of these? Please state them in the caption as well.

We have created a new **Fig. 2A** (updated from former **Fig. 2B**) which also visually shows the intensity cutoffs for hotspots and coldspots based in the intensity distribution of f_{CSR} , and we updated the figure caption accordingly.

2.57. Artificial noise

- (Figure 1) "raw, noise-free (green)" -> The manuscript states the raw version to be noise-free. Is the raw intensity data noise-free to begin with? How can a user assess this? Please elaborate.

We thank the reviewer for pointing this out. "noise-free" was intended to mean free of artificially added noise. To avoid misunderstanding we have dropped this term in favor of "raw" in the revised manuscript.

2.58. Significance

- Suppl. Figure 2's caption states "statistically significant". -> As mentioned in other comments, please elaborate on what property

exactly the significance is tested of, and state the statistical significance thresholds employed.

We believe that this has been thoroughly discussed in **Comment 2.5** and addressed in the revised manuscript.

2.59. POI-specific bandwidth

- "with an POI-specific bandwidth estimation" -> The values in the SPP are stated to be MOI-specific. When doing the kernel estimation, why go back to a POI-based estimation of the kernel parameter?

This appears to be a typo that has been fixed in the revised manuscript.

2.60. Spatial autocorrelation and biological relevance

- "spatial probabilistic mapping of analytes aids in outlining the significant presence or absence of analytes relative to vital tumor regions" -> This statement seems somewhat imprecise and generalizing. The approach presented reports whether there is spatially local concentration of an analyte (by comparing it to a CSR distribution that has random spatial distribution), it does not report whether the intensity (or absence/presence) of an analyte is significant. Analytes can be significantly present in terms of intensity without their spatial distribution concentrating in certain spots (and thus being spatially autocorrelated). The method presented here only seems to establish the significance of whether spatial autocorrelation occurs. One example would be an ion that is present across the entire tissue being measured. It can be significantly present in the sample, in that it shows up in amounts and with intensities that are significant and that cannot be easily confused with intensity noise levels. The method presented here would only seem to judge whether the ion's distribution spatially autocorrelates. Since inferential statistics and the determination of 'significance' carry a lot of scientific weight, please elaborate on this in the manuscript. Also, since the current statements could lead to misunderstandings, please reformulate wording such as "significant presence or absence" to more clearly separate out for the reader the concepts being tested.

We thank the reviewer for pointing this out. We believe that this has been thoroughly discussed in **Comments 2.5** and **2.20** and addressed in the revised manuscript.

2.61. Significant localization

- "statistically-validated significant localization" -> See other comments on this. Please be precise on

what property is being statistically validated. 'Localization' is insufficiently precise to differentiate between presence/intensity and spatial clustering/autocorrelation.

This has been addressed in the revised manuscript.

2.62. Maps of lipid classes

- "maps of entire lipid classes" -> How are these molecular ensembles defined? Are they groups of molecules that are defined by software on the basis of the measurements in the MSI data? If so, please provide how that works and what procedure was used to obtain the ensembles used in the manuscript. Are these groups of molecules manually curated? If so, please provide how it is avoided that POIs are counted more than once when they happen to match to more than one of the MOIs in the ensemble. Overall, please provide more clarity on the ensemble work.

All CPPM ensembles of MOIs described in this study were queried from the SwissLipids database (ex. PA lipid class), and their presence within the MSI data was verified against the respective annotations of METASPACE (for a given FDR as described in the respective figure captions). Any MOI that has not been identified by METASPACE as being present in the dataset was not included. This has been clarified in the revised manuscript (**Methods; "Collective Projections Probability Maps"** section).

2.63. Pixel resolution - registration

- "5 μm^2 per pixel" -> Do the authors mean a pixel of 5 μm by 5 μm , or really 2.236 μm by 2.236 μm , which amounts to 5 μm^2 per pixel?

This has been addressed in **Comment 2.34** and updated in the revised manuscript.

2.64. Pixel resolution - registration section

- "0.5 μm^2 per pixel" -> Same question as above.

This has been addressed in **Comment 2.34** and updated in the revised manuscript.

2.65. Registration

- "Matt's Mutual Information metric" -> Earlier in the manuscript, the "Advanced Mattes Mutual Information metric" is mentioned. Is this the same metric? If so, please keep the naming consistent. If not, please explain the difference.

It is the same metric. This has been corrected in the revised manuscript.

2.66. Centroided data

- *"The centroided MALDI FTICR dataset" -> The statements in the supplementary "MALDI Mass Spectrometry Imaging" section suggest full profile data, with low intensity values set to zero. That is not commonly considered centroided data. Is there maybe a step missing between the description in "MALDI Mass Spectrometry Imaging" and "Data Preprocessing"? Please provide clarity on the conditions of the data, and show a raw example in a graph. Particularly given the manuscript's work towards summarizing peaks together using a Gaussian weighting scheme, which does not seem to be required for a full profile spectrum, but would make sense for centroided data, it is important that the manuscript makes clear where the challenge being solved enters the data.*

We thank the reviewer for pointing this out. MALDI FTICR MSI data was saved as Profile Spectrum with a Data Reduction Factor of 97% in addition to the centroided mass spectra (SQLite peaks list) which are generated during data acquisition. The reduced profile data was used to sample a single spectrum for the estimation of the FWHM as a function of m/z axis, while the centroided data was preprocessed and ultimately served as an input for *molecularR*. This has been updated in the revised manuscript (**Methods**; "MALDI Mass Spectrometry Imaging" and "Data Preprocessing" sections).

2.67. Isotopes

- *Given the resolving powers involved and the use of FTICR measurements, how does the method address isotopes in the data?*

molecularR does not have an internal mechanism for filtering isotope peaks, however, this step is effectively handled by the POI annotation engine *Metaspace*.

2.68. Full mass spectrum

- *"One pixel representing a full mass spectrum was randomly chosen" -> Does the manuscript mean 'full' in terms of covering the complete m/z axis, or 'full' in terms of "full profile"?*

One pixel representing a continuous full profile mass spectrum. This has been updated in the revised manuscript (**Methods**; "Data Preprocessing" section).

2.69. Profile vs centroided

- *"To minimize data load, data was saved as Profile Spectrum with a*

Data Reduction Factor of 97%." -> The "Data Preprocessing" section in the manuscript states "The centroided MALDI FTICR dataset", while this statement seems to suggest full profile data, at least for the more abundant peaks. Please add more precision about what the conditions are of the data from which the method starts.

This has been addressed in **Comment 2.66** and updated in the revised manuscript (**Methods**; "MALDI Mass Spectrometry Imaging" and "Data Preprocessing" sections).

2.70. Sigma and FWHM

- "compute the expected data-dependent fwhm, which is then used to determine the sigma of a Gaussian envelop" -> The wording and the display in Figure 1 seem to indicate that sigma is matched to the FWHM. The sigma in most common definitions of the Gaussian distribution represents the standard deviation, which is not equal to the Gaussian peak width at half maximum. Are the authors using an alternative definition of the Gaussian distribution in which sigma is not the standard deviation? If so, please be precise on the definition of sigma. If the sigma does represent the standard deviation (which seems to be the case later on), further explanation is needed since the FWHM of a Gaussian distribution is roughly two-fold off from the standard deviation.

We thank the reviewer for pointing this out. There was in fact a miscommunication that has been discussed thoroughly in **Comments 2.4** and **2.6** and updated in the revised manuscript.

2.71. Artificially contaminated data

- (Suppl. Fig. 4) "contaminated data" -> Why call this 'contaminated'? Do the authors mean data changed by their algorithm? Please use a more descriptive label.

i.e. data with artificially noise added, we have updated this in the revised manuscript. Also see **Comment 2.57**.

2.72. Typos

Minor comments & typos

2.72.1 (Figure 1) "(green mesh)" -> The green mesh is hard to see in the image. Please adjust.

The green of former **Fig. 1** and **2** was made clearer in the new **Figs. 3B** and **5BCD**.

2.72.2 *"a mass resolution of 85k" -> Shouldn't this be resolving power? It's listed as a unitless value in the text.*

This has been updated in the revised manuscript.

2.72.3 *Please rewrite sentences such as "m/z values of observed POI and biologically relevant MOI like the potassium adduct of heme ([Heme + K]⁺), i.e., database entries with corresponding theoretical masses, typically differ."*

This has been updated in the revised manuscript.

2.72.4 *"MPMs enables"*

This has been updated in the revised manuscript.

2.72.5 *Suppl. Figure 2. lists "infliction = 2.4". Should this be 'inflection'?*

This figure has been completely revised. It is now **Suppl. Fig. 9**.

2.72.6 *"which equals to the number of points"*

This has been updated in the revised manuscript.

2.72.7 *"two exemplary MOIs" -> "two example MOIs"*

This has been updated in the revised manuscript.

REVIEWER COMMENTS

Reviewer #1 (Remarks to the Author):

The manuscript from D. Sammour et al has substantially changed in the right direction. The new computational experiments and figures made to address and clarify the concerns about the analytical noise, reproducibility and the advantages of the presented method are convincing. As such, I find the manuscript suitable for publication.

Reviewer #2 (Remarks to the Author):

The changes to the manuscript are appreciated and have clarified many details about the methods employed. However, reading the new version of the manuscript, several important issues remain and would need to be resolved before publication can be considered. Several issues pertain to claims made in the manuscript that do not seem to line up with the method implementations. Major comments focus on:

- molecularR results being unbiased and user-independent;
- reliability and statistical significance;
- precision in definition and wording.

Remarks on the adjusted manuscript text

[molecularR results being unbiased and user-independent]

- "Such molecular ensembles could be tailored by biomedical or pharmaceutical scientists to their research interests" -> If the ensembles can be tailored by human scientists, there seems to be inherent bias built into molecularR's results. This seems incompatible with the statement that the manuscript makes about "user-unbiased probabilistic molecular mapping". The methodology presented here can be useful, but it does not seem to be unbiased and should not be presented as such.

- (x) "This procedure enables the plotting of probabilistic "hotspot" and "coldspot" contours for any given MOI independent of how a user may perceive its spatial relative abundance or deficiency." -> The MPMs are based on user-defined (or at least human developer-defined) parameter and model choices, which may or may not be correct. Therefore, the MPMs are also subject to human bias just like any interpretation of ion images directly. The manuscript seems to suggest that MPMs are a more statistically sound or at least unbiased manner of looking at ion intensity data, but the method descriptions do not seem to support the necessary independence from human bias. The MPM approach can be useful, but please refrain from making strong statements on independence from users and bias.

- "molecular probability maps (MPMs) that may complement or replace ion images for data-driven and user-independent spatial analysis of MOI" -> Same remark as for (x). The manuscript describes several user-dependent (or at least developer-dependent, making them implicitly human-dependent) choices that drive the content of MPMs, which seems at odds with the statement that MPMs are user-independent. Furthermore, the manuscript demonstrates a specific example in Fig 3d where the data-driven setting does not perform necessarily optimally, and it is suggested that manual setting of parameters by the user might be needed. Therefore, the MPM approach can be useful, but please refrain from making strong statements on independence from users and bias.

- "full-width-at-half-maximum (FWHM) is plotted against m/z for one randomly chosen single-pixel full (profile) spectrum to obtain an indicator of the non-linear mass resolving power across a mass range for any given experiment" -> Since peak width can be influenced by the presence of other molecular species in the sample (e.g. space charging), how does taking the peak width from one pixel give a representative idea of the FWHM for an m/z across the whole tissue sample? Furthermore, if `moleculaR` is estimating the FWHM, it is implicitly running a peak detector or some form of peak estimation. These procedures have their own parameters and models to set. How are these set, and does this not add to the developer-made choices, with potential bias entering the MPMs?

- "For any metabolite that an MSI user may be interested in, a Gaussian centered on that MOI is computed based on estimates of FWHM at the MOI's m/z ." -> The authors state themselves that a Gaussian shape is not necessarily or always an optimal model for peaks in mass spectrometry, particularly for peaks in high-resolution FTICR spectra. Since `moleculaR` chooses to approximate peaks using a Gaussian shape, with a potential mismatch that comes with that assumption, does this choice not impact the final result? Do these user-made (or developer-made) choices not inject bias into the results?

- "moleculaR allows for a more application-specific surveying by manually setting the KDE bandwidth, which could be either fine-tuned against an orthogonal (ex. optical) method or inferred from what is theoretically expected for the object being imaged (ex. minimum theoretical amyloid plaque diameter)" -> This statement on manual setting of parameters suggests that moleculaR's results are not necessarily user-independent. If this is a conditional matter, then please specify under which conditions moleculaR is reliable in a data-driven fashion, and under which conditions it is not.

- "was set to 12 ppm" -> Why is that a good value to use here? How should the user of the software set this threshold for their own datasets? Should this not be made data-driven as well?

- "A locally-estimated scatterplot smoothing (LOESS) was used for fitting a smoothing curve to describe FWHM as a continuous function of the m/z axis (see below), which was then used to estimate FWHM at any given m/z ." -> It is appreciated that the authors make a theoretical argument for FWHM being a continuous function of m/z . However, there is a difference between the FWHM of the theoretical peak and the FWHM of the measured peak, and the manuscript seems to assume that these two properties are the same. The measured peak's FWHM deviates from its theoretical form by effects such as overlap with other peaks similar in m/z , whether isotopes are practically (fully) resolved, space-charging, etc. Even in the case that one can consider that these practical deviations from the theoretical FWHM are not influential enough to make a difference, the measured FWHM is going to be the result of an estimation of where a peak is in a (potentially noisy) mass spectrum. Since a peak detection procedure makes several assumptions (e.g. peak shape) and uses several parameters, any estimated FWHM will also be at least somewhat dependent on these assumptions and user/developer-defined parameters. It therefore seems optimistic to assume that the FWHM one can get from estimation in a measured mass spectrum is the same as the theoretical FWHM (at least for this assumption to hold for MSI data in general), and it seems optimistic that one can assume that practically estimated FWHMs will still describe a continuous function of m/z (despite the theoretical argument). Do the authors have information to make that claim, which would allow them to use it as an assumption here? If not, what is the impact on the results when the measurements don't meet the assumption made here?

- "Ion images, for instance, are user-perception-dependent" -> An ion image reports the ion intensity recorded by a mass spectrometer across a spatial coordinate plane. No specific human adjustment is typically directly involved in the generation of ion images from the data. If the authors mean that humans interpret ion images, that does not mean the ion images are "user-perception-dependent" themselves, but rather that the interpretation of the ion images is "user-perception-dependent". Please be precise. Furthermore, any image being seen with human eyes is then "user-perception-dependent", so this would make the authors' images "user-perception-dependent" as well. I realise the authors want to make an argument for the hot/coldspot contours being user-independent, but given the number of developer-made choices in generating these contours, I do not consider the contours to be free of human bias either (see my remarks in the original review and this re-review).

In short, this statement seems inaccurate towards ion images (not the interpretation of them), and seems meant to set up MPMs and CPPMs as not biased, which does not seem to bear out by the manuscript text that follows. Please explain or rewrite for accuracy.

- "neither consider nonlinearities in the resolving power of mass spectrometers, nor do they yet evaluate statistical significance of potentially differential spatial metabolite distributions." -> An ion image, or any image for that matter, does not "consider" anything, it can only report a value. An image can also not "evaluate" anything, it can be used as a data source on which to base an evaluation, but the evaluation process is orthogonal to the content of the image. Since statistical significance of various properties of the data can be evaluated on the basis of observations in ion images, these aspects do not necessarily count against ion images. The manuscript seeks to make an argument that ion images are not optimal (which they are for certain purposes and not for others). However, the arguments made in this statement do not succeed in conveying that. Please rewrite for accuracy.

[Reliability and statistical significance]

- "improves signal reliability" -> This is stated as a given in the manuscript, but in the rebuttal is stated as an expectation: "However, we expect them to have less impact on the resulting image intensities of MPMs (because of the Gaussian mass-window weighting; comments 2.2.1 and 2.4) and on the hotspot/coldspot contours (because of the internal KDE smoothing; comment 2.2.2) when compared to state-of-the-art ion image procedures." I agree there is an opportunity for the method to improve reliability, but not necessarily in all cases and it remains somewhat dependent on the assumptions made by the developers working out for the dataset at hand. As such, stating in the manuscript the expectation to improve signal reliability seems more accurate than the general statement made here.

- "probabilistic molecular mapping" -> A variable that varies between 0 and 1 is not necessarily a probability. For that claim to be made, it would be necessary to show that the variable reports the actual chance/probability of an event happening. Since the manuscript does not seem to provide that, the variable between 0 and 1 can of course be used for applications, but it does not seem sufficient to be considered a probability. In short, "molecular mapping" seems to be a statement that is supported by the manuscript, while "probabilistic molecular mapping" does not seem to be supported. Please rewrite the text and naming for precision.

- "molecular probability maps (MPMs)" -> It's not necessarily so that because MPMs provide a value between 0 and 1 that their content reports a probability. Please refrain from using the word

"probability" in this context. A true probability would be inherent to the observed data, and should not change in function of parameter choices, yet different parameter choices by the user deliver different MPM content. Thus, it seems that MPMs as reported by the authors do not report an actual probability (maybe an estimation?). Please explain or rewrite the text and naming for precision.

- "A note of caution: MPM contours can, in principle, be calculated for data from various MSI platforms, but it may be most meaningful for high-resolution MSI" -> If the MPM approach has conditions under which it does not perform, please elaborate. The manuscript makes strong statements about reliability and significance about the provided results. It would serve the understanding of the reader to understand under which circumstances the method can be relied upon and in which cases less so. Here, a specific example is given and mass resolution seems to be suggested as one parameter that influences meaningfulness. Please provide a discussion on these breakdown conditions in the main manuscript to accompany the statements on reliability and statistical significance there.

- Figure 2 -> The procedure for generating MPMs seems dependent on the distribution of ion intensity values present in the measured pixel locations. The generation of coldspot and hotspot contours furthermore relies on fCSR. However, if we were to measure more pixels of this tissue sample (simply measuring a larger ROI in the tissue slice), the additional intensities would move fCSR to a different value and the contours to different locations. For example, if the additional tissue area contains a lot of high-intensity pixels for the POI considered, then the fCSR would end up more to the right on the 'Intensities' axis, the upper and lower cutoffs would presumably end up moving, and therefore the coldspot and hotspot contours would be drawn in different locations on the tissue. Is this correct? If so, how can coldspot and hotspot contours be labeled as "significant" areas to highlight if their location changes dependent on whether one collects more or less tissue area. This could line up with the adjective 'relative' in the notion "significant relative spatial abundance". However, it does not seem to line up with the adjective 'significant' since whether an area in tissue is considered significant should not be influenced by whether or not other tissue areas as measured as well? Please explain and describe in the manuscript. The simulated data that has been added to test the validity of the MPM approach does not test these aspects since it does not increase or decrease the content of the "imaged tissue" substantially and the content of that added tissue. Specifically, the different simulated datasets do not compare 'significant' contour location movement when substantial amounts of pixels with content patterns are added without changing the original pattern (which would be the equivalent of enlarging the recorded tissue to collect additional tissue structure). This also further complicates the stated need for cross-dataset comparability, which is made explicit by the authors when they say "MPMs enable spatially-aware cross-tissue comparison of tryptophan". The within-tissue impact would need to be addressed, and the across-tissue impact separately as well.

- "Here, the CSR model of the test tissue is inferred from signal intensities of the reference tissue." -> Please elaborate on the validity of this experimental design. Unless there are guarantees around the intensity distributions of two datasets being perfectly matched between them (both in number and intensity values), it does not seem valid to use the intensity values of one dataset to judge the significance of the relative spatial abundance of the measured distribution in another dataset (since relative spatial abundance has been presented as a within-tissue evaluation up to this point in the manuscript). A more detailed description of this question can be found in my remark for (xx).

- (xx) "it is important to ensure that the signal intensities of both test and reference tissues are comparable by observing appropriate experimental design which deliberately minimizes technical variation (ex. placing them on the same slide to be measured in a single measurement) 6 and/or by relying on robust intensity normalization methods." -> This experimental design seems to remain flawed despite the condition formulated here. By varying the spatial location of intensities, but keeping the intensity values themselves, one can assess whether the spatial distribution is random or not. That seems to be the classical use of a CSR null hypothesis. If, on top of changing the spatial locations, one also uses other intensity values (such as from another dataset), then the reference being compared to says little about spatial distribution anymore and whether that spatial distribution is 'significant' cannot be assessed. Even if the biological intensities between the datasets are somehow matched as the authors describe here, the content of the tissue will be different (e.g. since one dataset has a tumour in it). This seems to invalidate the use of the reference intensities for the mutated CSR. Please explain in the manuscript why this experimental design remains valid.

- "added to n=10 randomly selected pixels" -> If the test is to assess method robustness, a more noisy example with higher n seems appropriate. The goal of a robustness assessment is to find out at what point the method starts breaking down. If that point is far from the natural MSI datasets one encounters, the method is (relatively) robust. If that point is easily reached, the method does not seem very robust. This simulated example comes across as not disruptive enough to find out where the method breaks.

- "CPPMs enable basic arithmetic manipulations on SPPs of multiple MOIs" -> Since the ion intensity values have a certain uncertainty around their measured value, and arithmetic operations on uncertain values can further extend the uncertainty around their end product, how should the uncertainty around the CPPM result be assessed? The propagation of uncertainty is not only dependent on the number of input variables, but also the particular arithmetic operations used on them. How should readers interpret this aspect of CPPMs for their own cases? Figure 5 shows several examples of division-based CPPMs, where two uncertain variables get divided by each other, which is an operation particularly prone to uncertainty amplification. Please provide guidance for users of `moleculaR` in the manuscript.

- "One pixel representing a continuous (profile) mass spectrum was randomly chosen from the corresponding profile data and exported as a CSV file via flexImaging software version 5.0 (Bruker Daltonics)." -> What if the one randomly chose pixel is not representative (enough) for the rest of the dataset? The mentioned elimination of <1% occurring peaks seems to address one possible form of being not representative, but introduces an additional percentage threshold to tune. Additionally, the spectrum of a single pixel is noisier than an average spectrum, which would suggest that the detected peaks in that spectrum and their estimated FWHM could be more noise sensitive. Is this the case? Does this have an effect on the quality of the FWHM estimation? If not, why not? If yes, how should this problem be addressed by a user of the software?

- "being a low-pass filter" and "This is the point, after which an increase in hKDE does not result in a considerable increase in the spatial autocorrelation of the smoothed density image." -> I agree that the KDE functions as a spatial low-pass filter. This means that it introduces spatial autocorrelation into the pattern it outputs, with the larger the hKDE used, the larger the spatial autocorrelation introduced (because short distance variation is wiped out). As expected, an increase in hKDE tends to result in an increase in spatial autocorrelation in the output pattern. However, the manuscript seems to make an argument that the knee point in that profile informs about the natural autocorrelation present in the measured ion distribution. Please explain why that knee point is not the place where so much small variation has been wiped out by the low-pass filter that no substantial additional autocorrelation can be observed? What is the reason for the knee point to be linked to the natural autocorrelation?

- "Even though Gaussian weighting does improve signal reliability, each ion intensity image will still contain an unknown amount of non-biological technical variability σ , which could be carried on to the composite image representation (error propagation)." -> This statement seems very important and should be made more prominently in the main manuscript, not only the methods section, since it impacts the composite image representation's use case. The figures in the main manuscript do not show measures of uncertainty (which I realize are hard to obtain), but that uncertainty is acknowledged to be there both in molecularR's results and in the original ion images. How should a user of this software's results interpret this, particularly since the reliability of the methods is mentioned explicitly in the text as advantages of molecularR over ion images?

- "were sampled from above and below the upper quartile of the empirical intensities of a MALDI-FTICR-MSI measurement of a human GB tissue sample" -> The signal to noise ratio of MALDI-FTICR data is commonly very good, and typically better than MALDI-TOF or other analyzer types. Using the intensities of MALDI-FTICR experiments seems beneficial to pass the test and seems not necessarily representative for the data types this software is suggested to be used on. Please do a test on a more challenging synthetic data set that uses lower signal to noise ratio measurements to sample from. While being shown to pass a high signal to noise test is ok, the software would be helped by being shown to work on a 'hard' noisy example.

[Precision in definition and wording]

- "nor do they yet evaluate statistical significance of potentially differential spatial metabolite distributions." -> The data property whose statistical significance is evaluated here is stated as "potentially differential spatial metabolite distributions". This seems to be different from the property being used elsewhere in the text, namely "relative spatial abundance". Is it the same, or is this a different property? Please explain and provide a clear definition of the property whose statistical significance is being evaluated. In this concept, how are spatial location and ion intensity combined to make relative spatial abundance? Please provide a formula or some precise statement on what is considered "relative spatial abundance" and what not?

- "ion images, neglect mass accuracy and the resolving power and instrument-dependent peak width at peak-of-interest (POI) m/z " -> This statement seems overly generalising. An ion image can be generated while ignoring mass accuracy, resolving power, and peak width, but this does not have to be the case. There are plenty of examples where ion images are generated without ignoring these aspects. Neglecting these aspects is a property of the study and its experimental design, not a general disadvantage of ion images. Additionally, if ignored, such aspects could also leave a trace in the MPMs and CPPMs generated by molecularR it seems (since molecularR works from the centroided data and the ignoring can happen upstream from molecularR). Please rewrite for accuracy.

- "ion images, <...> typically use the sum of ion intensities of all peaks present in a user-defined mass range centered on the POI m/z instead in a data-independent uniform weighting (i.e. all peaks treated equally) approach" -> Same remark as for "neglect mass accuracy".

- "to compute ion intensities, all peaks in a user-defined mass range centering on a POI are weighted equally and simply summed up" -> Same remark as for "neglect mass accuracy".

- "interference noise" -> What is the definition of 'interference noise'? Which physical process in mass spectrometry is modeled by it? Fig 4a suggests this to be uniformly distributed intensity values, but in a specific range from $\max(fMOI)$ to $5\max(fMOI)$ (in green). Why this intensity range exactly? Why not the full intensity range? Please motivate why this is an appropriate disturbance choice to demonstrate robustness against.

- "Mean and standard deviation of fGaussian(k) are equal to those of fMOI (k)" -> Please elaborate on why making the noise be of equal standard deviation as the signal is an appropriate demonstration of robustness? This choice seems to mimic a signal to noise ratio that is close to one, which is not necessarily common in mass spectrometry. Is that the intention?

- "While MOI's SPPs inherit the gridded spatial locations from MSI data, creating a CSR model on a spatial grid would directly violate the randomness criterion of such models." -> This statement seems internally inconsistent with statements elsewhere in the manuscript. In this sentence the case is made that conversion from a grid to a SPP is necessary to compare against a CSR, which is a benchmark where the locations of events are randomly distributed in the spatial plane. However, this does not seem to be the definition of CSR used elsewhere in the manuscript. In Figure 2a, the CSR is depicted and it seems that it is not the measurement locations in the spatial plane that are randomly distributed, but rather the content of the measurements while keeping the measurement locations in the same regular grid. Therefore, the CSR in Fig. 2a seems content-wise randomly distributed, not randomly distributed in terms of space (the latter being the classical definition of a CSR). This interpretation seems to be confirmed by the following statement by the authors: "For simplicity, random sampling is replaced by randomly permuting the intensities, which basically has the effect of spatial reshuffling of points, until they assume a spatial uniform Poisson process, thus effectively dissolving any spatial clustering or autocorrelation of signals (Fig.2A)." The authors state that spatially random sampling is replaced by random permuting of intensities, and the latter could technically be done directly on a regular measurement grid representation as well. Therefore, it seems like the CSR (at least the implementation used in the manuscript) is no inherent reason to move to SPPs in the first place. I appreciate the SPP view on MSI data, but this section does not seem to state a hard reason for requiring SPPs. Please explain if I misunderstood, or formulate alternative reasons in the manuscript for requiring SPPs.

[Language and typos]

- "neither consider" -> There is no plural set of subjects in this sentence. Please rewrite for clarity.

- "It thereby exemplarily fosters" -> The word "exemplary" seems incorrect here? Please explain how this approach does a very good job (= exemplary) at fostering the spatially-resolved investigation of ion milieus? That sort of relative performance conclusion seems to require benchmarking against other methods or the ground truth, which does not seem to be provided in the manuscript? See also "two exemplary MOIs" where the word seems to be in the same unsupported fashion.

Remarks on the rebuttal answers

- "The relation between FWHM and m/z (or m) is known in the field, especially for FTICR MS, but it is currently not used in data analysis." -> See my remark on the difference between the theoretical FWHM and the FWHM that can be practically retrieved and estimated from measurements.

- "The resulting curves suggest that the KDE bandwidth estimation method of MPM coincided with the highest (i.e. the optimum) DSC for all simulated SPPs. This in itself was an unexpected finding that underlines the strength of this approach." -> The simulated data is in terms of spatial structure very simple, with very little small distance variation to begin with. When applying KDE (and therefore a low-pass filter) on these spatial patterns, small distance variation is increasingly wiped out as hKDE grows. However, if the patterns have little small distance variation to begin with (as is the case here), it is not surprising that the wiping out of small distance variation has little effect on the comparison between the ground truth pattern and the spatially smoothed pattern. The concern of Reviewer 1 seems to be the wiping out of biologically real but small distance patterns, and that is not tested by these examples. Furthermore, the matter of intensity used also plays a role, since small distance patterns with high intensity content will resist better against being wiped out by the KDE, than small distance patterns with low intensity content.

- "For investigating this further, additional computational experimentation would be needed that involved multiple slides or measurements where robust intensity normalization becomes crucial. This, however, is beyond the scope of the proposed method, as it would require extensive studies of entire (patient) cohorts of samples, which were not available here." -> Understood, but please make a clearer statement in the manuscript that this is a use case that is not (yet) validated and verified.

- "Therefore, the smoothing the reviewer refers to does NOT eliminate signals but is only an internal mechanism that directly affects estimated hotspots/coldspot areas." -> Understood, but that does not mean that the smoothing has no effect on the results (= where a hot/coldspot's contours are drawn) and the fact that these move depending on the hKDE chosen. This puts the statistical significance argument a bit in question.

- "Since smoothing SPPs (and CSRs) with high KDE bandwidths could result in underappreciating fine spatial structures or secondary MPM hotspots, this effect must be balanced against the desired effect of improved signal reliability." -> This smoothing effect is further complicated by the particular intensity values of the POI. The balancing is therefore not only a matter of where but also how

much. The Kneede approach makes the hKDE selection automated, but it does not solve the fundamental issue of what small distance variation is ok to be removed from the significance calculation and whether that's an ok thing to do given what conclusions are going to be drawn from molecularR's results. Automation of the hKDE setting does not solve the fundamental issue, it only removes the decision from the user's view.

- "provide a user-independent and unbiased indication of where analytes' intensities have significant relative spatial abundance (or deficiency) that is unlikely to be seen if these signals were generated due to a random event (Fig. 2A)." -> The statement that MPMs would be unbiased does not seem to correspond with the authors' choices made in the algorithm that produces the MPMs. For example, the choice to approximate a peak in a mass spectrum as a Gaussian peak is a choice. It can be a perfectly good choice, but it is a choice, and all subsequent results are biased by that choice. There is a difference between being unbiased and making defensible choices, and the method described here seems to fall in the latter category. Along the way to making an MPM several choices are made, so it becomes quite hard to maintain that the results are truly unbiased. The same argument can be made for the adjective user-independent. Do the authors mean 'automated'? I think the latter could be more defensible than user-independent. Additionally, the user-independent adjective seems less appropriate with author statements like the following "molecularR also permits the user to manually set the KDE bandwidth against a theoretically acceptable value which tends to reduce the spatial constraint described above." If manual setting is a viable and sometimes necessary option, how can the process be user-independent. The automated determination of hKDE would only mean that the choice is taken from the user and performed by the developers of the software, not that it is independent of humans. Please acknowledge these matters in the main manuscript text.

- "As mentioned above, in our additional mouse brain MALDI FTICR MSI experiment with N=6 technical replicates (=adjacent tissue sections) in New Suppl. Fig. 21, MPM hotspot contours were very comparable. All considerations for MPMs, in principle, also apply for CPPMs." -> A positive example does not show that the method is reliable, it shows that the method is reliable in that specific dataset and choices case. A negative example would be more informative as it would demonstrate when the method starts failing and in which cases it's maybe safer to use.

- "Our data suggests that the proposed Gaussian mass-window weighting can help to resolve MOI ambiguities in a data-driven and user- unbiased way taking into account the mass resolving power at the analyte under study which can vary depending on m/z of the analyte, measurement parameters and the imaging platform being used. We refer editor and reviewers to detailed discussions of new Suppl. Fig. 4-7 above. The considerations above have been acknowledged in various parts of the revised manuscript." -> I agree that the Gaussian weighting can have a beneficial effect, but it should be more clearly acknowledged in the manuscript that this a choice by the authors and that it biases any followup results from this choice. The same should be stated for any other choices or parameter values. For example, if a choice is the right one for 90% of the datasets in the world, there is still 10% of the datasets for which it is not the right choice. The statements made in the manuscript around

being unbiased and about significance seem to suggest otherwise and could be misunderstood by readers.

- "It must be noted, however, that this is one of the very rare cases, where a user has access to something like a ground-truth" -> This is not a very reassuring statement. If the user does not have a ground truth to compare to, she is to use the value suggested by the software? Whether the parameter values (and their results) are right or wrong should not be influenced by whether or not there's a ground truth to falsify the suggested value with. If the authors want to make an argument around reliability of molecularR results, it is important to acknowledge these less than convenient aspects more prominently in the manuscript text so that the reader can evaluate themselves.

- "This means that while pixel intensities (of secondary as well as primary hotspots) are reduced with increased smoothing bandwidth, the quantile intensity cutoffs, which are used as thresholds for hotspots and coldspots, shrink towards the mean accordingly (Suppl. Fig. 11B)." -> Understood, but it does not mean that the location and 'significance' of the contours is not dependent on the chosen hKDE, and does not change if a different hKDE is used. Using the word 'significance' to describe these (quite variable) contours seems too strong a statement to make.

- "This, however, is beyond the scope of this manuscript and the proposed method, as it would require extensive studies of entire (patient) cohorts of samples, which were not available here." -> Understood, but the wording used to describe the generated results does not lie beyond the scope of this manuscript. The authors changed "significant analyte presence/absence" to "significant relative spatial abundance/deficiency" which seems to be a step in the right direction, but then changes the problem to what is "relative spatial abundance"? Due to the KDE operation, the concept seems to entangle intensity with spatial location, but I did not find a clear definition of what this is exactly. Furthermore, Fig4e seems to suggest that for the CSR of the test tissue the intensities of the reference tissue are used. This would mean that both the spatial location and the content are changed in the significance test for the test tissue and that only the spatial location is changed in the reference tissue. Is that a fair comparison? Does the test remain fair for the test tissue specifically? I understand that the authors want to compare between reference and test tissue, but the significance referred to in the manuscript is about whether the intensity pattern within a tissue is significantly different, so the significance test would need to be compared within-tissue not across tissue. If the authors want to change the definition of what is being tested, further elaboration in the text is necessary I think. Additionally, if the concept is to be named "relative spatial abundance", elsewhere in the manuscript 'relative' seems to refer to within-tissue relative change, while in this part of the manuscript 'relative' seems to mean within-tissue and across-tissue combined, which introduces an ambiguity: Is the tryptophan significantly different between test and reference because of across-tissue differences, because of within-tissue difference, or both? Please be precise.

- "According to the laws of error propagation (linear, Gaussian, other) arithmetic operations may always add (relative or absolute) uncertainty. However, we expect them to have less impact on the resulting image intensities of MPMs (because of the Gaussian mass-window weighting; comments 2.2.1 and 2.4) and on the hotspot/coldspot contours (because of the internal KDE smoothing; comment 2.2.2) when compared to state-of-the-art ion image procedures." -> Why would this expectation be a valid assumption? Do the authors have a specific methodology in mind they are comparing to? Also, the argument of uncertainty is not a matter of comparing to what other methods do. The point is specific to the arithmetic operations suggested in this manuscript (but would also hold for arithmetic operations directly on ion images). For example, dividing by a denominator with an uncertainty connected to it will often have an amplifying effect on the uncertainty of the ratio, particularly if the denominator takes small values. The most extreme case would be if zero falls inside the uncertainty range of the denominator, where you would get immediate amplification of uncertainty of the ratio value to include infinity. This type of effect is orthogonal to whether the nature of the uncertainty is understood, and even if zero would not fall inside the uncertainty of the denominator the amplification effect on the ratio is still there, just less exorbitant. The manuscript should make clear that performing mathematical operations on ion intensities, including some of the ratios being taken in the examples, can have an uncertainty amplifying effect. This point should particularly be made in the main manuscript since the text makes an explicit argument for reliability and significance assessment.

- "We have therefore updated the method and used the Benjamini-Hochberg P-value correction approach throughout the revised manuscript." -> The change of the p-value correction approach directly influencing the MPM results demonstrates the point I've tried to make elsewhere that the results of this approach are not unbiased, they are biased by the choices made by the developers, e.g. the choice of p-value correction. Please avoid calling these approaches unbiased.

- "While MOI SPPs inherit the gridded spatial locations from MSI data, creating a CSR model on a spatial grid would directly violate the randomness criterion of such models. Therefore, in order to facilitate the direct and homogeneous comparison between MOI and a corresponding CSR model, MOI spatial intensities must be converted into an SPP representation" -> This argument for SPPs seems to be contradicted by the authors elsewhere where they state "For simplicity, random sampling is replaced by randomly permuting the intensities, which basically has the effect of spatial reshuffling of points, until they assume a spatial uniform Poisson process, thus effectively dissolving any spatial clustering or autocorrelation of signals (Fig.2A)." This latter operation can happen on a regular spatial grid, so does that then not "directly violate the randomness criterion of such models"? I seem to misunderstand what is meant here. Please explain.

- "Here, and especially for CPPMs, molecularR provides simple tools to filter out duplicated counts of the same m/z value by incorporating only the intensities of unique masses present in the computed CPPM." -> Were these tools turned on for the examples in the manuscript? Please mention them there more prominently.

- "We also explain how MPMs provide the user with the hotspot/coldspot contours that objectively evaluate the spatial relative abundance/ deficiency comparing that to the "classical" user perception-biased way of interpreting analytes distribution by visual inspection." -> The word 'objective' is subject to the same criticism of how the word 'unbiased' is used in the manuscript. See my remarks elsewhere.

Point-for-Point Reply to Reviewers – Revision #2

We would like to thank both reviewers for their important remarks and constructive comments made in both rounds of revision. Both reviewers have called for a high degree of precision in the description, which we appreciate and acknowledge. We have replied on >80 pages – with substantial contributions from PhD student Denis Abu-Sammour, but also from two mathematical and computational experts. The peer review has clearly made this manuscript much better. We also thank Reviewer 1 for recommending the manuscript for publication.

To address concerns raised in both rounds of revisions, we have performed a substantial number of additional MALDI imaging and computational experiments. We have added numerous explanatory, illustrative and supporting figures (three and 26 additional main and supplementary figures, respectively, including seven new supplementary Figures in this second round of revisions), and more than 100 major commits to the *moleculaR* github repository. This is testament to peer review excellence.

1. Reviewer #1

1.1. Comment

The manuscript from D. Sammour et al has substantially changed in the right direction. The new computational experiments and figures made to address and clarify the concerns about the analytical noise, reproducibility and the advantages of the presented method are convincing. As such, I find the manuscript suitable for publication.

We thank the reviewer for the constructive points raised in Revision-1 and for recommending our manuscript for publication.

2. Reviewer #2

General reply:

We appreciate all the excellent constructive points raised in both Revisions by the reviewer. We realized that certain claims like user-independence, which were heavily criticized by the reviewer, are actually not main aspects of the innovation presented in this study. Hence, we paid much attention in revision 2 to focus the manuscript on what we believe the true innovation is, namely i) the data-dependent weighting of ion intensities, ii) the statistical testing against complete spatial randomness models (and even other tissues in cross-tissue maps) to identify areas of increased/decreased relative spatial abundance iii) thorough testing of these maps against simulated ground truths as well as real data from various MSI instruments and applications, and iv) the computation of statistically tested collective projection maps of entire metabolite ensembles.

We trust that we have much improved the manuscript to clearly highlight the novel concepts that the *moleculaR* framework introduces to the MSI field. We do acknowledge, however, that certain aspects like the uncertainty associated with composite image scores and error propagation may not be fully addressed by the manuscript and should be considered a work-in-progress. We have made this fact transparent in the manuscript. We have tried our best to present supporting evidence for every method design consideration with supporting literature, argument, illustration, simulation and MSI

data of different modalities and tissues. As always in science, this will just be a step forward. Follow-up studies must be carried out, in order to further develop this interesting subject.

Many remarks made by reviewer 2 address aspects, which are also currently at the forefront in much more seasoned and advanced MS-based proteomics, e.g., the definition of experimentally testable ground truths – see, for example, the recent publication from Steve Gygi's laboratory in Nature Methods¹.

We hope and trust that the MSI community will eagerly discuss and use the new computational concepts implemented here and engage in further developing and updating the open-source *moleculaR* „project“.

2.1. Comment

The changes to the manuscript are appreciated and have clarified many details about the methods employed. However, reading the new version of the manuscript, several important issues remain and would need to be resolved before publication can be considered. Several issues pertain to claims made in the manuscript that do not seem to line up with the method implementations. Major comments focus on:

- *moleculaR results being unbiased and user-independent;*
- *reliability and statistical significance;*
- *precision in definition and wording.*

We thank the reviewer for the constructive points raised in both Revisions. Such a dedicated effort to guide and help make a manuscript much better is rare in today's science. We greatly appreciate this contribution.

[moleculaR results being unbiased and user-independent]

2.2. Tailored Molecular Ensembles

- "Such molecular ensembles could be tailored by biomedical or pharmaceutical scientists to their research interests" -> If the ensembles can be tailored by human scientists, there seems to be inherent bias built into moleculaR's results. This seems incompatible with the statement that the manuscript makes about "user-unbiased probabilistic molecular mapping". The methodology presented here can be useful, but it does not seem to be unbiased and should not be presented as such.

We thank the reviewer for this remark. There might be a misunderstanding here though. The message of this statement is that *moleculaR* provides scientists with the ability to visualize not only one molecular entity at a time (as is usually the case with ion images) but also a custom list of MOIs if needed. This list depends on their research question. For example, the researcher might want to investigate the collective intensity of a subset of saturated lipids within a certain lipid class.

¹ Gassaway BM, Li J, Rad R, Mintseris J, Mohler K, Levy T, Aguiar M, Beausoleil SA, Paulo JA, Rinehart J, Huttlin EL, Gygi SP. A multi-purpose, regenerable, proteome-scale, human phosphoserine resource for phosphoproteomics. Nat Methods. 2022; 19(11):1371-1375. doi: 10.1038/s41592-022-01638-5.

Researchers/end users thereby do define what molecules they are interested in, but they do not engage in or interfere with the computation in *moleculaR*.

As already outlined above, we realized that the wording of “unbiased” or “user-independent”, which is NOT essential for the innovative core of *moleculaR*, might distract the readers’ attention from the computational innovations that do constitute this core. Therefore, **we have removed these words in the entire manuscript**. Please, see also **Comments 2.3, 2.4, 2.36 and 2.38**.

2.3. Human/User/Developer-made Choices

- (x) *"This procedure enables the plotting of probabilistic “hotspot” and “coldspot” contours for any given MOI independent of how a user may perceive its spatial relative abundance or deficiency." -> The MPMs are based on user-defined (or at least human developer-defined) parameter and model choices, which may or may not be correct. Therefore, the MPMs are also subject to human bias just like any interpretation of ion images directly. The manuscript seems to suggest that MPMs are a more statistically sound or at least unbiased manner of looking at ion intensity data, but the method descriptions do not seem to support the necessary independence from human bias. The MPM approach can be useful, but please refrain from making strong statements on independence from users and bias.*

The following discussion addresses Comments 2.3, 2.4, 2.36, 2.38 and 2.46.

We thank the reviewer for insisting on this, and we appreciate the precision in description the reviewer is calling for. First, it is indeed important to distinguish between the developer and the (end) user. Given the fact that *moleculaR* also provides an interactive user-interface, we do not assume that the (end) user is a developer/bioinformatician. The end user who we had in mind when referring to a user (and even perhaps the inexperienced end user) is making many selections when dealing with current (vendor) software. These are distinct from developer-made choices. Of course, we do hope that in addition to end users, also developers / bioinformaticians will employ *moleculaR*, and these would obviously introduce their own design choices. As stated above though, we now clearly see the point the reviewer was making, but **feel that we would rather distract future readers with words like “unbiased” or “user-independent”**. Hence, we have chosen to remove them from the manuscript.

That said, we agree with the reviewer that statistical tests, regression models and machine learning algorithm all have assumptions, parameters, models and developer-made choices (i.e. method design consideration) embedded in them. As such, the reviewer’s statement (here and in other comments) suggests that all computational methods are intrinsically biased. Even if that was accepted, there would be a difference between these developer-made choices that are typically based on a lot of MSI expertise and (perhaps inexperienced) end users selecting mass windows and reporting images, the basis of which is then later hard to judge in peer-review. This is a discussion that the MSI field may want to engage in. We removed it from this manuscript.

To indicate though that our initial stance was not without precedent in the literature, we want to mention two high-profile papers: The well-known computational and visualization method t-SNE entails not only developer-made but also user-made parameter choices (such as perplexity and number of iterations). It has been described as an unbiased method for uncovering tumor subpopulations in a previous study ² and has been instrumental for establishing a DNA methylation-

based CNS tumor reference system in another landmark study that some of our co-authors contributed to.³ Several other published studies have also described their computational methods and analyses, which were not without assumptions, as unbiased.^{4,5,6} As scientists, we have tried our best to provide supporting evidence for every method design consideration (“developer-made choices”) with supporting literature, argument, illustration, simulation and MSI data of different modalities and tissues.

Within the previous revised version of the manuscript (and more so in the reply to revision-1), we used “unbiased” or “user-unbiased” to indicate that the method provides **i)** data-driven rendering of ion intensities (FWHM-based mass-window width and Gaussian mass-window weighting) and **ii)** spatial interpretation and localization of non-random spatial MOI patterns (MOI hotspots/coldspots). This is independent from the end user’s perspective/perception/ interpretation and is based instead on clearly outlined CSR-based statistical significance testing (**Comment 2.40**). In other words, we benchmarked the proposed *moleculaR* method against the typical interrogation procedure employed in MSI; on the one hand, an end user’s visual perception of intensity values, which is judging roughly whether an analyte had a relatively higher spatial abundance in a certain tissue morphology. On the other hand, MPMs, which provide MOI hotspot designations based on a clearly outlined statistical model. For example, as illustrated in **Fig 3B**, with ion images (top row) the researcher would be relying on his/her perspective to describe and localize MOI accumulation while MPMs (bottom row) clearly outlines hotspots and coldspots without leaving any room for personal user opinion (but based on developer choices indeed).

2.4. Human/User/Developer-made Choices

- *"molecular probability maps (MPMs) that may complement or replace ion images for data-driven and user-independent spatial analysis of MOI" -> Same remark as for (x). The manuscript describes several user-dependent (or at least developer-dependent, making them implicitly human-dependent) choices that drive the content of MPMs, which seems at odds with the statement that MPMs are user-independent. Furthermore, the manuscript demonstrates a specific example in Fig 3d where the data-driven setting does not perform necessarily optimally, and it is suggested that manual setting of parameters by the user might be needed. Therefore, the MPM approach can be useful, but please refrain from making strong statements on independence from users and bias.*

2.4.1. *The manuscript describes several user-dependent (or at least developer-dependent, making them implicitly human-dependent) choices that drive the content of MPMs, which seems at odds with the statement that MPMs are user-independent.*

² Abdelmoula, Walid M., et al. "Data-driven identification of prognostic tumor subpopulations using spatially mapped t-SNE of mass spectrometry imaging data." *Proceedings of the National Academy of Sciences* 113.43 (2016): 12244-12249.

³ Capper, David, et al. "DNA methylation-based classification of central nervous system tumours." *Nature* 555.7697 (2018): 469-474.

⁴ Ikonomidou, Laertis, et al. "The in vivo genetic program of murine primordial lung epithelial progenitors." *Nature communications* 11.1 (2020): 1-17.

⁵ Guo, G., et al. "Automated annotation and visualisation of high-resolution spatial proteomic mass spectrometry imaging data using HIT-MAP." *Nature communications* 12.1 (2021): 1-16.

⁶ Kim, Yejin, et al. "Multimodal phenotyping of Alzheimer’s disease with longitudinal magnetic resonance imaging and cognitive function data." *Scientific reports* 10.1 (2020): 1-10.

Please, see **Comment 2.3**. We do not disagree with the reviewer that any software is developer-dependent. This does not imply though that its design is not well thought through. Therefore, we tried hard to explain and validate our design choices. “User-dependent” was meant to refer to choices made by an end user (especially, non-MSI expert end users, who will hopefully utilize MSI more and more, and who, in our opinion, should be supported by additional computational concepts). We believe that the reviewer may agree with us that MSI is fortunately getting out of the MSI-techy corner right now. It is employed more and more by clinical and life scientists who do not even want to get into much MS details but who want to spatially analyze the molecules that their research uncovered.

2.4.2. Furthermore, the manuscript demonstrates a specific example in Fig 3d where the data-driven setting does not perform necessarily optimally, and it is suggested that manual setting of parameters by the user might be needed. Therefore, the MPM approach can be useful, but please refrain from making strong statements on independence from users and bias.

This example was used to represent the closest thing we could think of as a ground truth in real data to accompany our many examples and detailed analysis of simulated data. Amyloid plaques have been analyzed in many publications with classic methods like immunohistochemistry, but also using MSI. Our lab has contributed to several of these studies, and we have ourselves validated MSI signals of various amyloid peptides in comparison with wild-type mice, where these peptides/signals are absent.⁷ We agree that the way we presented this was distracting from the message we wanted to convey. Hence, we rephrased this, even though we had made it clear in the previous version that manual fine-tuning would have to be guided by orthogonal, quality-assured methods. In addition, we have taken out said strong statements (see above).

2.5. FWHM Model

- "full-width-at-half-maximum (FWHM) is plotted against m/z for one randomly chosen single-pixel full (profile) spectrum to obtain an indicator of the non-linear mass resolving power across a mass range for any given experiment" -> Since peak width can be influenced by the presence of other molecular species in the sample (e.g. space charging), how does taking the peak width from one pixel give a representative idea of the FWHM for an m/z across the whole tissue sample? Furthermore, if molecularR is estimating the FWHM, it is implicitly running a peak detector or some form of peak estimation. These procedures have their own parameters and models to set. How are these set, and does this not add to the developer-made choices, with potential bias entering the MPMs?

2.5.1. Since peak width can be influenced by the presence of other molecular species in the sample (e.g. space charging), how does taking the peak width from one pixel give a representative idea of the FWHM for an m/z across the whole tissue sample?

In general, we agree with the reviewer that there are developer’s design choices that we have made, which may not be considered parameter-free (see above). Again, we consider this different from end

⁷ Enzlein T, Cordes J, Munteanu B, Michno W, Serneels L, De Strooper B, Hanrieder J, Wolf I, Chávez-Gutiérrez L, Hopf C. Computational Analysis of Alzheimer Amyloid Plaque Composition in 2D- and Elastically Reconstructed 3D-MALDI MS Images. *Anal Chem.* 2020 Nov 3;92(21):14484-14493; Michno W, Stringer KM, Enzlein T, Passarelli MK, Escrig S, Vitanova K, Wood J, Blennow K, Zetterberg H, Meibom A, Hopf C, Edwards FA, Hanrieder J. Following spatial A β aggregation dynamics in evolving Alzheimer's disease pathology by imaging stable isotope labeling kinetics. *Sci Adv.* 2021 Jun 16;7(25):eabg4855

user bias, but have taken out “unbiased” or “user-independent”, in order to focus more clearly on the innovative concepts presented in the manuscript.

For a more specific response: We have provided evidence that FWHM estimation based on a single randomly chosen spectrum is very similar to the FWHM fitted curve based on 100 randomly chosen spectra in four different MALDI-FTICR-MSI measurements (**Suppl. Fig. 3**). We have also shown that the created model fit closely follows the theoretically expected behavior based on **equation 3** in **Methods** (**Suppl. Fig. 4A**). The location that the chosen full profile spectrum is taken from seems irrelevant. The same can be said for the identities of the peaks present in it. This is because it is assumed (and observed; **Suppl. Fig. 3AB**) that the dataset is measured with the same measurement parameters, and therefore resolving power as a function of m/z should be similar across the tissue. Moreover, to address the concern above, we have updated *moleculaR* such that it now also has the capability of employing more than one continuous spectra for FWHM model fitting, i.e. at least one full profile spectrum has to be used for FWHM estimation.

2.5.2. Furthermore, if moleculaR is estimating the FWHM, it is implicitly running a peak detector or some form of peak estimation. These procedures have their own parameters and models to set. How are these set, and does this not add to the developer-made choices, with potential bias entering the MPMs?

As described in **Methods**, *moleculaR* is designed to work with centroided MSI data in this study. This was generated during data acquisition by the measurement device with $SNR \geq 3$, which is commonly accepted in the MSI community as a reasonable threshold for peak detection. Therefore, the developer-made choices here are not arbitrary; they are based on what is commonly accepted in the MSI literature and on what is scientifically justified in comparison to theoretical expectation (**Suppl. Fig. 4A**). Please, also see **Comment 2.3** and **2.4**.

2.6. Gaussian Shape

- "For any metabolite that an MSI user may be interested in, a Gaussian centered on that MOI is computed based on estimates of FWHM at the MOI's m/z ." -> The authors state themselves that a Gaussian shape is not necessarily or always an optimal model for peaks in mass spectrometry, particularly for peaks in high-resolution FTICR spectra. Since moleculaR chooses to approximate peaks using a Gaussian shape, with a potential mismatch that comes with that assumption, does this choice not impact the final result? Do these user-made (or developer-made) choices not inject bias into the results?

As stated above, we distinguish between user-made and developer-made choices. The latter we agree on with the reviewer. We are providing literature and experimental evidence wherever possible, and we took out any wording like “unbiased”.

As a more specific response, we would like to emphasize that the use of the Gaussian, while not necessarily perfect in all cases, is preferred over not using any down weighting whatsoever. Furthermore, the use of a Gaussian to model peak shape in MSI has been previously shown in

numerous examples in the MSI literature including for FTICR data.^{8,9,10} Since the fitted Gaussians are approximations of peak shapes, there is always an error associated. However, based on what is shown in **Suppl. Fig. 4A** where a comparison is made to the theoretical model, there is no evidence for any result-altering or otherwise significant impact. Note that the Gaussian shape is not user-chosen. For the “developer-made choices” topic, please see the above.

2.7. KDE Bandwidth

- *"moleculaR allows for a more application-specific surveying by manually setting the KDE bandwidth, which could be either fine-tuned against an orthogonal (ex. optical) method or inferred from what is theoretically expected for the object being imaged (ex. minimum theoretical amyloid plaque diameter)" -> This statement on manual setting of parameters suggests that moleculaR's results are not necessarily user-independent. If this is a conditional matter, then please specify under which conditions moleculaR is reliable in a data-driven fashion, and under which conditions it is not.*

We thank the reviewer for this remark. We rephrased this. Please, refer to **Comment 2.4.2**. More specifically, knee point estimation via the *kneedle* method, provides a data-driven parameter setting for the KDE bandwidth (**Suppl. Fig. 10**).

2.8. Binning

- *"was set to 12 ppm" -> Why is that a good value to use here? How should the user of the software set this threshold for their own datasets? Should this not be made data-driven as well?*

The chosen 12 ppm refers to the maximum relative peak deviation used in the binning process in this study (see **Comment 2.40-Revision1** and **Methods** for a complete description). It equals $\Delta m/m$, where Δm = FWHM at m/z 400 \approx 0.0048 Da and m = m/z 400. Since the focus of this study was on the lipidome, m/z 400 was chosen, as it is the typical starting mass range at which lipids signals are observed. Likewise, if the study focus was on smaller molecules or metabolites, then m should be chosen appropriately.

2.9. Measured vs Theoretical FWHM

- *"A locally-estimated scatterplot smoothing (LOESS) was used for fitting a smoothing curve to describe FWHM as a continuous function of the m/z axis (see below), which was then used to estimate FWHM at any given m/z ." -> It is appreciated that the authors make a theoretical argument for FWHM being a continuous function of m/z . However, there is a difference between the FWHM of the theoretical peak and the FWHM of the measured peak, and the manuscript seems to assume that these two properties*

⁸ Djambazova, Katerina V., et al. "Resolving the complexity of spatial lipidomics using MALDI TMS imaging mass spectrometry." *Analytical Chemistry* 92.19 (2020): 13290-13297.

⁹ Boskamp, Tobias, et al. "Using the chemical noise background in MALDI mass spectrometry imaging for mass alignment and calibration." *Analytical Chemistry* 92.1 (2019): 1301-1308.

¹⁰ Källback, Patrik, et al. "msiQuant—quantitation software for mass spectrometry imaging enabling fast access, visualization, and analysis of large data sets." *Analytical chemistry* 88.8 (2016): 4346-4353.

are the same. The measured peak's FWHM deviates from its theoretical form by effects such as overlap with other peaks similar in m/z , whether isotopes are practically (fully) resolved, space-charging, etc. Even in the case that one can consider that these practical deviations from the theoretical FWHM are not influential enough to make a difference, the measured FWHM is going to be the result of an estimation of where a peak is in a (potentially noisy) mass spectrum. Since a peak detection procedure makes several assumptions (e.g. peak shape) and uses several parameters, any estimated FWHM will also be at least somewhat dependent on these assumptions and user/developer-defined parameters. It therefore seems optimistic to assume that the FWHM one can get from estimation in a measured mass spectrum is the same as the theoretical FWHM (at least for this assumption to hold for MSI data in general), and it seems optimistic that one can assume that practically estimated FWHMs will still describe a continuous function of m/z (despite the theoretical argument). Do the authors have information to make that claim, which would allow them to use it as an assumption here? If not, what is the impact on the results when the measurements don't meet the assumption made here?

We thank the reviewer for this suggestion. FWHM values used for estimating a FWHM vs m/z model are the measured peaks' FWHMs. It is, of course, true that the measured FWHM values will be affected by peak-overlap and other factors mentioned by the reviewer. Nevertheless, while this question of whether such a model is the best possible model is fair, it is still clear from our results that data processed this way is clearly improved (e.g., **Suppl. Fig. 7**). More importantly, the fitted LOESS model is expected to compensate for FWHM "abnormalities" by smoothing out deviations, as is evident, for example, in **Suppl. Fig. 4A** where data points (i.e. measured FWHM values) vary substantially on the y-axis, possibly for the aforementioned reasons. However, the proposed LOESS fit, which tries to model the data-specific FWHM as a function of mass, provides a good approximation of the true (theoretical) FWHM (dashed pink curve of **Suppl. Fig. 4A**, which is based on **equation 3** in **Methods**) despite these disturbances. Regarding bias and peak detection please see **Comments 2.3, 3.4** and **2.5.2**.

2.10. Ion Images

- "Ion images, for instance, are user-perception-dependent" -> An ion image reports the ion intensity recorded by a mass spectrometer across a spatial coordinate plane. No specific human adjustment is typically directly involved in the generation of ion images from the data. If the authors mean that humans interpret ion images, that does not mean the ion images are "user-perception-dependent" themselves, but rather that the interpretation of the ion images is "user-perception-dependent". Please be precise. Furthermore, any image being seen with human eyes is then "user-perception-dependent", so this would make the authors' images "user-perception-dependent" as well. I realise the authors want to make an argument for the hot/coldspot contours being user-independent, but given the number of developer-made choices in generating these contours, I do not consider the contours to be free of human bias either (see my remarks in the original review and this re-review). In short, this statement seems inaccurate towards ion images (not the interpretation of them), and seems meant to set up MPMs and CPPMs as not biased, which does not seem to bear out by the manuscript text that follows. Please explain or rewrite for accuracy.

2.10.1. *An ion image reports the ion intensity recorded by a mass spectrometer across a spatial coordinate plane. No specific human adjustment is typically directly involved in the generation of ion images from the data.*

Indeed, our statement was imprecise. It is, of course, true that ion intensity is recorded by a mass spectrometer across a spatial coordinate plane. During **rendering** of the ion images, however, end users typically define the extent of the mass-window where (peak or profile) signals are to be integrated either as an absolute value in Da or as a relative value in ppm (relative to m/z MOI). This is seen in major software platforms dedicated to analysis and visualization of MSI data (ex. FlexImaging, SCiLS LAB; Bruker Daltonics) as well bioinformatics packages (ex. Cardinal, MALDIquant).

2.10.2. *If the authors mean that humans interpret ion images, that does not mean the ion images are "user-perception-dependent" themselves, but rather that the interpretation of the ion images is "user-perception-dependent". Please be precise.*

True! We have updated this statement in the revised manuscript. The rendering and interpretation of the ion images are user-perception-dependent.

2.10.3. *Furthermore, any image being seen with human eyes is then "user-perception-dependent", so this would make the authors' images "user-perception-dependent" as well.*

We tend to disagree here, as the focus of this study is the computation of statistically tested MOI hotspots computed by the MPM method. The rendering and interpretation of ion images is subject to end user perception bias, and this has been commented on in the MSI literature (which we cite). This is true for many other image modalities as well. Our argument throughout the manuscript is not about another type of image, but rather about MOI hotspot/ coldspot contours that identify and visualize areas of MOI's significant increased/decreased relative spatial abundance. These may be subject to developer choices, which we validated as much as possible, but not so much to end user perception bias.

2.10.4. *I realise the authors want to make an argument for the hot/coldspot contours being user-independent, but given the number of developer-made choices in generating these contours, I do not consider the contours to be free of human bias either (see my remarks in the original review and this re-review). In short, this statement seems inaccurate towards ion images (not the interpretation of them), and seems meant to set up MPMs and CPPMs as not biased, which does not seem to bear out by the manuscript text that follows. Please explain or rewrite for accuracy.*

Again, we made a clearer distinction between developer-made choices, which can be tested and evaluated as we did and what an end user may do. More importantly, we have removed "unbiased" or "user-independent" from the manuscript.

2.11. Ion Images

- *"neither consider nonlinearities in the resolving power of mass spectrometers, nor do they yet evaluate statistical significance of potentially differential spatial metabolite distributions." -> An ion image, or any image for that matter, does not "consider" anything, it can only report a value. An image can also not "evaluate" anything, it can be used as a data source on which to base an evaluation, but the evaluation process is orthogonal to the content of the image. Since statistical significance of various properties of the data can be evaluated on the basis of observations in ion images, these aspects do not necessarily count against ion images. The manuscript seeks to make an argument that ion images are*

not optimal (which they are for certain purposes and not for others). However, the arguments made in this statement do not succeed in conveying that. Please rewrite for accuracy.

We have updated these statements in the revised manuscript. It is the processing of raw data into ion images, which does not take into account the aforementioned aspects.

[Reliability and statistical significance]

2.12. Reliability

- "improves signal reliability" -> This is stated as a given in the manuscript, but in the rebuttal is stated as an expectation: "However, we expect them to have less impact on the resulting image intensities of MPMs (because of the Gaussian mass-window weighting; comments 2.2.1 and 2.4) and on the hotspot/coldspot contours (because of the internal KDE smoothing; comment 2.2.2) when compared to state-of-the-art ion image procedures." I agree there is an opportunity for the method to improve reliability, but not necessarily in all cases and it remains somewhat dependent on the assumptions made by the developers working out for the dataset at hand. As such, stating in the manuscript the expectation to improve signal reliability seems more accurate than the general statement made here.

We appreciate the reviewer's message that he/she sees an opportunity for the method to improve reliability. We have changed the abstract to clarify that this is an expectation, but perhaps no certainty in all cases. The abstract now reads "moleculaR is expected to improve" instead of simply "improves".

2.13. Probabilistic Molecular Mapping

- "probabilistic molecular mapping" -> A variable that varies between 0 and 1 is not necessarily a probability. For that claim to be made, it would be necessary to show that the variable reports the actual chance/probability of an event happening. Since the manuscript does not seem to provide that, the variable between 0 and 1 can of course be used for applications, but it does not seem sufficient to be considered a probability. In short, "molecular mapping" seems to be a statement that is supported by the manuscript, while "probabilistic molecular mapping" does not seem to be supported. Please rewrite the text and naming for precision.

We thank the reviewer for this remark. The pixel intensities valued between 0 and 1 do not represent a probability, and we did not suggest that. The word "probabilistic" is meant to suggest that we introduced probability-based concepts to compute hotspots/coldspots, i.e. areas of increased relative spatial abundance or deficiency using the MPM method. These, probability-based contours are then mapped: Our study proposes a method that identifies intensities that would unlikely occur if these signals were generated by a random event or random spatial process. Internally, this process is based on null-hypothesis significance testing of each pixel intensity in $\rho_{MOI}(x, y)$ against a significance level α based on the $f_{CSR}(k)$ null distribution (**Fig. 2a**). The resulting MPMs are defined as composite representation of MOI (Gaussian-weighted) intensities, with MOI hotspots and/or coldspots superimposed as polygonal contours identifying areas of MOI significant relative spatial abundance and deficiency (i.e. areas unlikely to occur randomly), respectively. As such, we believe that the use of "probabilistic molecular mapping" (but **not** probability mapping) here is justified.

We realized that the statistical concept was perhaps not as clearly presented, as it should have. It was only briefly described in the methods and in one Figure legend. We have therefore added a new **Suppl. Fig. 9**, to illustrate the concept more clearly. We have also updated the method section and the main manuscript. We have also rescaled all intensity values in all figures showing ion images and MPMs throughout the manuscript and supplement to arbitrary intensity values between 0 and 10 to avoid misinterpretation.

An alternative terminology like “*molecular mapping*” would not do the innovative statistical evaluation and visualization tools justice, since, for instance, ion images are molecular maps, too. However, we would be open to suggestions for other, perhaps more fitting terminology.

2.14. Molecular Probability Maps

- “*molecular probability maps (MPMs)*” -> *It's not necessarily so that because MPMs provide a value between 0 and 1 that their content reports a probability. Please refrain from using the word "probability" in this context. A true probability would be inherent to the observed data, and should not change in function of parameter choices, yet different parameter choices by the user deliver different MPM content. Thus, it seems that MPMs as reported by the authors do not report an actual probability (maybe an estimation?). Please explain or rewrite the text and naming for precision.*

It is indeed not a probability on a 0,1 scale, and it was never intended to be. In line with our **Comment 2.13**, we have adopted “molecular probabilistic maps” instead of “molecular probability maps” throughout the manuscript. Again, in our view the extensive statistical testing done by *moleculaR* is based on a probabilistic concept (see new **Suppl. Fig. 9**). We have rescaled all intensity values in all figures showing ion images and MPMs throughout the manuscript and supplement to arbitrary intensity values between 0 and 10 to avoid misinterpretation.

2.15. High-mass Resolution MSI

- “*A note of caution: MPM contours can, in principle, be calculated for data from various MSI platforms, but it may be most meaningful for high-resolution MSI*” -> *If the MPM approach has conditions under which it does not perform, please elaborate. The manuscript makes strong statements about reliability and significance about the provided results. It would serve the understanding of the reader to understand under which circumstances the method can be relied upon and in which cases less so. Here, a specific example is given and mass resolution seems to be suggested as one parameter that influences meaningfulness. Please provide a discussion on these breakdown conditions in the main manuscript to accompany the statements on reliability and statistical significance there.*

We thank the reviewer for pointing this out. The quoted statement was **not** meant to indicate that MPMs are less suitable or generally break down when applied on lower mass-resolution MSI data. Instead, this statement referred to bespoke observations that the relatively lower-mass resolution MALDI-TOF-MSI data added in revision ($R \sim 5000$ is uncharacteristically low for modern instruments) was on several occasions not capable of distinguishing MOIs under study when compared to the higher-mass resolution MALDI-timsTOF and -FTICR MSI data as could be observed in **Suppl. Fig. 6-8** and **17**. This topic requires follow-up studies, since the present study focused on high resolution MSI that is typically employed in spatial metabolomics. The statement has been updated in the revised manuscript to state exactly that.

2.16. Addition of High-intensity Pixels

- Figure 2 -> The procedure for generating MPMs seems dependent on the distribution of ion intensity values present in the measured pixel locations. The generation of coldspot and hotspot contours furthermore relies on f_{CSR} . However, if we were to measure more pixels of this tissue sample (simply measuring a larger ROI in the tissue slice), the additional intensities would move f_{CSR} to a different value and the contours to different locations. For example, if the additional tissue area contains a lot of high-intensity pixels for the POI considered, then the f_{CSR} would end up more to the right on the 'Intensities' axis, the upper and lower cutoffs would presumably end up moving, and therefore the coldspot and hotspot contours would be drawn in different locations on the tissue. Is this correct? If so, how can coldspot and hotspot contours be labeled as "significant" areas to highlight if their location changes dependent on whether one collects more or less tissue area. This could line up with the adjective 'relative' in the notion "significant relative spatial abundance". However, it does not seem to line up with the adjective 'significant' since whether an area in tissue is considered significant should not be influenced by whether or not other tissue areas as measured as well? Please explain and describe in the manuscript. The simulated data that has been added to test the validity of the MPM approach does not test these aspects since it does not increase or decrease the content of the "imaged tissue" substantially and the content of that added tissue. Specifically, the different simulated datasets do not compare 'significant' contour location movement when substantial amounts of pixels with content patterns are added without changing the original pattern (which would be the equivalent of enlarging the recorded tissue to collect additional tissue structure). This also further complicates the stated need for cross-dataset comparability, which is made explicit by the authors when they say "MPMs enable spatially-aware cross-tissue comparison of tryptophan". The within-tissue impact would need to be addressed, and the across-tissue impact separately as well.

We thank the reviewer for raising this point. In our case, we measured the entire population that is available, i.e. the tissue. This is the typical case in MSI practice. Hypothetically, we could measure a larger area that perhaps could contain additional hotspots, but practically that is not possible. Nevertheless, we conducted additional simulation experiments to address this case: The ground-truth total area remained the same, but additional areas of high-intensity points were superimposed (**Suppl. Fig. 26**).

As a general reply preceding a more specific one, we would like to highlight that in statistics the expression "significant" itself is normally not an absolute term. Statistical significance helps to clarify if a result is likely due to random chance or to some factor of interest. Quantifying statistical significance involves taking a sample of some population of interest. However, the size of the sample and the variation in the underlying population both directly influence statistical significance through sampling error while the significance level α , a p-value threshold often set at 0.05, has been subject to increased criticism.¹¹ Please see **Comment 2.40** for a detailed discussion.

Before discussing this additional simulated example though, it is important to note that f_{CSR} is not obtained directly from the Gaussian-weighted pixel intensities (**Fig.1** and first row left of **Fig. 2a**) but rather represents the probability density function of intensities obtained from the KDE-smoothed spatial density function $\rho_{CSR}(x,y)$. In other words, f_{CSR} (**Fig. 2a-bottom left**) holds the intensities of $\rho_{CSR}(x,y)$ (**Fig. 2a-second row left**) but not intensities of the CSR model (i.e. "reshuffled" MOI intensities). This makes all the difference. Let $H_{MOI}(i)$ and $H_{CSR}(i)$ denote the histograms of

¹¹ McShane, Blakeley B., et al. "Abandon statistical significance." *The American Statistician* 73.sup1 (2019): 235-245.

intensities i obtained from SPP_{MOI} and CSR_{MOI} , respectively, which are of course identical since $H_{CSR}(i)$ is a spatially “reshuffled” version of $H_{MOI}(i)$. Including more tissue area, presumably with a lot of high-intensity pixels will increase the area of the tissue window Φ_{tissue} and, of course, impact H_{MOI} and, hence, H_{CSR} . However, these new high-intensity pixels will be randomly reassigned (based on a homogeneous Poisson process) to new positions (i.e. new spatial context with point neighbourhood of varying intensities) within an increased tissue window area. Then, KDE is applied, and being a low-pass filter, as rightfully noted by the reviewer in the previous revision, it caps abrupt spatial changes in intensities low and high alike. This can be observed in the examples provided for f_{CSR} in **Suppl. Fig. 12b**. There, f_{CSR} converges to a normal distribution effectively smoothing out high-frequency spatial changes in f_{CSR} but not so much in f_{MOI} since the intensities in $\rho_{MOI}(x, y)$, unlike $\rho_{CSR}(x, y)$ have a spatial tendency to them (i.e. spatially autocorrelated).

In order to provide further evidence to the above explanation, we tested MPMs for stability against simulated data when external secondary areas of high-intensity points were spotted into the simulated spatial point pattern (SPP) either cumulatively or iteratively. For cumulative addition of high-intensity areas to the ground truth, the computed MOI hotspot contours of the tested ground truth element (green filled central area) were rather not affected (**Suppl. Fig. 26a**). Additionally, 100 secondary areas of high intensity with varying radii were spotted iteratively at varying positions randomly assigned around the green area (**Suppl. Fig. 26b**). At each iteration, MPM and DSC values were computed as in the previous step. Similarly, the estimated hotspot contours of the main simulated hotspot were rather not affected as indicted by the box/violin-plot of the computed DSC values. Based on the preceding explanation and the provided new **Suppl. Fig. 26**, we do not expect that changes in measurement area or addition of high-intensity pixels would have a major effect on the estimated hotspot/coldspot contours.

2.17. Reference Tissue

- "Here, the CSR model of the test tissue is inferred from signal intensities of the reference tissue." -> Please elaborate on the validity of this experimental design. Unless there are guarantees around the intensity distributions of two datasets being perfectly matched between them (both in number and intensity values), it does not seem valid to use the intensity values of one dataset to judge the significance of the relative spatial abundance of the measured distribution in another dataset (since relative spatial abundance has been presented as a within-tissue evaluation up to this point in the manuscript). A more detailed description of this question can be found in my remark for (xx).

- (xx) "it is important to ensure that the signal intensities of both test and reference tissues are comparable by observing appropriate experimental design which deliberately minimizes technical variation (ex. placing them on the same slide to be measured in a single measurement) 6 and/or by relying on robust intensity normalization methods." -> This experimental design seems to remain flawed despite the condition formulated here. By varying the spatial location of intensities, but keeping the intensity values themselves, one can assess whether the spatial distribution is random or not. That seems to be the classical use of a CSR null hypothesis. If, on top of changing the spatial locations, one also uses other intensity values (such as from another dataset), then the reference being compared to says little about spatial distribution anymore and whether that spatial distribution is 'significant' cannot be assessed. Even if the biological intensities between the datasets are somehow matched as the authors describe here, the content of the tissue will be different (e.g. since one dataset has a tumour in it). This seems to invalidate the use of the reference intensities for the mutated CSR. Please explain in the manuscript why this experimental design remains valid.

We thank the reviewer for his/her important remark. We have revised the proposed cross-tissue testing method, updated the associated figures (**Fig. 3e** and **Suppl. Fig. 18**) and provided more detail in **Methods** (new section **Cross-tissue MPM; CT-MPM**) and new Supplementary figures (**Suppl. Fig. 19** and **Suppl. Fig. 20**) that illustrate the idea behind it. We also added new **Suppl. Fig. 2** to make it clearer that CT-MPM are an additional concept that builds on canonical MPMs. CT-MPMs are even stricter than MPMs, since a second test (and null hypothesis) is introduced. Briefly, in a first step, MPMs are computed for the test tissue. Hence, relative spatial abundance remains a within-tissue evaluation. We agree with the reviewer that comparison of the spatial aspect (i.e. autocorrelation) with that of another tissue would not make sense.

For a more comprehensive answer: CT-MPMs are considered useful in situations where there is need for comparing the intensity distributions of metabolite or drug signals between test and reference tissues, e.g. those dosed with a drug or carrying certain mutations versus controls. Here, CT-MPMs enable spatial statistical testing where normally only signal intensities are used for statistical comparisons (e.g., box/violin plots of **Fig. 3e**) that disregard the spatial localizations of MOIs under study. Consequently, we do not just ask if there were hotspots in the test tissue. In addition ask if **i)** the intensity distribution of the test tissue is significantly different from the one of a reference tissue and **ii)** which pixel locations in the test tissue carry intensities that are unlikely to belong to the distribution of reference tissue pixel intensities. The reasoning is that while a test tissue may display hotspots (=areas of increased spatial abundance), these may not be a feature that distinguishes, for instance, a tumor- from a non-tumor sample, if their respective intensity distributions were virtually indistinguishable (see **Suppl. Fig. 19c**). We want to emphasize that CT-MPMs are about an additional rejection criterion that evaluates a non-spatial aspect.

This is achieved by first finding areas of significant relative spatial abundance/deficiency (testing against the spatial null hypothesis; “within-tissue” MPM) as shown in **Fig. 2a**. Then the (distribution-free) empirical cumulative distribution function is inferred from the signal intensities of the reference tissue and all signal intensities of the test tissue are tested against it in order to find the likelihood of them (i.e. signal intensities of the test tissue) being drawn from the signal distribution of the reference tissue. Similar to the “within-tissue” MPM method described above, the p-value threshold, beyond which the null hypothesis is rejected, is set to a significance level $\alpha = 0.05$, and Benjamini-Hochberg correction is applied to account for the inherent multiple testing problem. Only test tissue intensities, which reject both the spatial (within test-tissue), and Test-vs-Reference intensity distributions null hypotheses are designated as having significant cross-tissue relative abundance/deficiency (**Fig. 3e**, **Suppl. Fig. 18**, new **Suppl. Fig. 19** and new **Suppl. Fig. 20**).

As is the case with classical drug/metabolite intensity distribution testing of Test-vs-Reference tissues, which is commonly encountered in the MSI literature, it is indeed important to ensure that the signal intensities of both test and reference tissues are comparable. This requires observing appropriate experimental design, which deliberately minimizes technical variation (ex. placing them on the same slide to be measured in a single measurement)¹² and/or by relying on robust intensity normalization

¹² Balluff, B., Hopf, C., Porta Siegel, T., Grabsch, H. I. & Heeren, R. M. A. Batch Effects in MALDI Mass Spectrometry Imaging. *J Am Soc Mass Spectrom* **32**, 628-635, doi:10.1021/jasms.0c00393 (2021).

methods.^{13,14} To give the user a more complete picture, *moleculaR* will generate a boxplot comparison between the Test and Reference intensities with a statistical significance test (Mann-Whitney) superimposed in addition to the resulting CT-MPM.

We also added another example that uses CT-MPM to spatially localize areas of significant cross-tissue relative abundance of imatinib in a gastrointestinal stromal tumor (GIST) tissue sample when compared against a series of imatinib dilution spots in MALDI-TOF-MSI data from a previous study¹⁵ (new **Suppl. Fig. 20**). There, the reported mean imatinib content in that sample was 7.78 pmol (95% CI 7.28, 8.46 pmol) and 7.81 pmol (95% CI 7.63, 7.99 pmol) based on MALDI-TOF-MSI and UPLC-ESI-QTOF-MS quantification, respectively. Consecutive comparison of the imatinib-tissue content against four imatinib dilution spots (3.125, 6.25, 12.5 and 25 pmol) show a gradual decrease in the number of pixel-intensities detected as significant cross-tissue relative abundance of imatinib (hotspot contours of **Suppl. Fig. 20e-h**). Few, possibly outlier, intensities were detected when compared against the dilution spot of 12.5 pmol (**Suppl. Fig. 20g**) while no points were detected for the cross-tissue MPM against the dilution spot of 25 pmol (**Suppl. Fig. 20h**), which is way above the reported mean imatinib-tissue content. The cross-tissue test carried out against the imatinib dilution spot of 6.25 pmol, i.e. the closest to the reported mean imatinib-tissue content, revealed that the areas with significant cross-tissue relative abundance of imatinib (hotspot contours in **Suppl. Fig. 20f**) were spatially restricted and appear to coincide with the high-intensity pixels perceived on the imatinib intensity image of **Suppl. Fig. 20d**. Overall, we think that the above findings support the validity and applicability of the now clarified CT-MPM method. Detailed description has also been provided in the revised manuscript.

2.18. Intensity Artifacts

- "added to $n=10$ randomly selected pixels" -> *If the test is to assess method robustness, a more noisy example with higher n seems appropriate. The goal of a robustness assessment is to find out at what point the method starts breaking down. If that point is far from the natural MSI datasets one encounters, the method is (relatively) robust. If that point is easily reached, the method does not seem very robust. This simulated example comes across as not disruptive enough to find out where the method breaks.*

We thank the reviewer for this important remark. In our experience, intensity artifacts mentioned in the submitted manuscript appear as few single pixels of extremely high intensities (relative to the within-measurement intensities) in MSI measurements. These could result from tissue gaps (ex. tears), inhomogeneous matrix crystal spatial distribution, ion source contamination or abrupt chemical inhomogeneities. Unfortunately, we did not find any MSI literature that provided experimental data on this phenomenon. Therefore, we carried out another test where the number of added intensity artifacts (n) was varied between 10 and 5000 signals ($\approx 20\%$ of the total tissue pixels) randomly sampled and randomly placed within the tissue window (new **Suppl. Fig. 24**). Dice similarity coefficient (DSC) was computed between the estimated MPM hotspot contour of the raw and artificially contaminated data (black solid curve of **Suppl. Fig. 24b**) while also computing the normalized cross correlation (NCC) image similarity metric between the raw and artificially contaminated images (red

¹³ Boskamp, T. *et al.* Cross-Normalization of MALDI Mass Spectrometry Imaging Data Improves Site-to-Site Reproducibility. *Anal Chem* **93**, 10584-10592, doi:10.1021/acs.analchem.1c01792 (2021).

¹⁴ Veselkov, K. *et al.* BASIS: High-performance bioinformatics platform for processing of large-scale mass spectrometry imaging data in chemically augmented histology. *Sci Rep* **8**, 4053, doi:10.1038/s41598-018-22499-z (2018).

¹⁵ Abu Sammour, Denis, *et al.* "Quantitative mass spectrometry imaging reveals mutation status-independent lack of imatinib in liver metastases of gastrointestinal stromal tumors." *Scientific reports* 9.1 (2019): 1-9.

solid curve of **Suppl. Fig. 24b**). If we consider a DSC of 0.75 as a breakdown-threshold, then the method can withstand up to an (extremely unrealistic) $n = 450$ artificially added intensity artefacts. Please note that the NCC metric, often employed for quantifying similarity between images (here between the raw and “noise-contaminated” ion images), crosses the 0.5 mark at $n = 450$.

2.19. Arithmetic Operations

- *"CPPMs enable basic arithmetic manipulations on SPPs of multiple MOIs" -> Since the ion intensity values have a certain uncertainty around their measured value, and arithmetic operations on uncertain values can further extend the uncertainty around their end product, how should the uncertainty around the CPPM result be assessed? The propagation of uncertainty is not only dependent on the number of input variables, but also the particular arithmetic operations used on them. How should readers interpret this aspect of CPPMs for their own cases? Figure 5 shows several examples of division-based CPPMs, where two uncertain variables get divided by each other, which is an operation particularly prone to uncertainty amplification. Please provide guidance for users of *moleculaR* in the manuscript.*

We thank the reviewer for this important remark. It is widely accepted in the MSI field that ion intensities in MSI are accompanied by substantial uncertainty. In our recent review ¹⁶, a thorough discussion was presented on the fact that biological information is concealed by technical variation arising from various sources of the MALDI-MSI process and several suggestions have been proposed to limit such adverse effects. However, these uncertainties are unfortunately not completely avoidable. Inspired by recent studies which employ image ratios and composite scores to study metabolites and their spatial relations ^{17,18}, we attempted to provide the users of *moleculaR* with such computational tools. A quantitative assessment of this uncertainty and its propagation, which would require extensive orthogonal measurements on an entire tissue cohort and ideally a ground-truth scenario, has not been addressed in this study. nevertheless, whereas arithmetic operations always add (relative or absolute) uncertainty because of error propagation, we would expect them to have less impact on the resulting image intensities of MPMs/CPPMs (because of the Gaussian mass-window weighting and the centroided nature of the input data, which ensures that near-zero values do not land in the denominator) and on the hotspot/coldspot contours (because of the internal KDE smoothing) when compared to state-of-the-art ion image procedures. A word of caution has been added as guidance for *moleculaR* users to the revised Methods and in various parts of the main text.

2.20. FWHM Estimation – One Spectrum

- *"One pixel representing a continuous (profile) mass spectrum was randomly chosen from the corresponding profile data and exported as a CSV file via flexImaging software version 5.0 (Bruker Daltonics)." -> What if the one randomly chose pixel is not representative (enough) for the rest of the dataset? The mentioned elimination of <1% occurring peaks seems to address one possible form of being not representative, but introduces an additional percentage threshold to tune. Additionally, the spectrum of a single pixel is noisier than an average spectrum, which would suggest that the detected peaks in that spectrum and their estimated FWHM could be more noise sensitive. Is this the case? Does*

¹⁶ Balluff, Benjamin, et al. "Batch effects in MALDI mass spectrometry imaging." *Journal of the American Society for Mass Spectrometry* 32.3 (2021): 628-635.

¹⁷ Sun, Chenglong, et al. "Spatially resolved metabolomics to discover tumor-associated metabolic alterations." *Proceedings of the National Academy of Sciences* 116.1 (2019): 52-57.

¹⁸ Fack, Fred, et al. "Altered metabolic landscape in IDH-mutant gliomas affects phospholipid, energy, and oxidative stress pathways." *EMBO molecular medicine* 9.12 (2017): 1681-1695.

this have an effect on the quality of the FWHM estimation? If not, why not? If yes, how should this problem be addressed by a user of the software?

We'd like to refer the reviewer to our answer to **Comment 2.5.1**.

2.21. Knee-point

- "being a low-pass filter" and "This is the point, after which an increase in hKDE does not result in a considerable increase in the spatial autocorrelation of the smoothed density image." -> I agree that the KDE functions as a spatial low-pass filter. This means that it introduces spatial autocorrelation into the pattern it outputs, with the larger the hKDE used, the larger the spatial autocorrelation introduced (because short distance variation is wiped out). As expected, an increase in hKDE tends to result in an increase in spatial autocorrelation in the output pattern. However, the manuscript seems to make an argument that the knee point in that profile informs about the natural autocorrelation present in the measured ion distribution. Please explain why that knee point is not the place where so much small variation has been wiped out by the low-pass filter that no substantial additional autocorrelation can be observed? What is the reason for the knee point to be linked to the natural autocorrelation?

Spatial autocorrelation measures the degree of spatial co-dependency of observations (pixels or points) existing in a defined neighborhood size within a spatial window. The idea behind this step-wise evaluation of autocorrelation as a function of smoothing comes from scale-space theory¹⁹ in computer vision where an image is represented by a one-parameter (i.e. the smoothing scale) family of smoothed images. Since *moleculaR* implicitly uses kernel density estimation (KDE) for reasons discussed thoroughly in **Comment 2.8-Revision1 (Theoretical considerations section)**, estimating an optimal data-dependent smoothing scale (bandwidth) is of high importance, which, on its own, is a well-known challenge.²⁰ We reasoned that, in a Moran's I vs smoothing bandwidth plot, the point at which Moran's I statistic rate of change abruptly falls, is the scale at which exactly those random pixel fluctuations or "small variations" mentioned by the reviewer are "wiped out" or smoothed away and important spatial structures/features/patterns start dominating the spatial landscape. This reasoning seems to hold true when MPM results were tested against various types of artificially added noise. Here, we think that exactly this approach of filtering out "small variations" (which, for reasons provided in **Comments 2.32 and 2.35**, could be representative of small random noise-like fluctuations) and focusing on informative spatial structures makes this method robust in various aspects (pixel-to-pixel, section-to-section and slide-to-slide batch effects²¹; **Comment 2.2-Revision1**). Having said that we now see that using the words "inherent autocorrelation of MSI measurement" could be misleading, we have therefore, updated the text in the revised manuscript to include the description above.

2.22. Uncertainty

- "Even though Gaussian weighting does improve signal reliability, each ion intensity image will still contain an unknown amount of non-biological technical variability 6, which could be carried on to the composite image representation (error propagation)." -> This statement seems very important and should be made more prominently in the main manuscript, not only the methods section, since it impacts

¹⁹ Lindeberg, Tony. *Scale-space theory in computer vision*. Vol. 256. Springer Science & Business Media, 2013.

²⁰ Heidenreich, Nils-Bastian, Anja Schindler, and Stefan Sperlich. "Bandwidth selection for kernel density estimation: a review of fully automatic selectors." *AStA Advances in Statistical Analysis* 97.4 (2013): 403-433.

²¹ Balluff, Benjamin, et al. "Batch effects in MALDI mass spectrometry imaging." *Journal of the American Society for Mass Spectrometry* 32.3 (2021): 628-635.

the composite image representation's use case. The figures in the main manuscript do not show measures of uncertainty (which I realize are hard to obtain), but that uncertainty is acknowledged to be there both in molecularR's results and in the original ion images. How should a user of this software's results interpret this, particularly since the reliability of the methods is mentioned explicitly in the text as advantages of molecularR over ion images?

As in the case of ion images, *molecularR* does its calculations based on ion/signal intensities generated by the MALDI ionization process, which invariably adds technical variability to biology-related variability. Hence, *molecularR* assumes the presence of both. As discussed before and above, our main argument about reliability relates to the Gaussian down weighting of neighboring signals. Even though the presented manuscript is not focused on quantifying or describing analytical uncertainty and error propagation in MSI, we acknowledge the impact of error propagation in the revised manuscript in **Methods** but also in the main text within the context of composite scores. We would like to emphasize, however, that the main innovation of the manuscript are the probabilistic concepts. Throughout the revised manuscript, we have taken out notions of bias versus non-bias, in order to present these innovative concepts as a complementary option for the MSI community.

2.23. MALDI-TOF-based Synthetic Data

- "were sampled from above and below the upper quartile of the empirical intensities of a MALDI-FTICR-MSI measurement of a human GB tissue sample" -> The signal to noise ratio of MALDI-FTICR data is commonly very good, and typically better than MALDI-TOF or other analyzer types. Using the intensities of MALDI-FTICR experiments seems beneficial to pass the test and seems not necessarily representative for the data types this software is suggested to be used on. Please do a test on a more challenging synthetic data set that uses lower signal to noise ratio measurements to sample from. While being shown to pass a high signal to noise test is ok, the software would be helped by being shown to work on a 'hard' noisy example.

We thank the reviewer for this important remark. Just to be clear: The software was developed using high-resolution MSI data. We added a MALDI-TOF example upon request during peer review, and we are not suggesting that the software should be used on this data. As outlined above and in the manuscript, there is not enough data in hand to firmly rule out MALDI-TOF data for use with *molecularR*. In fact, we do provide some data to suggest that the MPM workflow can be useful for some MALDI-TOF work. We discuss this and ask for caution.

Nevertheless, we have created an additional **Suppl. Fig. 14**, which illustrates the MPM workflow tested on simulated SPPs with their intensities sampled for above and below the upper quartile of the empirical intensities of a MALDI-TOF-MSI measurement of a human gastrointestinal stromal tumor (GIST) tissue sample at 494.2662 m/z (imatinib $[M+H]^+$; $SNR \geq 3$) for the simulated hotspot and background areas, respectively. The MPM method was able to identify points exhibiting significant relative spatial abundance within the simulated hotspots (green points on density and surface plots in **Suppl. Fig. 14a-d**). This example may serve as a case in point that there may be use cases for *molecularR* use with MALDI-TOF data.

[Precision in definition and wording]

2.24. Differential Spatial Distribution

- "nor do they yet evaluate statistical significance of potentially differential spatial metabolite distributions." -> The data property whose statistical significance is evaluated here is stated as "potentially differential spatial metabolite distributions". This seems to be different from the property being used elsewhere in the text, namely "relative spatial abundance". Is it the same, or is this a different property? Please explain and provide a clear definition of the property whose statistical significance is being evaluated. In this concept, how are spatial location and ion intensity combined to make relative spatial abundance? Please provide a formula or some precise statement on what is considered "relative spatial abundance" and what not?

We thank the reviewer for this remark. We have updated the wording to be consistent throughout the revised manuscript. Clarification of what this statement implies has been provided in various parts in the manuscript. It is now also more clearly written in **Methods**.

2.25. Ion Images

- "ion images, neglect mass accuracy and the resolving power and instrument-dependent peak width at peak-of-interest (POI) m/z " -> This statement seems overly generalising. An ion image can be generated while ignoring mass accuracy, resolving power, and peak width, but this does not have to be the case. There are plenty of examples where ion images are generated without ignoring these aspects. Neglecting these aspects is a property of the study and its experimental design, not a general disadvantage of ion images. Additionally, if ignored, such aspects could also leave a trace in the MPMs and CPPMs generated by *moleculaR* it seems (since *moleculaR* works from the centroided data and the ignoring can happen upstream from *moleculaR*). Please rewrite for accuracy.

- "ion images, <...> typically use the sum of ion intensities of all peaks present in a user-defined mass range centered on the POI m/z instead in a data-independent uniform weighting (i.e. all peaks treated equally) approach" -> Same remark as for "neglect mass accuracy".

- "to compute ion intensities, all peaks in a user-defined mass range centering on a POI are weighted equally and simply summed up" -> Same remark as for "neglect mass accuracy".

We agree with the reviewer's statement that ignoring these important aspects (mass accuracy, resolving power, etc.) is not a property of ion images themselves. However, the important distinction here is that, out-of-the-box, ion image rendering typically does not provide the means or methods to take these important aspects into consideration. For the calculation of ion image intensities, typically the user defines the extent of the mass-window where (peak or profile) signals are to be **integrated** either as an absolute value in Daltons or a relative value in ppm (relative to m/z MOI). This could be seen in all major software platforms dedicated to analysis and visualization of MSI data (ex. FlexImaging, SCiLS LAB; Bruker Daltonics) as well bioinformatics packages (ex. Cardinal, MALDIquant). Previous studies^{22,23} have also reported this type of ion image generation, which is considered the norm within the MSI community.

In contrast, *moleculaR* is built with these aspects in mind. For example, *moleculaR* will not provide images or analysis if the user fails to provide at least one continuous spectrum for estimation of

²² Palmer, Andrew, et al. "Using collective expert judgements to evaluate quality measures of mass spectrometry images." *Bioinformatics* 31.12 (2015): i375-i384.

²³ Race, Alan M., and Josephine Bunch. "Optimisation of colour schemes to accurately display mass spectrometry imaging data based on human colour perception." *Analytical and bioanalytical chemistry* 407.8 (2015): 2047-2054.

FWHM, which is important for the calculation of both, mass-window width and Gaussian mass-window weighting. We have updated the corresponding statement for more precision in the revised manuscript.

2.26. Interference Noise

- "interference noise" -> *What is the definition of 'interference noise'? Which physical process in mass spectrometry is modeled by it? Fig 4a suggests this to be uniformly distributed intensity values, but in a specific range from $\max(f_{MOI})$ to $5\max(f_{MOI})$ (in green). Why this intensity range exactly? Why not the full intensity range? Please motivate why this is an appropriate disturbance choice to demonstrate robustness against.*

We thank the reviewer for this remark. There, unfortunately, was a mistake in the description of **Fig. 4**, which we have updated, in the revised manuscript. Interference noise is defined as an artificially added overlapping peak placed arbitrarily $2\sigma_G$ away from MOI m/z (bottom left panel of **Fig. 4b**) whose intensities are sampled from a Gaussian distribution (orange distribution of **Fig. 4a**) whose mean and standard deviation are equal to the mean and standard deviation of the MOI intensities. This type of noise is used to model the presence of a proximal interfering background peak (e.g., matrix peak or another metabolite) in the vicinity of MOI m/z . The noise type that is sampled from the mentioned uniform distribution (green area in **Fig. 4a**) is the so-called intensity artifacts, which has been discussed in **Comment 2.18** of this revision. From our experience, these intensity artifacts appear as few single pixels of extremely high intensities (relative to the within-measurement intensities) in MSI measurements. These could be attributed to tissue gaps (ex. tears), inhomogeneous matrix crystal spatial distribution, ion source contamination or abrupt chemical inhomogeneities in microenvironments. Unfortunately, we did not find any MSI literature that precisely describe these artifacts from a data-focused perspective but, based on our experience, these artifacts tend to occur rarely and randomly showing up in just a few pixels.

2.27. Noise mean and standard deviation

- "*Mean and standard deviation of $f_{Gaussian}(k)$ are equal to those of $f_{MOI}(k)$ " -> Please elaborate on why making the noise be of equal standard deviation as the signal is an appropriate demonstration of robustness? This choice seems to mimic a signal to noise ratio that is close to one, which is not necessarily common in mass spectrometry. Is that the intention?*

In **Fig. 4a**, $f_{Gaussian}(k)$ represents an intensity distribution from which the intensities of the artificial Gaussian and interference noise types are sampled. As rightfully noted by the reviewer, spiking noise with intensities sampled from a Gaussian distribution that has the same mean and standard deviation as the intensities of MOI m/z is rather an extreme case, which is not common in MSI, but which we used to show the robustness of the proposed method. We have shown (**Fig. 4c**, **Suppl. Fig. 22-24**) that despite this extreme case, MPMs are still capable of identifying hotspots/coldspots when compared to the artificially uncontaminated version of the data. Even more (positively) striking is how MPM hotspot contours are able to withstand Gaussian noise with σ_n up to 4 times σ_{MOI} retaining hotspot contours overlap of above 0.75 DSC, all this while observing clear visual degradation of the corresponding ion images which is also mirrored in their computed normalized cross correlation (NCC) relative to the raw ion image (**Suppl. Fig. 22**).

2.28. CSR Points Locations

- *"While MOI's SPPs inherit the gridded spatial locations from MSI data, creating a CSR model on a spatial grid would directly violate the randomness criterion of such models." -> This statement seems internally inconsistent with statements elsewhere in the manuscript. In this sentence the case is made that conversion from a grid to a SPP is necessary to compare against a CSR, which is a benchmark where the locations of events are randomly distributed in the spatial plane. However, this does not seem to be the definition of CSR used elsewhere in the manuscript. In Figure 2a, the CSR is depicted and it seems that it is not the measurement locations in the spatial plane that are randomly distributed, but rather the content of the measurements while keeping the measurement locations in the same regular grid. Therefore, the CSR in Fig. 2a seems content-wise randomly distributed, not randomly distributed in terms of space (the latter being the classical definition of a CSR). This interpretation seems to be confirmed by the following statement by the authors: "For simplicity, random sampling is replaced by randomly permuting the intensities, which basically has the effect of spatial reshuffling of points, until they assume a spatial uniform Poisson process, thus effectively dissolving any spatial clustering or autocorrelation of signals (Fig.2A)." The authors state that spatially random sampling is replaced by random permuting of intensities, and the latter could technically be done directly on a regular measurement grid representation as well. Therefore, it seems like the CSR (at least the implementation used in the manuscript) is no inherent reason to move to SPPs in the first place. I appreciate the SPP view on MSI data, but this section does not seem to state a hard reason for requiring SPPs. Please explain if I misunderstood, or formulate alternative reasons in the manuscript for requiring SPPs.*

We thank the reviewer for this remark. We now see that the corresponding statement in **Methods** might have been misleading. Two mechanisms are at work for creating the corresponding CSR model of a given MOI; **i**) creation of an equivalent SPP within the same tissue window Φ_{tissue} and same spatial point density Λ but which assumes a uniform Poisson spatial point process detached from the MSI coordinate grid (i.e. points are freely placed within Φ_{tissue}), then **ii**) points intensities of the SPP_{MOI} are randomly permuted and assigned to the points created in **i**. These two steps are equivalent to a spatial reshuffling of the points of SPP_{MOI} to create CSR_{MOI} . We have updated that statement in **Methods** for a more precise description.

[Language and typos]

2.29. Neither .., nor ..

- *"neither consider" -> There is no plural set of subjects in this sentence. Please rewrite for clarity.*

Please note that in the mentioned sentence "neither" is followed by a "nor".

2.30. Exemplary

- *"It thereby exemplarily fosters" -> The word "exemplary" seems incorrect here? Please explain how this approach does a very good job (= exemplary) at fostering the spatially-resolved investigation of ion milieus? That sort of relative performance conclusion seems to require benchmarking against other methods or the ground truth, which does not seem to be provided in the manuscript? See also "two exemplary MOIs" where the word seems to be in the same unsupported fashion.*

We thank the reviewer for this remark. Both “exemplarily” and “exemplary” were used to indicate “serving as an example, instance, or illustration”²⁴. To avoid confusion we have dropped/replaced these words.

[Remarks on the rebuttal answers]

2.31. FWHM

- *"The relation between FWHM and m/z (or m) is known in the field, especially for FTICR MS, but it is currently not used in data analysis." -> See my remark on the difference between the theoretical FWHM and the FWHM that can be practically retrieved and estimated from measurements.*

Well noted. Please refer to the discussion provided in **Comment 2.9**.

2.32. KDE and Small Variations

- *"The resulting curves suggest that the KDE bandwidth estimation method of MPM coincided with the highest (i.e. the optimum) DSC for all simulated SPPs. This in itself was an unexpected finding that underlines the strength of this approach." -> The simulated data is in terms of spatial structure very simple, with very little small distance variation to begin with. When applying KDE (and therefore a low-pass filter) on these spatial patterns, small distance variation is increasingly wiped out as h_{KDE} grows. However, if the patterns have little small distance variation to begin with (as is the case here), it is not surprising that the wiping out of small distance variation has little effect on the comparison between the ground truth pattern and the spatially smoothed pattern. The concern of Reviewer 1 seems to be the wiping out of biologically real but small distance patterns, and that is not tested by these examples. Furthermore, the matter of intensity used also plays a role, since small distance patterns with high intensity content will resist better against being wiped out by the KDE, than small distance patterns with low intensity content.*

We thank the reviewer for this remark. The above statement appears in **Comment 1.5-Revision1**. As discussed in **Comment 2.21** and later in **Comment 2.35**, the estimated knee point is indeed the point where small intensity fluctuations has been smoothed away. Here it is important to define the scale of such “small variations”. The reviewer seems to be concerned that such a method would completely wipe out important biological patterns during the iterative smoothing process. However, it is known from scale space theory²⁵, that when convolving a Gaussian kernel with an image, only image structures much smaller than the scale parameter (i.e. h_{KDE} of the Gaussian kernel) will be largely smoothed away in the process. Considering that, the computed h_{KDE} for all cases shown in the manuscript was between 2 and 2.6 pixels, structures that could be smoothed away are normally much smaller than 2 pixels. Moreover, as rightfully noted by the reviewer, structures with cross-sections smaller than this but with high intensity (indicating high SNR or analyte abundance) will largely resist this smoothing and would still be seen as hotspots. The test case of the APP NL-G-F Alzheimer’s disease mouse model brain (**Fig. 3d**) measured in linear mode MALDI-TOF (at unusually low SNR) provides a clear example of such a situation; sparsely distributed and fine amyloid plaques $A\beta$ 1-38 (m/z 4060.5), even before the proposed bandwidth tuning of **Suppl. Fig. 15**, were detected as hotspots.

²⁴ <https://www.merriam-webster.com/dictionary/exemplarily>

²⁵ Lindeberg, Tony. *Scale-space theory in computer vision*. Vol. 256. Springer Science & Business Media, 2013.

2.33. Further Investigation

- *"For investigating this further, additional computational experimentation would be needed that involved multiple slides or measurements where robust intensity normalization becomes crucial. This, however, is beyond the scope of the proposed method, as it would require extensive studies of entire (patient) cohorts of samples, which were not available here." -> Understood, but please make a clearer statement in the manuscript that this is a use case that is not (yet) validated and verified.*

We thank the reviewer for this remark. The above statement appears in **Comment 1.7-Revision1**. We have provided a note of caution in multiple locations in the revised manuscript that in case of cross-tissue MPMs, it is important to ensure that the signal intensities of both test and reference tissues are comparable by observing appropriate experimental design which deliberately minimizes technical variation (ex. placing them on the same slide to be measured in a single measurement) and/or by relying on robust intensity normalization methods. We have also included a statement that this type of evaluation requires further investigation on cohorts of tissues measured on multiple slides.

2.34. Smoothing Effect

- *"Therefore, the smoothing the reviewer refers to does NOT eliminate signals but is only an internal mechanism that directly affects estimated hotspots/coldspot areas." -> Understood, but that does not mean that the smoothing has no effect on the results (= where a hot/coldspot's contours are drawn) and the fact that these move depending on the h_{KDE} chosen. This puts the statistical significance argument a bit in question.*

We thank the reviewer for this remark. The above statement appears in **Comment 2.1-Revision1** which addressed the reviewer's concern that there might be *"the possibility for biological patterns to get substantially modified or removed in the process (MPM)"*. There, we argued that smoothing is an MPM-internal process and is not reflecting on resulting intensities in visualized SPPs/MPMs, thus ensuring that sparsely distributed signals including single-pixel signals are not altered (after applying the Gaussian mass-window weighting) or removed.

While it is true that h_{KDE} has a direct effect on the created hotspot/coldspot contours, two important internal mechanisms limit this effect. First, the proposed KDE bandwidth estimation serves as a stopping mechanism that prevents spatial over-smoothing as shown in **Suppl. Fig. 10** and **11**. Second, the same KDE procedure and thus the same degree of smoothing is applied for both SPP_{MOI} and CSR_{MOI} . This causes the CSR_{MOI} intensity distribution to shrink with increasing KDE bandwidth, as observed in **Suppl. Fig. 12b**. This means that while pixel intensities (of secondary as well as primary hotspots) within the MPM-internal scale-space representation are reduced with increased smoothing bandwidth, the quantile intensity cutoffs, which are used as thresholds for hotspots and coldspots, do also shrink proportionally (**Suppl. Fig. 12b**).

In line with **Comments 2.16** and **2.40**, we would also like to draw attention to the fact that the expression "statistically significant" itself is normally not an absolute term. This is because, generally speaking, quantifying statistical significance involves taking a sample of some population of interest. The size of the sample and the variation in the underlying population both directly influence statistical

significance through sampling error while the significance level α , a p-value threshold traditionally set at 0.05, has been subject to increased criticism.²⁶

Please, note that the word “significant” comes from a standard statistical test, which is shown in **Fig. 2a** and **Suppl. Fig. 9a** and described in detail in **Methods** (section **Molecular Probabilistic Map**), which is based on null-hypothesis significance testing of each pixel intensity in $\rho_{MOI}(x, y)$ against a significance level α based on the $f_{CSR}(k)$ null distribution, which is Benjamini-Hochberg corrected. We acknowledge that the resulting $f_{CSR}(k)$ depends on the estimated KDE bandwidth. However, the validity of the KDE bandwidth estimation has been thoroughly discussed in **Comments 2.16, 2.21, 2.32, 2.35** and **2.39**. Simulation-based evidence suggests that the proposed KDE bandwidth estimation results in an optimum h_{KDE} (**Suppl. Fig. 11; Comments 1.5-Revision1** and **2.8-Revision1**) showing accurate assignment of hotspots when compared to ground truth (**Fig. 2b, Suppl. Fig. 13 and 14**). Robustness analysis against artificially added noise provides further evidence of the stability of the hotspot/coldspot contours computed based on the proposed KDE estimation (**Fig. 4, Suppl. Fig. 21-24**).

Having stated the above, we still acknowledge that indeed the KDE bandwidth-dependence might be seen as a limitation to the proposed method resulting in an uncertainty that would require a MALDI-MSI-based phantom sample with controlled molecular content to quantify, which, unfortunately, has not yet been developed within the MSI community. It should be noted that other statistical concepts in MS-based omics like target-decoy searching in proteomics have been very successful and influential despite their known limitations²⁷.

2.35. Smoothing Small Variations

- "Since smoothing SPPs (and CSRs) with high KDE bandwidths could result in underappreciating fine spatial structures or secondary MPM hotspots, this effect must be balanced against the desired effect of improved signal reliability." -> This smoothing effect is further complicated by the particular intensity values of the POI. The balancing is therefore not only a matter of where but also how much. The Kneedle approach makes the hKDE selection automated, but it does not solve the fundamental issue of what small distance variation is ok to be removed from the significance calculation and whether that's an ok thing to do given what conclusions are going to be drawn from molecular's results. Automation of the hKDE setting does not solve the fundamental issue, it only removes the decision from the user's view.

In principle, we agree with this remark. As pointed out in **Comment 2.32 and 2.34** above, we do acknowledge some limitation of the KDE process. However, we would argue that exactly this step enables the method to be robust and versatile as indicated by the various figures and showcases put forth. In **Comment 2.32**, we discuss that only small variations (i.e. spatial structures) with cross-sections that are much smaller than the estimated h_{KDE} and of relatively low intensity would be largely smoothed away. Considering that the computed h_{KDE} for all cases shown in the manuscript was between 2 and 2.6 pixels, this implies that structures that could be largely smoothed away are normally much smaller than two pixels and of relatively low intensity. The question that could also be asked here, would such structures as the ones just described be even considered (if not completely

²⁶ McShane, Blakeley B., et al. "Abandon statistical significance." *The American Statistician* 73.sup1 (2019): 235-245.

²⁷ Chalkley, Robert J. "When target-decoy false discovery rate estimations are inaccurate and how to spot instances." *Journal of proteome research* 12.2 (2013): 1062-1064.

overlooked) as important by expert mass spectrometrists when looking at plain ion images? The answer will, of course, depend on the context, but given the fact that KDE uses location as well as intensity information, overlooking important small variations with high relative intensity (i.e. not random noise) would be unlikely, as shown in various simulation-based and noise-robustness experiments as well as tests on MALDI-MSI data of tissue samples. As a case in point, we examined an amyloid plaque MALDI-TOF dataset using our *plaquepicker* software, which is not KDE bandwidth-dependent, versus *molecularR*. Most notably, some single-pixel low-intensity signals, presumably A β 1-38 at m/z 4060.5, were not considered as hotspots by the latter. Given the known size distribution of amyloid plaques, these single-pixels may very well represent plaques, as we point out in the manuscript. There is, however, no way of knowing this for sure based on the MSI data (at this spatial step-size) alone.

2.36. User-Independent

- "provide a user-independent and unbiased indication of where analytes' intensities have significant relative spatial abundance (or deficiency) that is unlikely to be seen if these signals were generated due to a random event (Fig. 2A)." -> The statement that MPMs would be unbiased does not seem to correspond with the authors' choices made in the algorithm that produces the MPMs. For example, the choice to approximate a peak in a mass spectrum as a Gaussian peak is a choice. It can be a perfectly good choice, but it is a choice, and all subsequent results are biased by that choice. There is a difference between being unbiased and making defensible choices, and the method described here seems to fall in the latter category. Along the way to making an MPM several choices are made, so it becomes quite hard to maintain that the results are truly unbiased. The same argument can be made for the adjective user-independent. Do the authors mean 'automated'? I think the latter could be more defensible than user-independent. Additionally, the user-independent adjective seems less appropriate with author statements like the following "molecularR also permits the user to manually set the KDE bandwidth against a theoretically acceptable value which tends to reduce the spatial constraint described above." If manual setting is a viable and sometimes necessary option, how can the process be user-independent. The automated determination of hKDE would only mean that the choice is taken from the user and performed by the developers of the software, not that it is independent of humans. Please acknowledge these matters in the main manuscript text.

2.36.1. For example, the choice to approximate a peak in a mass spectrum as a Gaussian peak is a choice. It can be a perfectly good choice, but it is a choice, and all subsequent results are biased by that choice. There is a difference between being unbiased and making defensible choices, and the method described here seems to fall in the latter category. Along the way to making an MPM several choices are made, so it becomes quite hard to maintain that the results are truly unbiased. The same argument can be made for the adjective user-independent. Do the authors mean 'automated'? I think the latter could be more defensible than user-independent.

As stated above, we have removed all user-bias and user-dependence terminology from the manuscript.

2.36.2. Additionally, the user-independent adjective seems less appropriate with author statements like the following "molecularR also permits the user to manually set the KDE

bandwidth against a theoretically acceptable value which tends to reduce the spatial constraint described above." If manual setting is a viable and sometimes necessary option, how can the process be user-independent.

see above.

2.37. Negative Example

- "As mentioned above, in our additional mouse brain MALDI FTICR MSI experiment with N=6 technical replicates (=adjacent tissue sections) in New Suppl. Fig. 21, MPM hotspot contours were very comparable. All considerations for MPMs, in principle, also apply for CPPMs." -> A positive example does not show that the method is reliable, it shows that the method is reliable in that specific dataset and choices case. A negative example would be more informative as it would demonstrate when the method starts failing and in which cases it's maybe safer to use.

We agree with the reviewer that negative results are generally more powerful than positive ones in science. (We are fans of Popper's Falsification Principle). Nevertheless, not one but several positive examples have been provided in the manuscript, featuring different modalities, data types, and tissue samples:

1. MALDI-FTICR-MSI data of human GB tissues (**Fig. 3b, Fig. 5, Suppl. Fig. 27, Suppl. Fig. 29-34**).
2. MALDI-FTICR-MSI data of mouse brain tissues (**Fig. 3c, Suppl. Fig. 6, Suppl. Fig. 17, Suppl. Fig. 28**).
3. MALDI-FTICR-MSI data of IDH-mutant and -wildtype human GB tissues (**Suppl. Fig. 18**).
4. MALDI-FTICR-quantification MSI data of porcine liver tissue (**Suppl. Fig. 16**).
5. MALDI-TOF-quantification MSI data of gastrointestinal stromal tumor tissue (**Suppl. Fig. 20**).
6. MALDI-timsTOF-MSI data of mouse brain tissue (**Suppl. Fig. 7, Suppl. Fig. 17**).
7. MALDI-TOF-MSI data of mouse brain tissue (**Suppl. Fig. 8, Suppl. Fig. 17**).
8. MALDI-TOF-MSI data of an APP NL-G-F Alzheimer's disease mouse model (**Fig. 3d, Suppl. Fig. 15**).
9. Simulation data derived from MALDI-FTICR-MSI (**Fig. 2b, Suppl. Fig. 11, Suppl. Fig. 13**) and MALDI-TOF-MSI (**Suppl. Fig. 14**).

As for negative examples, computational tests comparing the method's performance in the presence of artificially added noise have been provided as an attempt to generate negative examples (ex. **Suppl. Fig. 22-24**). Moreover, we have found and shown one example of poor MALDI-TOF data with $R \sim 5000$ (**Suppl. Fig. 8**), where the beneficial effect of Gaussian weighting was not observed. Hence, breaking points clearly exist, but it would be beyond the scope of this manuscript to work this out in even more detail. It should be stated, however, that this example cannot be generalized to MALDI-TOF data in general.

2.38. User-Independent

- "Our data suggests that the proposed Gaussian mass-window weighting can help to resolve MOI ambiguities in a data-driven and user- unbiased way taking into account the mass resolving power at the analyte under study which can vary depending on m/z of the analyte, measurement parameters and the imaging platform being used. We refer editor and reviewers to detailed discussions of new Suppl. Fig. 4-7 above. The considerations above have been acknowledged in various parts of the revised

manuscript." -> I agree that the Gaussian weighting can have a beneficial effect, but it should be more clearly acknowledged in the manuscript that this a choice by the authors and that it biases any followup results from this choice. The same should be stated for any other choices or paramater values. For example, if a choice is the right one for 90% of the datasets in the world, there is still 10% of the datasets for which it is not the right choice. The statements made in the manuscript around being unbiased and about significance seem to suggest otherwise and could be misunderstood by readers.

We appreciate the precision in description the reviewer is calling for. We now understand the reviewer's concern when using the word "unbiased" which the reviewer clearly explained in **Comment 2.36**. As stated above, we have removed all user-bias and user-dependence terminology from the manuscript.

2.39. KDE Bandwidth Against a Ground-truth

- "It must be noted, however, that this is one of the very rare cases, where a user has access to something like a ground-truth" -> This is not a very reassuring statement. If the user does not have a ground truth to compare to, she is to use the value suggested by the software? Whether the parameter values (and their results) are right or wrong should not be influenced by whether or not there's a ground truth to falsify the suggested value with. If the authors want to make an argument around reliability of molecular results, it is important to acknowledge these less than convenient aspects more prominently in the manuscript text so that the reader can evaluate themselves.

Again, we agree with the reviewer that having experimental and not just simulated ground truths would be important for the MSI field as well as for more advanced non-spatial omics technologies. Even for proteomics, this topic is still early days and a very active field of research. For instance, Steve Gygi's lab just published a ground-truth approach to phosphoproteomics in Nature Methods²⁸. We acknowledged this aspect more prominently in the manuscript now.

2.40. Significance

- "This means that while pixel intensities (of secondary as well as primary hotspots) are reduced with increased smoothing bandwidth, the quantile intensity cutoffs, which are used as thresholds for hotspots and coldspots, shrink towards the mean accordingly (Suppl. Fig. 11B)." -> Understood, but it does not mean that the location and 'significance' of the contours is not dependent on the chosen hKDE, and does not change if a different hKDE is used. Using the word 'significance' to describe these (quite variable) contours seems too strong a statement to make.

We thank the reviewer for this remark. It should be stated that the expression "statistically significant" itself is not an absolute term. This is because, generally speaking, statistical significance is directly influenced by factors such as sampling error (ex. sample size), the variation in the underlying population, whether or not p-value adjustment is carried out and the chosen significance level α , a p-value threshold traditionally set to 0.05 but which also varies across disciplines. Within the context of our manuscript, the word "significant" comes from a standard statistical test which is shown in **Fig. 2a**

²⁸ Gassaway BM, Li J, Rad R, Mintseris J, Mohler K, Levy T, Aguiar M, Beausoleil SA, Paulo JA, Rinehart J, Huttlin EL, Gygi SP. A multi-purpose, regenerable, proteome-scale, human phosphoserine resource for phosphoproteomics. Nat Methods. 2022 Nov;19(11):1371-1375. doi: 10.1038/s41592-022-01638-5.

and **Suppl. Fig. 9a** and described in detail in **Methods** (section **Molecular Probabilistic Map**). It is based on null-hypothesis significance testing of each pixel intensity in $\rho_{MOI}(x, y)$ against a significance level α based on the $f_{CSR}(k)$ null distribution, Benjamini-Hochberg corrected. Since we clearly acknowledge that the resulting $f_{CSR}(k)$ depends on the estimated KDE bandwidth, we prefer to stick with the wording “significant”, but to point more prominently to this dependence on setting of a spatial filter. Even in conventional non-spatial data analysis (like ANOVA), data is frequently pre-filtered as part of pre-processing, but nevertheless statistical significance tests are performed.

2.41. Relative Spatial Abundance

- *“This, however, is beyond the scope of this manuscript and the proposed method, as it would require extensive studies of entire (patient) cohorts of samples, which were not available here.” -> Understood, but the wording used to describe the generated results does not lie beyond the scope of this manuscript. The authors changed “significant analyte presence/absence” to “significant relative spatial abundance/deficiency” which seems to be a step in the right direction, but then changes the problem to what is “relative spatial abundance”? Due to the KDE operation, the concept seems to entangle intensity with spatial location, but I did not find a clear definition of what this is exactly. Furthermore, Fig4e seems to suggest that for the CSR of the test tissue the intensities of the reference tissue are used. This would mean that both the spatial location and the content are changed in the significance test for the test tissue and that only the spatial location is changed in the reference tissue. Is that a fair comparison? Does the test remain fair for the test tissue specifically? I understand that the authors want to compare between reference and test tissue, but the significance referred to in the manuscript is about whether the intensity pattern within a tissue is significantly different, so the significance test would need to be compared within-tissue not across tissue. If the authors want to change the definition of what is being tested, further elaboration in the text is necessary I think. Additionally, if the concept is to be named “relative spatial abundance”, elsewhere in the manuscript ‘relative’ seems to refer to within-tissue relative change, while in this part of the manuscript ‘relative’ seems to mean within-tissue and across-tissue combined, which introduces an ambiguity: Is the tryptophan significantly different between test and reference because of across-tissue differences, because of within-tissue difference, or both? Please be precise.*

2.41.1. *The authors changed “significant analyte presence/absence” to “significant relative spatial abundance/deficiency” which seems to be a step in the right direction, but then changes the problem to what is “relative spatial abundance”? Due to the KDE operation, the concept seems to entangle intensity with spatial location, but I did not find a clear definition of what this is exactly.*

We thank the reviewer for this important remark. The mentioned statement appears in **Comment 2.10-Revision1**. We followed a recommendation by the reviewer here. Indeed the manuscript lacked a clear definition for the property “significant relative spatial abundance/deficiency”. At its core, a two-sided statistical test is employed to identify MOI intensities exhibiting spatial patterns/structures that are unlikely to occur randomly by comparing them to a corresponding complete spatial randomness model. Therefore, significant relative spatial abundance or deficiency of a given MOI could be described as the presence of statistically significant non-random spatial patterns of accumulation or depletion of MOI intensities within the tissue sample. The above description was added to the revised manuscript.

2.41.2. Furthermore, Fig4e seems to suggest that for the CSR of the test tissue the intensities of the reference tissue are used. This would mean that both the spatial location and the content are changed in the significance test for the test tissue and that only the spatial location is changed in the reference tissue. Is that a fair comparison? Does the test remain fair for the test tissue specifically? I understand that the authors want to compare between reference and test tissue, but the significance referred to in the manuscript is about whether the intensity pattern within a tissue is significantly different, so the significance test would need to be compared within-tissue not across tissue. If the authors want to change the definition of what is being tested, further elaboration in the text is necessary I think. Additionally, if the concept is to be named “relative spatial abundance”, elsewhere in the manuscript ‘relative’ seems to refer to within-tissue relative change, while in this part of the manuscript ‘relative’ seems to mean within-tissue and across-tissue combined, which introduces an ambiguity: Is the tryptophan significantly different between test and reference because of across-tissue differences, because of within-tissue difference, or both? Please be precise.

We thank the reviewer for this important remark. In **Comment 2.17**, the reviewer has also provided a thorough description of his/her concern. Based on that, we have revised the proposed cross-tissue testing method, provided separate definitions for within-tissue and cross-tissue MPMs, updated the associated figures (**Fig. 3e** and **Suppl. Fig. 18**) and provided more detail in **Methods** (new section **Cross-tissue MPM**) and new Supplementary figures (**Suppl. Fig. 19** and **Suppl. Fig. 20**) that illustrate the idea behind it. We have also provided a thorough discussion in **Comment 2.17**.

2.42. Error Propagation

- "According to the laws of error propagation (linear, Gaussian, other) arithmetic operations may always add (relative or absolute) uncertainty. However, we expect them to have less impact on the resulting image intensities of MPMs (because of the Gaussian mass-window weighting; comments 2.2.1 and 2.4) and on the hotspot/coldspot contours (because of the internal KDE smoothing; comment 2.2.2) when compared to state-of-the-art ion image procedures." -> Why would this expectation be a valid assumption? Do the authors have a specific methodology in mind they are comparing to? Also, the argument of uncertainty is not a matter of comparing to what other methods do. The point is specific to the arithmetic operations suggested in this manuscript (but would also hold for arithmetic operations directly on ion images). For example, dividing by a denominator with an uncertainty connected to it will often have an amplifying effect on the uncertainty of the ratio, particularly if the denominator takes small values. The most extreme case would be if zero falls inside the uncertainty range of the denominator, where you would get immediate amplification of uncertainty of the ratio value to include infinity. This type of effect is orthogonal to whether the nature of the uncertainty is understood, and even if zero would not fall inside the uncertainty of the denominator the amplification effect on the ratio is still there, just less exorbitant. The manuscript should make clear that performing mathematical operations on ion intensities, including some of the ratios being taken in the examples, can have an uncertainty amplifying effect. This point should particularly be made in the main manuscript since the text makes an explicit argument for reliability and significance assessment.

We thank the reviewer for this important remark. The mentioned statement appears in **Comment 2.11-Revision1**. In **Comment 2.11-Revision1**, we provide a discussion on why we think that the mentioned uncertainty might have less impact on ratios computed with *moleculaR* in comparison to the ones generated with classical ion images. Perhaps most importantly, we explain what precautions we have taken to ensure that the stated most extreme case of zero falling inside the uncertainty range

of the denominator is virtually ruled out. However, we fully acknowledge that uncertainty will still be associated with such operations. A word of caution has been added to the main text and reiterated in **Methods** in the revised manuscript.

2.43. P-value Correction

- "We have therefore updated the method and used the Benjamini-Hochberg P-value correction approach throughout the revised manuscript." -> The change of the p-value correction approach directly influencing the MPM results demonstrates the point I've tried to make elsewhere that the results of this approach are not unbiased, they are biased by the choices made by the developers, e.g. the choice of p-value correction. Please avoid calling these approaches unbiased.

As stated above, we have removed all user-bias and user-dependence terminology from the manuscript.

We do note, however, that virtually all statistical tests are carried out with a chosen p-value correction. In many omics fields, the Benjamini-Hochberg correction is the method of choice. This type of correction is generally not considered an unwarranted bias.

2.44. CSR Points Locations

- *"While MOI SPPs inherit the gridded spatial locations from MSI data, creating a CSR model on a spatial grid would directly violate the randomness criterion of such models. Therefore, in order to facilitate the direct and homogeneous comparison between MOI and a corresponding CSR model, MOI spatial intensities must be converted into an SPP representation" -> This argument for SPPs seems to be contradicted by the authors elsewhere where they state "For simplicity, random sampling is replaced by randomly permuting the intensities, which basically has the effect of spatial reshuffling of points, until they assume a spatial uniform Poisson process, thus effectively dissolving any spatial clustering or autocorrelation of signals (Fig.2A)." This latter operation can happen on a regular spatial grid, so does that then not "directly violate the randomness criterion of such models"? I seem to misunderstand what is meant here. Please explain.*

We thank the reviewer for this remark. The mentioned statement appears in **Comment 2.16-Revision1**. We believe this misunderstanding has been thoroughly discussed in **Comment 2.28**. We have updated that statement in **Methods** for a more precise description.

2.45. filterDuplicates

- *"Here, and especially for CPPMs, molecularR provides simple tools to filter out duplicated counts of the same m/z value by incorporating only the intensities of unique masses present in the computed CPPM." -> Were these tools turned on for the examples in the manuscript? Please mention them there more prominently.*

We thank the reviewer for this remark. The mentioned statement appears in **Comment 2.19-Revision1**. The mentioned filtering tools were indeed applied to all examples shown in the manuscript, which is now mentioned in **Methods (Collective Projections Probabilistic Map section)**. This is

accomplished via the method *filterDuplicates* in the *moleculaR* package, which has been also described in the code vignette provided with the published package.

2.46. Objective

- "We also explain how MPMs provide the user with the hotspot/coldspot contours that objectively evaluate the spatial relative abundance/ deficiency comparing that to the "classical" user perception-biased way of interpreting analytes distribution by visual inspection." -> The word 'objective' is subject to the same criticism of how the word 'unbiased' is used in the manuscript. See my remarks elsewhere.

We thank the reviewer for this remark. The word "objective" only appeared in the rebuttal, but never in the actual manuscript. The mentioned statement appears in **Comment 2.51-Revision1**. We acknowledge the reviewer's concern and have updated the revised manuscript accordingly. Please also see **Comment 2.3, 2.4**.

REVIEWERS' COMMENTS

Reviewer #2 (Remarks to the Author):

The authors have addressed most of my comments. The manuscript seems suitable to progress to publication.

Some minor notes on the rebuttal

"This was generated during data acquisition by the measurement device with $\text{SNR} \geq 3$, which is commonly accepted in the MSI community as a reasonable threshold for peak detection. Therefore, the developer-made choices here are not arbitrary; they are based on what is commonly accepted in the MSI literature and on what is scientifically justified in comparison to theoretical expectation (Suppl. Fig. 4A)." -> I'm not a fan of this type of argument. Something being commonly accepted in the community is not necessarily the same as that something being good/correct/optimal.

"we distinguish between user-made and developer-made choices" -> Why? Are developers inherently more knowledgeable? I'm not sure this stance necessarily holds, particularly for certain sub-areas of mass spectrometry I think. Also, developer-made parameter choices carry some danger in that they are often less accessible for adjustment by a user, will tend to cater to the majority of use cases (but sacrificing applicability in non-standard cases), and in many cases users might not be aware certain developer-made choices were made for them. Stating developer-made choices explicitly, but also investigating/stating the impact these choices may/can have on results is important I think.

"we would be open to suggestions for other, perhaps more fitting terminology." -> I don't think the use of the word 'probabilistic' is correct in this context, particularly since uncertainty is not tracked in these MPM images. Instead of Molecular Probabilistic Maps, something like Molecular Localization Maps or Molecular Hot/coldspot Maps seems more telling of what is happening.

Point-for-Point Reply to Reviewers – Revision #3

1. Reviewer #2

1.1. Comment

The authors have addressed most of my comments. The manuscript seems suitable to progress to publication.

We appreciate all the excellent constructive points raised by the reviewer in all three revision rounds and thank him/her for recommending our manuscript for publication.

Some minor notes on the rebuttal:

1.2. Comment

"This was generated during data acquisition by the measurement device with $SNR \geq 3$, which is commonly accepted in the MSI community as a reasonable threshold for peak detection. Therefore, the developer-made choices here are not arbitrary; they are based on what is commonly accepted in the MSI literature and on what is scientifically justified in comparison to theoretical expectation (Suppl. Fig. 4A)." -> I'm not a fan of this type of argument. Something being commonly accepted in the community is not necessarily the same as that something being good/correct/optimal.

We thank the reviewer for this important remark. This statement appears in **Comment 2.5.2-Revision2**. Spectral centroiding (a.k.a. peak picking) is a process of converting a spectral peak (peak or not being decided by an algorithm) presumably corresponding to an ion to a centroid m/z value with an associated intensity. Most vendors of mass spectrometers often include centroiding in their measurement platforms, which reduces data size and facilitates data storage and handling especially for high-resolving power analyzers (e.g. FTICR, Orbitrap). To our knowledge, centroiding based on $SNR \geq 3$ is the *de facto* default for most measurement platforms as well as commercial and research software packages. Historically, this seems to come from the IUPAC guidelines for defining the limit of detection for quantifying trace elements using analytical methods, which is normally set to $\mu + 3\sigma$, where μ and σ are the mean and standard deviation of the signal of the blank (i.e. detector noise). This allows for a confidence level of >99% that a detected signal above that threshold is not a random event (three-sigma rule).¹ The guidelines mentioned above assume a normal distribution of the blank signal, which may not be the case for MALDI MSI noise (as previously pointed out by the reviewer in **Comment 2.9-Revision1**). However, even for general non-normally distributed variables, Chebyshev's inequality ensures a confidence level of at least 88.8% while for any unimodal distribution (e.g. Poisson-distributed noise), the probability that a detected signal above the three-sigma threshold is not a random event is at least 95% by the Vysochanskij–Petunin inequality. Setting SNR to higher levels might further increase the confidence in the detected signals, however, this will also increase false negatives deeming some possibly valid (i.e. non-random) signals as noise.

¹ Long, Gary L., and James D. Winefordner. "Limit of detection. A closer look at the IUPAC definition." *Analytical chemistry* 55.7 (1983): 712A-724A.

Having said, we do agree with the reviewer that commonly accepted does not necessarily mean optimal. However, the commonly accepted value of $SNR \geq 3$ as a threshold for centroiding does not seem to be lacking a sound scientific basis and, to our knowledge, this specific threshold has not been refuted by any previous study.

1.3. Comment

"we distinguish between user-made and developer-made choices" -> Why? Are developers inherently more knowledgeable? I'm not sure this stance necessarily holds, particularly for certain sub-areas of mass spectrometry I think. Also, developer-made parameter choices carry some danger in that they are often less accessible for adjustment by a user, will tend to cater to the majority of use cases (but sacrificing applicability in non-standard cases), and in many cases users might not be aware certain developer-made choices were made for them. Stating developer-made choices explicitly, but also investigating/stating the impact these choices may/can have on results is important I think.

We thank the reviewer for this remark. The mentioned statement appears in **Comment 2.6-Revision2** but is also linked to **Comments 2.2-** and **2.3-Revision2**. We completely agree with reviewer that method design considerations and general usage choices must always be clearly stated and described for both developers/bioinformaticians as well as end-users. The statement above was meant to make a clearer distinction between the possible users of the proposed framework which was not clearly outlined in the first round of revision but was thoroughly discussed in **Comments 2.2-Revision1**. We do not disagree with the reviewer that any software is developer-dependent and for this reason all method design choices must be well thought through, transparent and clear to the end-user. In the manuscript and revision rounds, we have tried our best to provide supporting evidence for every method design consideration ("developer-made choices") with supporting literature, argument, illustration, simulation and MSI data of different modalities and tissues.

1.4. Comment

"we would be open to suggestions for other, perhaps more fitting terminology." -> I don't think the use of the word 'probabilistic' is correct in this context, particularly since uncertainty is not tracked in these MPM images. Instead of Molecular Probabilistic Maps, something like Molecular Localization Maps or Molecular Hot/coldspot Maps seems more telling of what is happening.

We thank the reviewer for this remark. The mentioned statement appears in **Comment 2.13-Revision2**. Our method includes generating an MPM image and an empirical probability distribution of MPM intensities under the null hypothesis. We use this probability distribution as an intensity scale that allows to translate an MPM intensity to the estimated probability that the corresponding spot is in the interior of a hot- or coldspot region. Thus, the method allows to derive a probabilistic interpretation of the intensities represented by the MPM image, which is why we consider the terminology substantiated and appropriate.